# CodecSep: Prompt-Driven Universal Sound Separation on Neural Audio Codec Latents

## Abstract

Text-guided sound separation supports flexible audio editing across media and assistive applications, but existing models like AudioSep are too compute-heavy for edge deployment. Neural Audio Codec-based models such as CodecFormer and SDCodec are compute efficient but limited to fixed-class separation. We introduce CodecSep, the first NAC-based model for on-device universal, text-driven separation. CodecSep combines DAC compression with a transformer masker modulated by CLAP-derived FiLM parameters. Across six open-domain benchmarks under matched training/prompt protocols, CodecSep surpasses AudioSep in separation fidelity (SI-SDR) while remaining competitive in perceptual quality (ViSQOL) and matching or exceeding fixed-stem baselines (TDANet, Sudo rm-rf, CodecFormer, SDCodec). In code-stream deployments, it needs just 1.35 GMACs end-to-end—$\sim54\times$ less compute ($25\times$ architecture-only) than spectrogram domain separators like AudioSep—while remaining fully bitstream-compatible.

## 1 Introduction

We propose CodecSep, a text-conditioned universal sound separation (USS) framework that marries the interpretability of prompt-driven extraction with the efficiency of neural audio codecs (NACs). To our knowledge, CodecSep is the first system to bridge NACs with USS: it conditions a transformer *masker* on CLAP text embeddings Wu et al. (2023) via Feature-wise Linear Modulation (FiLM) Perez et al. (2018), and performs separation directly in the codec encoder latent space. This design introduces semantic control while preserving the low compute footprint of codec representations, enabling on-device and real-time deployment.

Flexible, real-time separation on bandwidth- or compute-constrained platforms remains challenging. Classic models disentangle sources from complex mixtures Vincent et al. (2018) but are often domain-specific (e.g., speech/music) and heavy. Recent text-guided systems like AudioSep Liu et al. (2024) extend encoder–masker–decoder designs (e.g., Conv-TasNet-style Luo & Mesgarani (2019)) by injecting semantics from BERT/CLAP through FiLM layers Devlin et al. (2019); Wu et al. (2023); Perez et al. (2018). However, spectrogram/waveform-domain separators trained with SI-SDR-style losses Luo & Mesgarani (2019); Le Roux et al. (2019) are compute-intensive and sensitive to compression artifacts, often pushing inference to the cloud.

NACs such as SoundStream, Encodec, and DAC Zeghidour et al. (2022); Défossez et al. (2022); Kumar et al. (2023) compress audio to discrete tokens with Residual Vector Quantization (RVQ), providing compact, perceptually aligned latents useful for generation and conditioned synthesis Borsos et al. (2023); Wang et al. (2023; 2024); Du et al. (2024). Prior codec–separation hybrids (CodecFormer Yip et al. (2024b), SDCodec Bie et al. (2024)) are lightweight and high-fidelity but target fixed stems (e.g., speech separation or speech vs. music vs. SFX); extending them to open-domain, prompt-conditioned USS is non-trivial (cf. §2, para. 3).

CodecSep adopts a frozen DAC encoder–decoder backbone and inserts a FiLM-conditioned transformer masker that predicts a soft mask over codec latents. CLAP-derived text embeddings Wu et al. (2023) are mapped to per-layer FiLM parameters, modulating the masker's intermediate activations to align the selected latent subspace with the query semantics. Operating on compact codec features cuts memory traffic and MACs compared to spectrogram-domain pipelines, while preserving the codec's inductive biases (periodicity, timbre, transients). In doing so, CodecSep delivers interpretable, prompt-guided separation with markedly lower compute without sacrificing separation fidelity. Crucially, conditioning via text embeddings enables open-vocabulary operation of NAC-based separation.

We evaluate CodecSep across: (i) *in-domain text-guided separation* on dnr-v2 Petermann et al. (2022); (ii) *cross-domain generalization* on five open-domain corpora (AudioCaps Kim et al. (2019), ESC-50 Piczak, Clotho-v2 Drossos et al. (2020), AudioSet-eval Gemmeke et al. (2017), VGGSound Chen et al. (2020a)); (iii) *three prompt granularities* (fixed-stem, generic three-stem, and fine-grained SFX); (iv) *paraphrase robustness*; (v) an *architectural ablation* comparing decoder-style generation to our transformer masker; (vi) a *bandwidth-scaling study* extending CodecSep to higher sampling rates; and (vii) *compute benchmarking* on small GPUs. We benchmark against the SOTA text-guided baseline, AudioSep Liu et al. (2024), under matched data and prompt protocols. Across all benchmarks, CodecSep consistently surpasses AudioSep in SI-SDR while remaining competitive in ViSQOL, and it degrades more gracefully under prompt paraphrasing. In deployment-typical code-stream settings, CodecSep runs at just 1.35 GMACs end-to-end—∼54× less compute (and ∼25× architecture-only) than AudioSep—while remaining fully compatible with bitstream interfaces.

## 2 RELATED WORK

Classical sound separation systems frequently adopt an encoder–masker–decoder design in which an encoder produces STFT-like latents, a masker predicts source-specific masks, and a decoder reconstructs waveforms. Representative models include DPTNet Chen et al. (2020b), SepFormer Subakan et al. (2021), and TDANet Li et al. (2023), the last introducing a top-down attention scheme that blends global and local attention to capture multi-scale acoustic structure. Beyond masking pipelines, several works generate waveforms directly in the time domain (Wave-UNet Stoller et al. (2018), Demucs Défossez et al. (2019); Défossez et al. (2021)) or operate fully in the complex STFT domain with joint magnitude–phase modeling (MM-DenseLSTM Takahashi et al. (2018), DCCRN Hu et al. (2020), Spleeter Hennequin et al. (2020)), underscoring the breadth of design choices.

Moving from domain-specific separation to *universal* sound separation (USS), supervised systems typically rely on Permutation Invariant Training (PIT) Yu et al. (2017); Kavalerov et al. (2019), while unsupervised methods such as MixIT Wisdom et al. (2020) learn directly from mixtures. Both paradigms assume a fixed maximum number of sources and output all estimates indiscriminately, requiring a post-hoc identification step (cf. Appendix A for detailed failure modes). A recent PIT-trained USS model is Sudo rm-rf Tzinis et al. (2022a), a parameter-efficient time-domain separator that is based on ConvTasNet-style encoder–masker–decoder architecture with adaptive encoder/decoder modules. It downsamples input waveform to STFT-like latents before separation and is PIT-trained to separate mixtures with up to four sources. Query-Guided Sound Separation (QSS) addresses this limitation of PIT or MixIT models by conditioning extraction on external queries—visual cues, audio, class labels, or text. Text queries are compact, expressive, and capture high-level semantics without requiring additional reference signals. AudioSep Liu et al. (2024) follows this direction by injecting BERT Devlin et al. (2019) or CLAP Wu et al. (2023) embeddings via FiLM Perez et al. (2018) at intermediate masker layers to steer separation toward the query source. BiModalSS Mahmud et al. (2024) extends AudioSep with attention-based conditioning and more efficient training strategies. FiLM-conditioned variants of Sudo rm-rf Tzinis et al. (2022b; 2023) have also been explored for class-guided separation using one-hot or multi-hot conditioning vectors; however, such label-based conditioning does not extend naturally to open-domain text queries and cannot handle unseen classes.

Neural audio codecs (NACs) have recently been integrated into separation pipelines to improve efficiency. CodecFormer separates directly in DAC Kumar et al. (2023) latent space with a transformer trained using negative SI-SDR, and CodecFormer-EL Yip et al. (2024a) adds an embedding-level objective to align separator outputs with encoder latents. SDCodec Bie et al. (2024) embeds separation inside the codec by assigning dedicated RVQ branches to speech, music, and SFX and summing their codes to form mixture representations. However, neither design trivially extends to open-domain, prompt-conditioned USS: SDCodec's hardwired, per-stem RVQ branches do not scale to open vocabularies (adding branches explodes parameters, while a mixture-invariant reformulation collapses into "three generic codecs" with no stem-specific RVQ specialization). MixIT-style training of CodecFormer still presupposes a maximum number of sources, conflicting with universal separation.

## 3 METHOD

CodecSep adapts the 16 kHz DAC Kumar et al. (2023) codec backbone for text-driven universal sound separation (USS). We use a transformer masker that estimates soft masks over codec latents

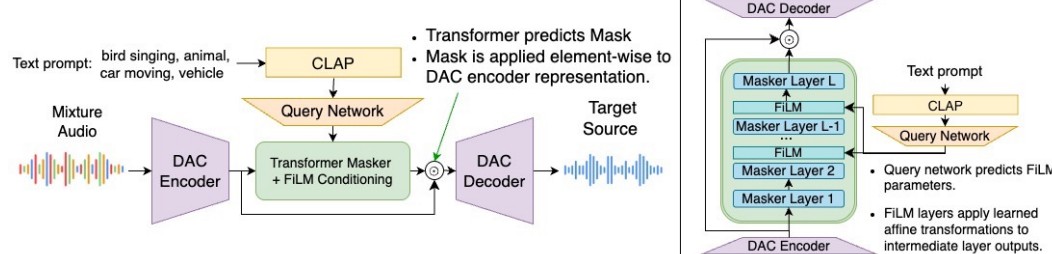

Figure 1: An overview of CodecSep. (Left) The full pipeline for text-guided USS. (Right) The integration of text conditioning into intermediate layers of transformer masker via FiLM layers.

and inject text conditioning via Feature-wise Linear Modulation (FiLM) Perez et al. (2018) using CLAP text embeddings Wu et al. (2023). FiLM is applied to intermediate transformer activations so the query semantics steer separation. Figure 1 (Left) shows the overall text-guided pipeline; Figure 1 (Right) highlights the FiLM-conditioned masker. Compared to STFT-domain AudioSep, operating in compact codec latents yields markedly lower compute and is amenable to edge deployment.

### 3.1 DESIGN RATIONALE: FiLM-CONDITIONED MASKING IN NAC LATENT SPACE

**Descript Audio Codec (DAC) Backbone.** We use DAC Kumar et al. (2023) as encoder–decoder. Following Encodec/SoundStream, DAC uses fully-convolutional encoder/decoder with the periodic Snake activation $x + \sin^2 x$ (replacing LeakyReLU) to bias periodic audio modeling. Residual vector quantization (RVQ) compresses encoder outputs with factorized codes and $\ell_2$-normalized codebooks. For a 1 s audio $x(t)$ at $F_s{=}24$ kHz compressed to $R{=}6000$ bps, the encoder $Enc(\cdot)$ downsamples by $M{=}320$ to $T{=}F_s/M{=}75$ frames of latents $Z=[z_t \in \mathbb{R}^d]_{t=1}^T$ ($d$: channel width) with $r{=}R/T{=}80$ bits/frame. RVQ allocates $r_i{=}r/N_q{=}10$ bits across $N_q{=}8$ codebooks (size $2^{10}{=}1024$). Given $Z$, $Quant(\cdot)$ yields discrete codes $A=[a_t \in [1024]^8]$, which map to embeddings $e_t = \sum_{i=1}^8 e_t^i$; $Dec(\cdot)$ upsamples $E=[e_t]$ back to waveform $y(t)$.

**Text-guided Sound Separation in Spectrogram-domain (AudioSep):**

$$x(t) \xrightarrow{\text{STFT}} X \in \mathbb{C}^{F \times T_{spec}} \xrightarrow{Spec(X, e_\tau)} \tilde{Y}_s = |\hat{M}_s| \odot |X| \exp\left(\angle X + \angle \hat{M}_s\right) \xrightarrow{\text{ISTFT}} \tilde{y}_s(t),$$

where $Spec(\cdot, e_\tau)$ is a FiLM-conditioned masker that predicts a magnitude mask $|\hat{M}_s| \in [0,1]^{F \times T_{spec}}$ ($F$: frequency bins, $T_{spec}$: spectrogram frames) and a phase residual $\angle \hat{M}_s$ given the complex STFT $X$ of audio $x(t)$ and text-embedding $e_\tau$.

**Text-guided Sound Separation in NAC latent-domain (CodecSep):**

$$x(t) \xrightarrow[\text{DAC}]{Enc(\cdot)} Z \in \mathbb{R}^{d \times T} \xrightarrow{Mask(Z, e_\tau)} \tilde{Z}_s = M_s \odot Z \xrightarrow[\text{DAC}]{Dec(\cdot)} \tilde{y}_s(t),$$

with frozen DAC backbone $Enc(\cdot), Dec(\cdot)$ and a FiLM-conditioned transformer masker $Mask(\cdot, e_\tau)$ that estimates mask $M_s \in [0,1]^{d \times T}$ applied element-wise to DAC latent $Z$.

**Why NAC latents vs. spectrograms.** Operating on NAC latents $Z$ slashes dimensionality while preserving perceptual factors. For 1 s audio at 32 kHz, complex STFT with $N{=}1024$ and hop size $M{=}320$ samples has $T_{\text{spec}} \approx 100$ frames and $F = 2 \times 1024$ (Re+Im) scalars per frame, so $F \cdot T_{spec} \approx 204{,}800$. A 16 kHz DAC with width $d{=}64$ and the same $M$ yields $T \approx 50$ and $d \cdot T{=}64 \times 50{=}3{,}200$ ($\sim 64\times$ smaller), shrinking $Q/K/V$ and MLP sizes and easing self-attention. Similarly, for 32 kHz NACs like EnCodec, $T \approx 50$ with $d{=}128$, so attention/MLPs still operate on $\sim 32\times$ smaller latents than complex STFTs. Crucially, $Enc(\cdot)$ organizes $Z$ on a discriminative, perceptually aligned manifold , making selection (masking) easier than representation learning from raw $X$. In spectrogram systems $Spec(.)$, the separator must first learn a high-level latent from $X$ (via CNN/UNet) and then separate, coupling abstraction and masking and inflating parameters/compute. Waveform separators $Wave(.)$ such as Sudo rm-rf Tzinis et al. (2022a;b; 2023) likewise downsample the waveform into STFT-like intermediate latents via 1D convolutions before encoding, resulting in latents with similar dimensionality and thus inheriting the same challenges as spectrogram systems.

**Masking over codec latents (leveraging the codec prior).** Because the DAC codec induces a strong semantic prior in its latent space via residual vector quantization (RVQ) and perceptual/adversarial

training, we mask the codec latents rather than generate sources from scratch as in CodecFormer. RVQ creates a coarse-to-fine hierarchy in $Z=Enc(x) \in \mathbb{R}^{d \times T}$: early stages capture coarse structure (e.g., low-frequency content, timbre), while later stages refine residual detail (e.g., high-frequency components, transients) Wang et al. (2023). We exploit this organization with a FiLM-conditioned transformer masker that predicts a soft mask $M_s \in [0,1]^{d \times T}$ and applies it element-wise, yielding source latent estimate $\tilde{Z}_s = M_s \odot Z$. In contrast to learning a generator $Gen(Z, e_\tau) : Z \rightarrow \tilde{Z}_s$ as in CodecFormer, learning a *mask* $Mask(Z, e_\tau) : Z \rightarrow M_s$ on the compact, semantically organized codec manifold both exploits the codec prior more effectively and yields a more stable optimization that converges faster. Moreover, masking in the denoised, low-dimensional codec space is fundamentally easier than masking in the high-dimensional, noisy spectrogram domain. This selection-centric design (i) constrains learning to modulation of existing latent content, (ii) avoids hallucination and reduces leakage because no new signals are synthesized, and (iii) preserves long-horizon structure (periodicity, timbre, transients) already organized by the codec, yielding stable, low-artifact separations.

**Why FiLM inside the masker.** Placing FiLM in the masker (not in $Enc/Dec$) targets the selection step, preserves the codec manifold, and adds negligible overhead (two vectors per layer). Conditioning is non-iterative (single forward pass), maintaining low latency for edge/server workflows.

**Continuous latents $Z$ vs. discrete codes $A$ (and deployment path).** For training and analysis, we operate on *continuous* latents $Z=Enc(x) \in \mathbb{R}^{d \times T}$: (i) gradients flow cleanly through $Mask(\cdot, e_\tau)$ and $Dec(\cdot)$ with a frozen codec (no straight-through estimators), yielding stable convergence; (ii) RVQ pretraining *regularizes $Z$* so pitch, timbre, onsets/transients, and textures are hierarchically organized, providing a richer, more disentangled signal for FiLM; and (iii) $Z$ avoids run-to-run variance from codebook utilization (e.g., late RVQ sensitivity, bitrate truncation), reducing the need for special regularizers. For deployments with compressed bitstreams, we reconstruct embeddings by codebook lookup and use the same masker:

$$A = [a_t \in [1024]^{N_q} \mid t \in [T]], \qquad e_t \;=\; \sum_{i=1}^{N_q} \text{lookup}\big(a_t^{(i)}\big), \tag{1}$$

$$E = [e_t]_{t=1}^{T} \;\approx\; Z, \qquad \tilde{E}_s \;=\; M_s \odot E, \qquad \tilde{y}_s(t) \;=\; Dec(\tilde{E}_s) \tag{2}$$

When a codes-out interface is desired, we re-quantize masked embeddings and optionally decode:

$$\hat{A}_s = Quant(\tilde{E}_s), \qquad \hat{E}_s = \text{lookup}(\hat{A}_s), \qquad \tilde{y}_s(t) = Dec(\hat{E}_s).$$

By design, $E \approx Z$ at the operating bitrate and $Dec(E)$ already yields high-fidelity reconstructions; because our separator is a masker (selective modulation) rather than a generator, swapping $Z \rightarrow E$ preserves the semantics needed for separation with no architectural change. While we report results on $Z$ to isolate separator performance and maintain stable optimization, we also evaluate the bitstream path by feeding reconstructed embeddings $E$ (codes-in) to the same trained masker without any fine-tuning; performance remains competitive relative to the $Z$ path. The residual gap can be narrowed with light fine-tuning the masker on $E$ or optimizing an embedding-consistency loss (cf. CodecFormer-EL) in place of, or alongside, SI-SDR: $\mathcal{L}_{\text{emb}} = \sum_s \|\tilde{E}_s - Z_s\|_1$. In deployment, the variant simply replaces the masker input with $E$ and optionally re-quantizes for codes-out as,

$$x(t) \xrightarrow[\text{On Edge}]{Quant(Enc(\cdot))} A \xrightarrow[\text{Codes In}]{lookup(A)} E \approx Z \xrightarrow[\text{On Server}]{Mask(E, e_\tau)} \tilde{E}_s = M_s \odot E \xrightarrow[\text{Codes Out}]{Quant(\tilde{E}_s)} \hat{A}_s.$$

**Deployment advantages over audio-stream separators.** In realistic pipelines, edge devices already run a codec and transmit code streams rather than raw audio. Traditional spectrogram-based $Spec(.)$ and waveform-based $Wave(.)$ separators, however, operate on the audio stream: they first convert audio to STFT or STFT-like representations (often via 1D convolutions), and then must *decode $\rightarrow$ separate on $X \rightarrow$ re-encode*, incurring additional latency and energy cost. In contrast, CodecSep performs *masking directly in the codec domain* and can output code streams without any decode–re-encode cycle. Concretely, with codec costs $C_{Enc}, C_{Dec}$, spectrogram or audio-stream separator (AudioSep) cost $C_{Spec}$, and CodecSep masker cost $C_{Mask}$:

Compute Cost for Code-stream input: AudioSep $= C_{Dec} + C_{Spec} + C_{Enc}$, CodecSep $= C_{Mask}$.

We treat the codebook lookup $C_{lookup}$ and quantization $C_{Quant}$ costs as negligible ($\approx 0$) and omit the CLAP text-encoder cost since it is shared across all models. Figure 2 illustrates a typical edge–server

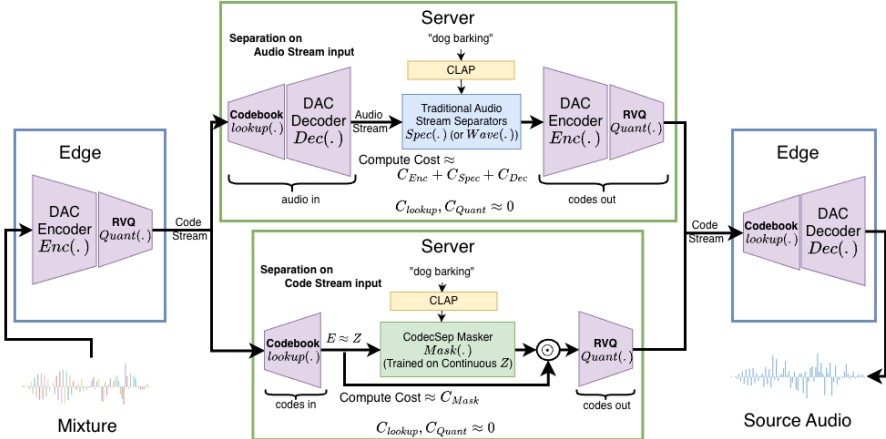

Figure 2: Typical edge–server deployment comparing compute requirements of conventional audio-stream separators (audio in → codes out) versus CodecSep discrete inference (codes in → codes out).

deployment and compares compute requirements for conventional audio-stream separators (audio in → codes out) versus CodecSep's code-stream separator (codes in → codes out). As shown, the CodecSep masker operates on $Z/E$ with small $(d, T)$ where $|Z| \ll |X|$, dramatically reducing attention and MLP activations and enabling tighter batching and lower memory bandwidth. Interface compatibility is immediate when only codes $A$ are available: perform a lookup to obtain $E$, apply the FiLM-conditioned masker, and optionally re-quantize to produce $\hat{A}_s$. CodecSep thus eliminates redundant decode/re-encode cycles in server workflows yielding low-latency, high-fidelity separation at scale. See Appendix B for a full discussion covering all of the aforementioned rationale in §3.1.

## 3.2 OUR MODEL

**Masking (Not Generating): FiLM-Conditioned Transformer masker over NAC Latents for text-guided sound separation.** We consider a mono mixture as $x(t) = \sum_{s \in \mathcal{S}} y_s(t)$, where $x(t)$ is the observed waveform, $\mathcal{S}$ is the (unbounded) set of source classes/instances present, and $y_s(t)$ is the waveform of source $s$. Passing $x(t)$ through the frozen DAC encoder yields codec latents $Z \in \mathbb{R}^{d \times T}$ ($d$: channel width, $T$: latent frames). The masker $Mask(\cdot)$ operates in this latent space to predict an element-wise mask that selects the target source. We adopt a CodecFormer-style transformer with $L{=}16$ layers, width $d{=}256$, and Snake activations. Given a natural-language query $\tau$ (e.g., "dog barking", "speech and music"), we compute a CLAP text embedding $e_\tau \in \mathbb{R}^d$. A lightweight query network $query(.)$ maps $e_\tau$ to per-layer FiLM parameters $(\gamma^l, \beta^l) \in \mathbb{R}^d$ for $l \in \{2, \ldots, L{-}1\}$, applied channel-wise to intermediate activations $H^l \in \mathbb{R}^{d \times T}$: $\tilde{H}^l = \text{FiLM}(H^l; \gamma^l, \beta^l) = \gamma^l \odot H^l + \beta^l$. The final transformer output $H^L$ is then passed through a single 1D convolutional head to produce the prompt-conditioned mask $M_s \in [0, 1]^{d \times T}$. We obtain source latents $\tilde{Z}_s = M_s \odot Z$ and decode with the frozen codec decoder to get the waveform estimate $\tilde{y}_s = Dec(\tilde{Z}_s)$, bypassing RVQ lookup.

**Training Objective** We supervise on mixtures with prompts spanning *speech*, *music*, and diverse (possibly compositional) *SFX*. Besides per-source reconstruction, we encourage mixture consistency by decoding the summed latent estimates $\tilde{x} = g(\sum_s \tilde{Z}_s)$. The loss maximizes SI-SDR Luo & Mesgarani (2019); Le Roux et al. (2019) for both sources and mixture: $\mathcal{L} = -\sum_s \text{SI-SDR}(y_s, \tilde{y}_s) - \text{SI-SDR}(x, \tilde{x})$. During training, DAC and the CLAP text encoder are frozen; we update only the FiLM-conditioned masker $Mask(\cdot)$ and the query network $query(\cdot)$.

## 4 EXPERIMENTS

**Datasets.** We evaluate across a controlled multi-stem corpus and multiple open-domain benchmarks. For in-domain experiments, we adapt Divide and Remaster v2 (dnr-v2) Petermann et al. (2022) from fixed-label three-stem separation to universal, prompt-driven separation by replacing source labels with natural-language queries. Speech and music are queried with broad category prompts (e.g., "speech", "music"), while SFX stems are queried using long-form, compositional prompts ($\geq 2$

Table 1: Results: Separation Performance, Universal Sound Separation (**dnr-v2-test**)

| Model | Metric (↑) | Music | Speech | Sfx |
|---|---|---|---|---|
| AudioSep | SI-SDR | $-2.5^{\pm 4.06}$ | $4.9^{\pm 4.21}$ | $-0.3^{\pm 5.39}$ |
| (zero-shot) | ViSQOL | $2.9^{\pm 0.63}$ | $3.1^{\pm 0.56}$ | $\mathbf{2.6^{\pm 0.77}}$ |
| Bimodal | SI-SDR | $-6.8^{\pm 2.73}$ | $1.8^{\pm 2.78}$ | $-6.36^{\pm 3.57}$ |
| (zero-shot) | ViSQOL | $2.5^{\pm 0.57}$ | $2.6^{\pm 0.51}$ | $2.3^{\pm 0.72}$ |
| AudioSep + dnr-v2 | SI-SDR | $-5.6^{\pm 2.89}$ | $7.7^{\pm 3.0}$ | $-4.7^{\pm 3.68}$ |
| | ViSQOL | $2.6^{\pm 0.57}$ | $2.5^{\pm 0.37}$ | $2.3^{\pm 0.7}$ |
| Sudo rm-rf + FiLM + dnr-v2 | SI-SDR | $-6.7^{\pm 2.62}$ | $2.0^{\pm 2.76}$ | $-6.6^{\pm 3.71}$ |
| | ViSQOL | $2.7^{\pm 0.59}$ | $2.9^{\pm 0.45}$ | $2.3^{\pm 0.72}$ |
| **CodecSep + dnr-v2** | SI-SDR | $\mathbf{1.2^{\pm 3.29}}$ | $\mathbf{10.0^{\pm 2.92}}$ | $\mathbf{0.9^{\pm 4.22}}$ |
| | ViSQOL | $\mathbf{2.9^{\pm 0.57}}$ | $\mathbf{3.2^{\pm 0.45}}$ | $2.3^{\pm 0.73}$ |
| **CodecSep + dnr-v2** | SI-SDR | $-0.2^{\pm 3.55}$ | $8.3^{\pm 2.60}$ | $-1.0^{\pm 4.20}$ |
| (**codes in : codes out, zero-shot**) | ViSQOL | $2.5^{\pm 0.52}$ | $3.0^{\pm 0.44}$ | $2.3^{\pm 0.67}$ |
| **CodecSep + dnr-v2** | SI-SDR | $-6.8^{\pm 2.77}$ | $2.0^{\pm 2.84}$ | $-6.8^{\pm 3.83}$ |
| (ablate **Masker**) | ViSQOL | $2.5^{\pm 0.58}$ | $2.6^{\pm 0.50}$ | $2.1^{\pm 0.74}$ |

overlapping sources) synthesized from FSD50K's hierarchical annotations Fonseca et al. (2022), combining fine-grained classes with parent categories (e.g., "dog barking, *Animal*, engine rumbling, *motor vehicle*"). To assess cross-domain generalization, we form three-source mixtures on AudioCaps Kim et al. (2019) (used for both training and testing in our open-domain setting) and construct test-only three-source mixtures from ESC-50 Piczak, Clotho-v2 Drossos et al. (2020), AudioSet-eval Gemmeke et al. (2017), and VGGSound Chen et al. (2020a). Dataset construction details, clip durations, split statistics, and segmentation rules are provided in the Appendix C.

**Evaluation.** We compare CodecSep against representative spectrogram-, waveform- and codec-domain baselines—TDANet Li et al. (2023); Pons et al. (2024), Sudo rm-rf Tzinis et al. (2022a), CodecFormer Yip et al. (2024b), SDCodec Bie et al. (2024), and the text-guided audio stream separators AudioSep Liu et al. (2024), BiModalSS Mahmud et al. (2024) and Sudo rm-rf + FiLM Tzinis et al. (2022b; 2023). We report objective signal fidelity via scale-invariant signal-to-distortion ratio (SI-SDR) Luo & Mesgarani (2019); Le Roux et al. (2019) and perceptual quality via ViSQOL Chinen et al. (2020), which measures spectro–temporal similarity between the estimate $\tilde{x}$ and reference $x$ and maps it to a 1–5 MOS-LQO scale. Following prior work (e.g., SDCodec Bie et al. (2024)), we use ViSQOL as a proxy MOS score for perceptual listening quality and complement it with a human MOS-LQS study comparing real-world outputs from CodecSep (trained on dnr-v2) and the publicly released AudioSep. To quantify efficiency, we report multiply–accumulate operations (MACs), inference time, memory footprint using `torchinfo`[1] under matched input durations (2 s) and batching (batch size 2) on a 24GB NVIDIA A30 GPU. Details on evaluation workflow for each benchmark are deferred to Appendix D.

**Training.** Unless otherwise stated, the DAC codec Kumar et al. (2023) and CLAP text encoder Wu et al. (2023) remain frozen; we train the FiLM-conditioned transformer masker and the lightweight query network end-to-end with an Adam optimizer Kingma & Ba (2017) and a plateau-based learning-rate schedule Mukherjee et al. (2019). We produce two variants of CodecSep, trained separately on dnr-v2 and on AudioCaps, to study distributional effects; we denote them with the suffixes "+dnr-v2" and "+AudioCaps". For fair comparison, the 3-stem versions of TDANet, Sudo rm-rf, and CodecFormer are re-trained from scratch on dnr-v2 using our setup. AudioSep is evaluated both as the publicly released checkpoint and when re-trained under matched protocols. We similarly re-train Sudo rm-rf+FiLM under the same matched settings. Pretrained checkpoints of BiModalSS and SDCodec are used as released by the authors. To reflect realistic deployments where signals traverse compression pipelines, inputs to non-codec baselines are passed through a full-band stereo-capable 48 kHz EnCodec during both training and inference. Full hyper-parameters, iteration schedules, batch configurations, and hardware details are deferred to the Appendix E.

---

[1] https://github.com/tyleryep/torchinfo

### 4.1 RESULTS AND DISCUSSIONS

Tables 1–6 present universal sound separation results, prompt–granularity analyses, architectural ablations, cross-dataset generalization, paraphrase robustness, and full inference complexity under matched training/evaluation protocols. Table 1 reports dnr-v2 test results for speech, music, and SFX: text-guided models use generic prompts for speech/music and ground-truth compositional captions for SFX; we also include a masker ablation to isolate the role of the Transformer masker. Table 2 studies SFX prompt granularity across three regimes—(i) fixed-stem, non-text baselines (TDANet, CodecFormer, SDCodec), (ii) generic 3-stem prompts ({music, speech, sfx}), and (iii) a universal setup with fine-grained, compositional SFX prompts—thereby aligning input conditions for fair comparison with fixed-head systems. Table 3 isolates architecture: decoder-style generation (CodecFormer) vs. an unguided 3-stem masker variant and its text-guided counterpart, all operating in the codec latent space. To assess out-of-domain generalization, Table 4 benchmarks on five additional open-domain corpora (ESC-50, Clotho-v2, AudioSet, VGGSound, AudioCaps) with mixtures of three randomly sampled sources and prompts drawn from captions (not tied to fixed labels).. Table 6 probes prompt paraphrasing robustness by replacing the generic speech/music prompts with unseen synonymic variants at inference (zero-shot paraphrase test). Finally, Table 5 compares end-to-end and architecture-only GMACs for spectrogram-domain separation versus codec-latent masking, including the practical code-stream case. All models are evaluated against the original (uncompressed) ground truth; our methods are highlighted in **bold**; and we report mean and standard deviation ($1\sigma$).

**Sound Separation Performance on dnr-v2 (cf. Table 1).** CodecSep+dnr-v2 outperforms both pretrained AudioSep (zero-shot) and retrained AudioSep+dnr-v2 across all categories, with sizable SI-SDR gains in speech ($10.0$ vs. $4.9/7.7\,\mathrm{dB}$), music ($1.2$ vs. $-2.5/-5.6\,\mathrm{dB}$), and SFX ($0.9$ vs. $-0.3/-4.7\,\mathrm{dB}$). In ViSQOL, CodecSep matches or exceeds AudioSep in speech and music while slightly trailing on SFX, likely reflecting differences in SFX prompt distributions (AudioSep's diverse training vs. CodecSep's compositional SFX prompts from dnr-v2). Importantly, our *bitstream-native* variant—CodecSep+dnr-v2 (codes in: codes out, zero-shot)—evaluates the CodecSep+dnr-v2 masker directly on reconstructed embeddings $E$ from code streams (§3.1, para. 6–7) *without* any fine-tuning: it incurs a modest drop relative to the continuous-latent path (about $1$–$2\,\mathrm{dB}$ SI-SDR across sources; small ViSQOL deltas for music/speech and parity on SFX), yet still surpasses AudioSep+dnr-v2 on SI-SDR for all sources (e.g., music: $-0.2$ vs. $-5.6\,\mathrm{dB}$; speech: $8.3$ vs. $7.7\,\mathrm{dB}$; SFX: $-1.0$ vs. $-4.7\,\mathrm{dB}$). Compared to pretrained AudioSep (zero-shot), the codes-in:codes-out variant improves SI-SDR on speech/music but lags on SFX SI-SDR and ViSQOL. These results indicate that a deployment-friendly, *no-finetuning* bitstream path is already competitive; as discussed in §3.1, para. 6 , light fine-tuning on $E$ or an embedding-consistency loss can close the residual gap. As part of our ablation, the lightweight CodecSep+dnr-v2 (ablate Masker) removes the transformer masker and applies FiLM directly to the encoder; it attains SI-SDR comparable to AudioSep+dnr-v2 with better perceptual speech quality, but overall separation quality drops due to FiLM perturbing the mixture latents held by the encoder. Beyond AudioSep, CodecSep also outperforms the USS-pretrained BiModalSS model and the retrained text-conditioned Sudo rm-rf + FiLM baseline. The heavier attention-based conditioning used in BiModalSS does not generalize well to the open-domain mixtures in dnr-v2, leading to degraded performance. CodecSep also outperforms the Sudo rm-rf + FiLM model trained under the same universal setting. This gap is likely due to two factors. First, continuous CLAP embeddings provide far richer semantic conditioning than the fixed one-hot or multi-hot vectors that Sudo rm-rf + FiLM was designed for, causing the model to struggle with open-domain prompts. Second, applying FiLM across all U-Conv blocks while using only a single Conv1d layer for audio encoding can destabilize internal representations, especially when conditioned with high-dimensional continuous embeddings. Since AudioSep also outperforms both by a large margin, we use AudioSep as the primary baseline in our further experiments.

**Effect of SFX Prompt Granularity during training.** We evaluate three regimes: (i) *fixed-stem* baselines without text guidance (TDANet, Sudo rm-rf, CodecFormer, SDCodec) (cf. Table 2), (ii) *generic 3-stem* prompts ("music/speech/sfx") (cf. Table 2), and (iii) *universal* prompting that retains generic cues for speech/music but uses fine-grained, compositional SFX descriptions (cf. Table 1). For (ii) and (iii), we train and evaluate separate versions of both CodecSep and AudioSep on each prompt setting. Across settings, CodecSep matches or exceeds fixed-stem baselines on SFX while maintaining comparable speech and music quality, and its USS variant outperforms Sudo rm-rf across all stems with modest margins (cf. Table 2). Under matched generic-prompt training/evaluation, CodecSep remains robust and surpasses spectrogram-domain AudioSep, indicating that effective

Table 2: Results: Impact of SFX Prompt Granularity on Universal Sound Separation (**dnr-v2-test**)

| Model | Metric (↑) | Music | Speech | Sfx |
|---|---|---|---|---|
| **3-Stem: Fixed stem baselines (no text-guidance)** | | | | |
| TDANet | SI-SDR | $1.8^{\pm 3.55}$ | $10.2^{\pm 2.91}$ | $1.4^{\pm 4.90}$ |
| | ViSQOL | $2.9^{\pm 0.58}$ | $3.1^{\pm 0.43}$ | $2.4^{\pm 0.72}$ |
| Sudo rm-rf | SI-SDR | $-0.9^{\pm 4.01}$ | $9.0^{\pm 2.60}$ | $0.6^{\pm 4.74}$ |
| | ViSQOL | $2.7^{\pm 0.59}$ | $2.9^{\pm 0.45}$ | $2.3^{\pm 0.72}$ |
| CodecFormer | SI-SDR | $-5.7^{\pm 3.44}$ | $2.3^{\pm 2.32}$ | $-6.5^{\pm 4.36}$ |
| | ViSQOL | $2.2^{\pm 0.47}$ | $2.5^{\pm 0.49}$ | $2.1^{\pm 0.67}$ |
| SDCodec | SI-SDR | $\mathbf{1.9^{\pm 3.68}}$ | $\mathbf{11.3^{\pm 2.98}}$ | $\mathbf{1.8^{\pm 4.08}}$ |
| | ViSQOL | $\mathbf{3.0^{\pm 0.56}}$ | $\mathbf{3.5^{\pm 0.40}}$ | $\mathbf{2.6^{\pm 0.73}}$ |
| **3-Stem: {"music","speech","sfx"} as generic prompt** | | | | |
| AudioSep | SI-SDR | $\mathbf{-2.5^{\pm 4.06}}$ | $4.9^{\pm 4.21}$ | $-6.7^{\pm 4.73}$ |
| (zero-shot) | ViSQOL | $\mathbf{2.9^{\pm 0.63}}$ | $3.1^{\pm 0.56}$ | $2.1^{\pm 0.68}$ |
| AudioSep + dnr-v2 | SI-SDR | $-6.2^{\pm 2.77}$ | $\mathbf{7.7^{\pm 3.11}}$ | $-2.1^{\pm 3.90}$ |
| | ViSQOL | $2.6^{\pm 0.57}$ | $2.5^{\pm 0.37}$ | $2.4^{\pm 0.74}$ |
| **CodecSep + dnr-v2** | SI-SDR | $-7.7^{\pm 2.84}$ | $4.6^{\pm 2.48}$ | $\mathbf{0.6^{\pm 4.15}}$ |
| | ViSQOL | $2.5^{\pm 0.55}$ | $2.7^{\pm 0.49}$ | $\mathbf{2.4^{\pm 0.70}}$ |

Table 3: Results: Architectural advantages in using CodecFormer decoder as masker (**dnr-v2-test**)

| Model | Metric (↑) | Music | Speech | Sfx |
|---|---|---|---|---|
| CodecFormer | SI-SDR | $-5.8^{\pm 3.44}$ | $2.3^{\pm 2.32}$ | $-6.5^{\pm 4.36}$ |
| | ViSQOL | $2.2^{\pm 0.47}$ | $2.5^{\pm 0.49}$ | $2.1^{\pm 0.67}$ |
| **CodecSep + dnr-v2** | SI-SDR | $1.2^{\pm 3.35}$ | $10.0^{\pm 2.91}$ | $0.9^{\pm 4.18}$ |
| **(unguided, 3-stem)** | ViSQOL | $2.8^{\pm 0.55}$ | $3.1^{\pm 0.45}$ | $\mathbf{2.5^{\pm 0.72}}$ |
| **CodecSep + dnr-v2** | SI-SDR | $\mathbf{1.2^{\pm 3.29}}$ | $\mathbf{10.0^{\pm 2.92}}$ | $\mathbf{0.9^{\pm 4.22}}$ |
| **(text-guided)** | ViSQOL | $\mathbf{2.9^{\pm 0.57}}$ | $\mathbf{3.2^{\pm 0.45}}$ | $2.3^{\pm 0.73}$ |

separation does not hinge on carefully crafted prompts (cf. Table 1– 2). Moreover, replacing the "sfx" label with detailed SFX prompts consistently sharpens SFX extraction and, importantly, improves speech and music stems as well—suggesting that richer SFX supervision enhances overall scene disentanglement (cf. Table 1). While these controlled studies cover multiple prompt granularities, we expect additional gains from training on larger, more diverse corpora with a spectrum of prompt specificities, which we leave for future work.

**Why use a Transformer masker instead of a decoder?(cf. Table 3).** We compare (i) CodecFormer (decoder-style source generation), (ii) CodecSep (unguided, 3-stem) which uses the CodecFormer Transformer as a masker over codec latents, and (iii) CodecSep (text-guided). The results on dnr-v2-test exhibit two clear trends. First, replacing decoder-style generation with masking consistently strengthens separation across music, speech, and SFX. This aligns with our design rationale: in the DAC latent domain, the masker modulates existing, semantically structured content instead of synthesizing new signals, which (a) reduces artifacts and cross-talk leakage, (b) preserves long-range periodicity/timbre and transient organization already encoded by the codec, and (c) stabilizes optimization compared to end-to-end generation. Second, adding text guidance yields a further uniform improvement. The masker formulation concentrates Transformer capacity on selection ("where/how much" to pass) rather than generation ("what" to produce).

**Further benchmarking on ESC-50, Clotho-v2, AudioSet, VGGSound, & AudioCaps (cf. Table 4).** Extending beyond dnr-v2, we evaluate both systems on five additional open-domain benchmarks spanning environmental sounds (ESC-50), audio captioning-style corpora (Clotho-v2, AudioCaps), weakly labeled web-scale audio (AudioSet), and visually grounded audio (VGGSound). Under matched training data and prompt protocols, CodecSep+dnr-v2 consistently outperforms AudioSep+dnr-v2 across all five datasets in both separation fidelity and perceptual quality.

**Full inference complexity (cf. Table 5).** Across all six dimensions of computational analysis—GMACs, inference time, parameter count, forward/backward memory, and full memory footprint—CodecSep demonstrates a substantial efficiency advantage over spectrogram-domain baselines, especially in the deployment-realistic code-stream mode. In terms of hardware-agnostic compute (cf. Table 5a ), CodecSep requires only 1.35 GMACs when operating on codec bitstreams, compared to 73.6 GMACs for AudioSep and 56.5 GMACs for Sudo rm-rf, yielding roughly $54\times$ and $40\times$ reductions, respectively (and $25\times/12\times$ under the architecture-only comparison). These savings translate directly to latency (cf. Table 5b): CodecSep achieves 0.19 s inference in code-stream mode—$8\times$ faster than AudioSep and approximately $10\times$ faster than Sudo rm-rf—despite having a moderately larger parameter count. Parameter efficiency mirrors this trend (cf. Table 5d): when fed code streams, CodecSep uses only 16.3M parameters, markedly smaller than AudioSep (112.8M) and Sudo rm-rf (78.8M), reflecting its lightweight, masker-only design. Memory usage shows the most pronounced differences. For forward/backward activations (cf. Table 5e), CodecSep requires just 28 MB in code-stream mode, compared to 1.58 GB for AudioSep and 2.84 GB for Sudo rm-rf—over $50\times$ to $100\times$ reductions. The overall footprint follows suit (cf. Table 5f): full memory usage for CodecSep is only 76.5 MB, versus 2.03 GB (AudioSep) and 3.15 GB (Sudo rm-rf), yielding roughly $27\times$ and $41\times$ reductions, respectively. waveforms (audio-stream input), where CodecSep must additionally run the codec encoder/decoder, it still consumes less memory than Sudo rm-rf (1.14 GB vs. 2.05 GB), demonstrating that codec compute, while non-trivial, is not the dominant bottleneck. These results collectively indicate that code-stream processing—already standard in real systems—is where CodecSep is maximally efficient, achieving extreme reductions in compute, memory, and latency. This makes CodecSep uniquely suited for scalable deployment on edge–server pipelines, where audio is almost always exchanged as codec bitstreams rather than raw waveforms.

**Additional experiments (cf. Appendix F–J) .** For readability, five extended studies are deferred to the Appendix, which also details data construction, prompt protocols (generic vs. universal and paraphrased variants), training/evaluation splits, and metrics. *(i) Robustness to prompt paraphrasing (cf. Appendix F):* We test lexical robustness by replacing the generic *speech* and *music* prompts with three unseen paraphrases each (e.g., "spoken voice," "people talking"; "instrumental music," "band playing"). This zero-shot paraphrase test requires models trained on generic labels to handle synonymic descriptors at inference. While both CodecSep+dnr-v2 and AudioSep+dnr-v2 degrade under paraphrasing, CodecSep remains consistently stronger (Table 6). This experiment focuses on lexical variants; paraphrases involving temporal structure (e.g., "applause follows a song") are left for future work. *(ii) Generalization across open-domain datasets (cf. Appendix G):* On AudioCaps (derived from AudioSet), the spectrogram baseline AudioSep benefits from distributional alignment and attains the strongest absolute scores, yet CodecSep+dnr-v2 generalizes competitively—surpassing AudioSep+dnr-v2 in SI-SDR at comparable ViSQOL (cf. Table 7). When both are retrained on AudioCaps, CodecSep+AudioCaps again outperforms AudioSep+AudioCaps in separation quality. The same trend holds on the more challenging dnr-v2 test set, where mixtures often contain speech, music, and multiple overlapping SFX; both models drop in absolute performance, but CodecSep retains its advantage in SI-SDR. *(iii) Relative–gain summaries (cf. Appendix H):* We report percent improvements of CodecSep over AudioSep under matched training data and prompt settings; CodecSep shows large SI-SDR gains on dnr-v2 with positive (though smaller) gains under paraphrased prompts, and consistent improvements across ESC-50, Clotho-v2, AudioSet, VGGSound, and AudioCaps, with modest ViSQOL deltas (cf. Table 8). *(iv) Higher bandwidth (cf. Appendix I):* Replacing the frozen 16 kHz DAC with a 48 kHz, stereo-capable EnCodec leaves the masking architecture unchanged but increases latent-frame count and high-frequency detail, making separation harder and reducing SI-SDR/ViSQOL (Table 9). At higher $F_s$, gains depend mainly on backbone capacity and data rather than architectural modifications. Evaluations of 24 kHz and 44.1 kHz DAC variants show the same pattern: with fixed masker capacity, performance declines monotonically as sampling rate increases, reflecting the difficulty of wide-bandwidth separation, while DAC consistently outperforms EnCodec at matched bandwidths. *(v) Reconstruction study:* We probe *source leakage* with a *single-source reconstruction* diagnostic on dnr-v2 : each input contains one target source and models are either prompted with the matching caption (text-guided) or routed to the corresponding fixed head (non–text-guided). We also report *mixture reconstruction* by summing predicted source stems for a mixture and comparing to the original. This is a leakage/consistency check—not a primary separation metric; full setup/results are in Appendix J (cf. Table 10).

**Subjective evaluation (MOS–LQS).** We ran a human evaluation test with $n=20$ participants on 20 dnr-v2 3-stem test mixtures, comparing paired outputs from CodecSep+dnr-v2 and the official AudioSep model using fixed *speech/music* prompts and per-clip *sfx* prompts. Raters scored each

Table 4: Results: Benchmarking on **ESC-50**, **Clotho-v2**, **AudioSet**, **VGGSound**, **AudioCaps**

| Model | Metric ($\uparrow$) | ESC-50 | Clotho-v2 | AudioSet | VGGSound | AudioCaps |
|---|---|---|---|---|---|---|
| AudioSep | SI-SDR | $-7.8^{\pm 14.46}$ | $-8.6^{\pm 17.0}$ | $-7.6^{\pm 11.42}$ | $-7.0^{\pm 12.65}$ | $-6.4^{\pm 11.48}$ |
| + dnr-v2 | ViSQOL | $2.3^{\pm 1.12}$ | $2.1^{\pm 1.08}$ | $2.1^{\pm 1.00}$ | $2.2 \pm 1.10$ | $\mathbf{2.3^{\pm 1.08}}$ |
| **CodecSep** | SI-SDR | $\mathbf{-5.9^{\pm 11.55}}$ | $\mathbf{-6.0^{\pm 11.10}}$ | $\mathbf{-6.4^{\pm 10.53}}$ | $\mathbf{-6.1^{\pm 12.12}}$ | $\mathbf{-6.1^{\pm 11.62}}$ |
| **+ dnr-v2** | ViSQOL | $\mathbf{2.3^{\pm 1.13}}$ | $\mathbf{2.3^{\pm 1.09}}$ | $\mathbf{2.2^{\pm 1.0}}$ | $\mathbf{2.3^{\pm 1.11}}$ | $2.2^{\pm 1.16}$ |

Table 5: Full Inference Compute Benchmarking Across Six Settings on 24GB NVIDIA A-30 GPU

(a) Inference GMACs ($\downarrow$)

| Model | Audio Stream I/O | Code Stream I/O | Architecture-only |
|---|---|---|---|
| AudioSep | 33.5 | 73.6 | 33.5 |
| Sudo rm-rf | **16.44** | 56.54 | 16.44 |
| **CodecSep** | 41.45 | **1.35** | **1.35** |

*Codec GMACs:* Enc=12.28, Dec=27.82

(b) Inference Time (s) ($\downarrow$)

| Model | Audio Stream I/O | Code Stream I/O | Architecture-only |
|---|---|---|---|
| AudioSep | **0.33** | 1.63 | 0.33 |
| Sudo rm-rf | 0.51 | 1.81 | 0.51 |
| **CodecSep** | 1.49 | **0.19** | **0.19** |

*Codec Inference Times (s):* Enc=1.16, Dec=0.14

(c) Parameter count (in million) ($\downarrow$)

| Model | Audio Stream I/O | Code Stream I/O | Architecture-only |
|---|---|---|---|
| AudioSep | 39 | 112.8 | 39 |
| Sudo rm-rf | **5** | 78.8 | **5** |
| **CodecSep** | 90.1 | **16.3** | 16.3 |

*Codec Parameter Count (in million):* Enc=21.5, Dec=52.3

(d) Parameter-only Memory Footprint (MB) ($\downarrow$)

| Model | Audio Stream I/O | Code Stream I/O | Architecture-only |
|---|---|---|---|
| AudioSep | 156.13 | 447.62 | 156.13 |
| Sudo rm-rf | **20.07** | 311.25 | **20.07** |
| **CodecSep** | 339.89 | **48.4** | 48.4 |

*Codec Parameter-only Memory Footprint (MB):* Enc=85.1, Dec=206.39

(e) Forward/Backward Pass Memory Footprint (MB) ($\downarrow$)

| Model | Audio Stream I/O | Code Stream I/O | Architecture-only |
|---|---|---|---|
| AudioSep | **804.3** | 1580.6 | 804.3 |
| Sudo rm-rf | 2032.13 | 2836.13 | 2032.13 |
| **CodecSep** | 804.4 | **28.06** | **28.06** |

*Codec Forward/Backward Pass Memory Footprint (MB):* Enc=310.48, Dec=465.82

(f) Full Memory Footprint (MB) ($\downarrow$)

| Model | Audio Stream I/O | Code Stream I/O | Architecture-only |
|---|---|---|---|
| AudioSep | **960.43** | 2028.22 | 960.43 |
| Sudo rm-rf | 2052.2 | 3147.68 | 2052.2 |
| **CodecSep** | 1144.25 | **76.46** | **76.46** |

*Codec Full Memory Footprint (MB):* Enc=395.58., Dec=672.21

stem independently in randomized order on the MOS–LQS scale (1=bad, 5=excellent); we report mean $\pm 1\sigma$. Overall, CodecSep scored $3.34^{\pm 1.00}$ vs. AudioSep $2.61^{\pm 1.04}$. By source, CodecSep achieved $3.17^{\pm 1.01}$ (music), $3.37^{\pm 0.97}$ (sfx), and $3.49^{\pm 1.00}$ (speech), while AudioSep obtained $2.49^{\pm 0.95}$, $2.84^{\pm 1.16}$, and $2.50^{\pm 1.02}$, respectively. These outcomes align with objective trends (SI-SDR/ViSQOL) and indicate consistent perceptual gains for CodecSep. Paired model outputs and reference stems are included in the supplementary materials for side-by-side listening.

**Extension to multi-modal prompting.** Because conditioning enters only via a fixed-dimensional query embedding $e_\tau$ that drives FiLM in the masker, the architecture is agnostic to the prompt modality. Concretely, one can replace the text encoder with (i) an *audio* encoder to accept audio prompts $\left(e_\tau^{\text{aud}}\right)$, (ii) an *image/vision–language* encoder (e.g., CLIP) to accept image prompts $\left(e_\tau^{\text{vis}}\right)$, or (iii) a lightweight fusion (e.g., gated additive or attention pooling) of $\left(e_\tau^{\text{text}}, e_\tau^{\text{aud}}, e_\tau^{\text{vis}}\right)$ to support mixed prompts—all without modifying the masker or the codec.

## 5 CONCLUSION

CodecSep advances text-guided universal sound separation by operating directly in NAC latents with a FiLM-conditioned Transformer masker (not a decoder), outperforming audio stream separators like AudioSep across dnr-v2 and five open-domain datasets (AudioCaps, ESC–50, Clotho–v2, AudioSet, VGGSound) under matched training/prompt protocols. In code-stream deployments, it needs just 1.35 GMACs end-to-end—$\sim 54\times$ less compute ($25\times$ architecture-only) than spectrogram domain separators like AudioSep—while remaining fully bitstream-compatible. Ablations indicate that DAC latents are sufficiently structured that masking over them yields stronger separation than decoding/generating sources from the latents; a MOS–LQS study corroborates perceptual gains. We also demonstrate a codes in: codes out route that operates on reconstructed embeddings without fine-tuning, highlighting deployment readiness. Looking ahead,we will broaden prompt coverage with temporal/relational and referring-expression cues, extend separation to higher bandwidths and stereo/spatial audio (e.g., 48 kHz stereo Encodec; HO–DirAC Hold et al. (2024), SpatialCodec Xu et al. (2024)), and support audio/image or mixed prompts. The predicted masks and FiLM responses also reveal how CodecSep selectively activates latent components, offering a natural avenue for future interpretability work via mask visualization and reconstruction-consistency checks. We discuss the limitations of our work in Appendix K. We provide supplementary code to facilitate reproducibility.

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

## A  FAILURE MODES OF PIT AND MIXIT FOR UNIVERSAL SOUND SEPARATION

Permutation-Invariant Training (PIT) and Mixture-Invariant Training (MixIT) have historically been effective for closed-domain separation tasks where the number of underlying sources is known, fixed, or varies within a narrow, well-defined range. However, their underlying assumptions lead to structural limitations when applied to open-domain universal source separation (USS), where mixtures may contain an arbitrary and potentially large number of heterogeneous sound events. In this section, we summarize the key failure modes observed when training with PIT/MixIT to extend fixed-stem models to open-domain mixtures.

A core limitation of PIT/MixIT is the requirement to specify a maximum number of output sources, denoted by $N$. During training, the model produces exactly $N$ outputs for every mixture, and the PIT or MixIT objective establishes a correspondence between these outputs and the underlying reference sources (or intermediate MixIT partitions). This design is brittle in scenarios where the true number of sources varies widely. When the mixture contains more than $N$ sources—a frequent occurrence in open-domain audio—the model has no mechanism to create additional outputs. Instead, it suffers source collision and collapses multiple sources into a single output stem, resulting in unavoidable leakage, loss of fine structure, and a sharp degradation in separation quality. The post-hoc identification step cannot recover the missing sources, because the model never produced separate representations for them in the first place; those sources simply do not exist within the model's output space.

Conversely, when the mixture contains fewer than $N$ sources, the model is still obligated to return $N$ outputs. This mismatch introduces new problems: several outputs correspond to no actual source and become "inactive" stems, while others may capture residual background energy or hallucinated content. These false positives degrade metrics such as SI-SDR and create ambiguity during evaluation because the model does not encode which stems are meant to be meaningful. Such outputs also make deployment difficult, as downstream systems must decide which stems to trust and which to ignore.

The reliance on a fixed maximum number of sources $N$ also places a heavy burden on both training stability and computational cost. As $N$ increases, the permutation space in PIT expands combinatorially, and MixIT assignments become increasingly complex, making training slow, unstable, and in many cases prone to divergence. In open-domain datasets such as dnr-v2, mixtures may contain eight or more concurrent sound events, forcing PIT/MixIT baselines to adopt impractically large values of $N$ to avoid source collisions. In practice, such configurations are computationally prohibitive and empirically unreliable.

These limitations collectively illustrate why PIT and MixIT, despite their historical success in speech separation and other closed-set tasks, are poorly suited for open-domain universal separation. Their fixed-output architecture is fundamentally mismatched to real-world mixtures that contain highly variable and unpredictable numbers of sources. In contrast, CodecSep bypasses this bottleneck entirely through free-form text-guided inference: the model extracts only the requested source category, emits no unused stems, and scales naturally to mixtures with arbitrary levels of overlap. This flexibility enables CodecSep to support both closed-set and open-domain use cases, while also providing a foundation for future extensions to fine-grained extraction of individual speaker stems, instrument stems, or sound-effect stems.

# B    DESIGN RATIONALE: FiLM-CONDITIONED MASKING IN NAC LATENT SPACE

**Problem setup and pipeline contrast.**    Let $x(t) \in \mathbb{R}$ be a mono mixture with sources $\{y_s(t) | s \in \mathcal{S}\}$, where $x(t)$ is expressed as $x(t) = \sum_{s \in \mathcal{S}} y_s(t)$.

**Spectrogram-domain (AudioSep):**

$$x \xrightarrow{\text{STFT}} X \in \mathbb{C}^{F \times T_{spec}} \xrightarrow{g(X, e_\tau)} \tilde{Y}_s = |\hat{M}_s| \odot |X| \exp(\angle X + \angle \hat{M}_s) \xrightarrow{\text{ISTFT}} \tilde{y}_s(t),$$

where $g(\cdot, e_\tau)$ denotes a FiLM-conditioned, complex-domain spectrogram separator that predicts a magnitude mask $|\hat{M}_s| \in [0,1]^{F \times T_{spec}}$ and a phase residual $\angle \hat{M}_s$, conditioned jointly on the mixture spectrogram $X$ (obtained by the STFT of $x$) and the text embedding $e_\tau$. The predicted magnitude mask and phase residual are then applied element-wise to form the source spectrogram $\tilde{Y}_s$ which is subsequently transformed back to the time domain via ISTFT to obtain $\tilde{y}_s(t)$.

**CodecSep (NAC latent-domain):**

$$x \xrightarrow{Enc(\cdot)} Z \in \mathbb{R}^{d \times T} \xrightarrow{Mask(Z, e_\tau)} \tilde{Z}_s = M_s \odot Z \xrightarrow{Dec(\cdot)} \tilde{y}_s(t),$$

with a frozen DAC backbone $Enc(\cdot), Dec(\cdot)$, and a FiLM-conditioned transformer masker $Mask(\cdot, e_\tau)$ that estimates a soft mask $M_s \in [0,1]^{d \times T}$ conditioned on both the codec latents $Z = Enc(x)$ and the text embedding $e_\tau$. The predicted mask $M_s$ is applied element-wise to $Z$ to produce source-specific latents $\tilde{Z}_s = M_s \odot Z$, which are subsequently decoded by $Dec(\cdot)$ to obtain the separated waveform $\tilde{y}_s(t)$.

**Dimensionality reduction and compression.**    Operating on NAC latents $Z$ slashes dimensionality while preserving perceptual factors. For 1 s audio at 32 kHz, complex STFT with $N=1024$ and hop size $M=320$ samples has $T_{\text{spec}} \approx 100$ frames and $F = 2 \times 1024$ (Re+Im) scalars per frame, so $F \cdot T_{spec} \approx 204{,}800$. A 16 kHz DAC with width $d=64$ and the same $M$ yields $T \approx 50$ and $d \cdot T = 64 \times 50 = 3{,}200$ ($\sim 64\times$ smaller), shrinking $Q/K/V$ and MLP sizes and easing self-attention. Similarly, for 32 kHz NACs like EnCodec, $T \approx 50$ with $d=128$, so attention/MLPs still operate on $\sim 32\times$ smaller latents than complex STFTs.

## B.1    WHY NAC LATENTS (VS. SPECTROGRAMS) AND HOW THE CODEC PRIOR ENABLES SEPARATION?

**STFT vs. NAC encoding.**    The STFT is a *linear* projection from $x(t)$ to $X \in \mathbb{C}^{F \times T}$ and does not explicitly preserve or organize the intrinsic semantic structure of audio. Consequently, spectrogram-based separators (e.g., AudioSep) require an additional learned encoder–decoder (typically CNN/ResUNet) to first compress $X$ into high-dimensional latent features and then decode spectrogram masks that are discriminatively trained for separation. This couples *semantic abstraction* and *separation* into one network, increasing parameters and MACs and forcing the model to learn structure "from scratch" in a redundant, noisy representation.

**NAC encoder as a semantic prior.** Neural audio codecs (DAC) are trained with perceptual, adversarial, and codebook objectives that encourage the encoder $Enc(\cdot)$ to map $x$ into compact, structured latents:

$$Enc(.) : x \mapsto Z \in \mathbb{R}^{d \times T}.$$

These latents lie on a *discriminative compressed manifold* $\mathcal{M}_{\text{latent}}$ in which semantically meaningful factors (e.g., pitch, timbre, transients) are disentangled and aligned for downstream use. In our system, we operate directly on the *continuous* latents $Z$ (from $Enc$), use a FiLM-conditioned transformer masker to predict a separation mask $M_k$ from a textual query $\tau_k$, and form the estimate

$$\tilde{Z}_k = M_k \odot Z, \qquad \tilde{y}_k = Dec(\tilde{Z}_k),$$

thereby leveraging the codec's structured manifold for masking instead of learning a new representation.

**Separation mapping in latent vs. spectrogram space.** In CodecSep, the separator learns

$$Mask(.,.) : Z \in \mathcal{M}_{\text{latent}} \to \hat{Z}_k,$$

which is easier to optimize because the input is denoised, compressed, and semantically organized by the codec. In contrast, spectrogram models must learn

$$g(.,.):\ X \in \mathbb{C}^{F \times T}\ \rightarrow\ \hat{Y}_k,$$

over a noisier, higher-dimensional space *without* semantic compression.

**Hierarchical (RVQ) structure that benefits separation.** The codec applies Residual Vector Quantization (RVQ) to $Z$, producing discrete codes $A = [a_t \in [K]^{N_q}]_{t=1}^{T}$ with $K = 1024$ and $N_q$ quantizers. Codebook lookup yields embeddings

$$e_t\ =\ \sum_{i=1}^{N_q} \mathrm{lookup}(a_t^{(i)}), \qquad E = [e_t | t \in [T]] \approx Z,$$

and $Dec(E)$ reconstructs the waveform. The RVQ cascade induces a natural coarse-to-fine hierarchy: the first quantizer captures coarse structure (e.g., low-frequency content, speaker/instrument timbre, global acoustic traits), while later quantizers refine residual details (e.g., high-frequency components, onsets/transients, background textures). This hierarchy mirrors the discriminative cues needed for separation and is directly exploitable by a mask-based transformer.

**Loss-induced organization of the latent space.** The DAC objective shapes $Z$ using complementary terms:

- *Multi-scale spectral loss* $\mathcal{L}_{\mathrm{mel}}$ to preserve perceptually relevant frequency content at multiple time scales;
- *Time-domain reconstruction loss* $\mathcal{L}_{\mathrm{feat}} = \|x(t) - \tilde{y}(t)\|_1$ for fidelity and stability;
- *Multi-resolution adversarial loss* $\mathcal{L}_{\mathrm{adv}}$ with (i) multi-period waveform discriminators (pitch/periodicity) and (ii) multi-band STFT discriminators (fine spectral detail), plus a *feature-matching* term $\mathcal{L}_{\mathrm{feat}}^{G}$;
- *Codebook loss* $\mathcal{L}_{\mathrm{code}}$ to ensure compact, diverse, well-utilized codes and reinforce RVQ's coarse-to-fine disentanglement.
- *Quantizer dropout (RVQ stage dropout)*: randomly disabling a subset of RVQ stages during training to discourage over-reliance on late codebooks and encourage smoother coarse-to-fine residual allocation, yielding more bitrate-robust and well-structured RVQ representations.

The overall loss is

$$\mathcal{L}_{\mathrm{DAC}}\ =\ \lambda_{\mathrm{mel}}\mathcal{L}_{\mathrm{mel}}\ +\ \lambda_{\mathrm{feat}}\mathcal{L}_{\mathrm{feat}}\ +\ \lambda_{\mathrm{adv}}\mathcal{L}_{\mathrm{adv}}\ +\ \lambda_{\mathrm{code}}\mathcal{L}_{\mathrm{code}},$$

yielding latents that are (i) *denoised* (robust to low-level artifacts), (ii) *semantic* (preserve pitch, timbre, temporal structure), (iii) *disentangled* (coarse-to-fine RVQ), and (iv) *efficient* (bitrate-aware constraints).

**Separation benefits from the codec prior (and contrast to spectrogram baselines).** Because $Z$ is already semantically organized, the FiLM-conditioned masker operates on a representation that encodes the right factors for selection, not generation. This (i) reduces compute and memory (the separator acts on compact $Z$ instead of $X$), (ii) accelerates convergence and improves stability (masking over a clean manifold), and (iii) improves robustness to prompt variation (text FiLM modulates semantically aligned channels). In contrast, spectrogram systems must learn a task-specific latent from $X$ and perform separation *jointly*, which increases parameter count and MACs, slows convergence, and can lead to overfitting in the absence of the codec's inductive bias.

## B.2 TRANSFORMER-BASED MASKER VS. DECODER-STYLE GENERATION (SPECIFIC DESIGN CHOICE).

In CodecSep, we replace decoder-style source generation (as in CodecFormer, which directly predicts the target waveform/spectrogram) with a *Transformer-based, FiLM-conditioned masker* that outputs a soft mask over the codec latents. Concretely,

$$M_s\ =\ Mask(Z, e_\tau) \in [0,1]^{d \times T}, \qquad \tilde{Z}_s\ =\ M_s \odot Z, \qquad \tilde{y}_s(t)\ =\ Dec(\tilde{Z}_s).$$

This choice has the following concrete advantages (all in the NAC latent domain):

- **Efficient and stable training.** Predicting $M_s$ to *modulate* existing latent content is a simpler, more constrained learning problem than end-to-end *generation* of $\tilde{Z}_s$ or $\tilde{y}_s(t)$ with a decoder head. Working in $Z$ avoids the instability commonly observed in direct waveform/spectrogram prediction, leading to faster convergence and lower training variance under the same optimization settings.

- **Direct leverage of codec-disentangled latents.** DAC/RVQ pretraining organizes $Z$ into a coarse-to-fine, semantically disentangled manifold (periodicity, timbre, transients). The masker exploits this structure by *gating along* these axes to isolate sources, rather than using $Z$ merely as input to a decoder that *generates* new latents/waveforms. This turns separation into selection on a well-structured space, improving identifiability while reducing parameters and data demands.

- **Minimal distortion through modulation (no hallucination).** The separator does not synthesize new content; it rescales and selects what is already encoded in $Z$. Forming $\tilde{Z}_s = M_s \odot Z$ preserves the mixture's latent structure and reduces artifacts relative to encoder–decoder separation pipelines that generate source signals from scratch, thereby limiting hallucinations and leakage.

- **Preservation of long-term temporal/spectral structure.** The NAC encoder has already organized periodicity, timbre, and transient structure in $Z$. Masking retains this organization across long contexts, whereas fully convolutional decoders trained to *generate* sources often exhibit long-term inconsistencies (e.g., drift over time or loss of periodic cues) when reconstructing from scratch.

- **Efficient use of Transformer capacity.** Instead of *synthesizing* source signals, the masker learns to *gate* semantically organized channels in $Z$, which is a substantially lighter optimization problem than decoder-style generation. FiLM conditioning steers the transformer to decide *where/how much* information to pass—not *what* to generate—so parameters and compute in the $Q/K/V$ and MLP projections are concentrated on selection and attenuation.

## B.3 WHY WE INTEGRATED FiLM CONDITIONING INTO NAC-BASED SEPARATION

We deliberately integrate *Feature-wise Linear Modulation (FiLM)* Perez et al. (2018) *inside* the transformer masker to inject text semantics while preserving the codec manifold and keeping $Enc(\cdot)/Dec(\cdot)$ frozen.

**Targeted placement (masker, mid-layers).** Given a CLAP text embedding $e_\tau$, a lightweight query network $query(\cdot)$ produces per-layer affine parameters

$$(\gamma^l, \beta^l)_{l=1}^L = query(e_\tau), \quad \gamma^l, \beta^l \in \mathbb{R}^d,$$

which modulate intermediate activations $H^l \in \mathbb{R}^{d \times T}$ for $l \in \{2, \dots, L-1\}$:

$$\tilde{H}_s^l = \text{FiLM}(H^l; \gamma^l, \beta^l) = \gamma^l \odot H^l + \beta^l.$$

Placing FiLM in the *masker* (not in $Enc$ or $Dec$) keeps the codec latent distribution intact and confines conditioning to the *selection* step.

**Lightweight computation (overhead and parameterization).** FiLM adds only small per-layer vectors $(\gamma^l, \beta^l)$ and a compact $query(\cdot)$ MLP; it does *not* increase sequence length, attention heads, or the quadratic attention cost. The extra FLOPs/params are negligible relative to multi-head attention and MLP blocks, aligning with CodecSep's efficiency goals.

**Non-iterative inference (single forward pass).** FiLM applies in a single pass through the masker. Unlike iterative conditioning mechanisms (e.g., flow-matching-based sampling), there are *no* sampling steps, thus preserving low latency for edge and hybrid deployments.

**Manifold preservation and stability.** Because FiLM scales/shifts existing channels of $H^l$ rather than rewriting $Z$ or generating $\tilde{Z}_s$ from scratch, the NAC manifold structure (periodicity, timbre, transients) is preserved. Empirically this reduces training variance and mitigates long-horizon inconsistencies common with generator-style heads.

### B.4 WHY REPORT RESULTS ON $Z$ (CONTINUOUS LATENTS) AND HOW TO EXTEND TO CODE STREAMS $A \rightarrow E$

**Why we evaluate on continuous latents $Z$.** We choose to perform—and therefore report—separation on the *continuous* DAC encoder latents $Z = Enc(x) \in \mathbb{R}^{d \times T}$ for the following concrete reasons:

1. **End-to-end gradient flow for separation.** Our training updates only the masker and query networks while keeping the codec frozen. Using $Z$ allows straightforward backpropagation through $Mask(\cdot, e_\tau)$ and $Dec(\cdot)$ without dealing with discrete indices or straight-through estimators; gradients are well-behaved and convergence is consistently stable.

2. **Representational fidelity and disentanglement.** During codec pretraining, the RVQ cascade *regularizes* $Z$ so that pitch, timbre, onsets/transients, and background textures are cleanly and hierarchically organized. $Z$ therefore provides a richer, more disentangled signal for FiLM-conditioned masking than hard code indices, which are subject to quantization coarsening.

3. **Training stability and variance.** In our setting (frozen codec), operating on $Z$ avoids variability from codebook utilization dynamics (e.g., late-stage RVQ sensitivity, bitrate truncation). Empirically, this reduces run-to-run variance and removes the need for specialized regularizers when training the separator.

**How to extend the same model to discrete code streams.** For deployment scenarios where the input is a compressed bitstream, we operate on the *reconstructed embeddings* $E$ obtained from the codes $A$ via codebook lookup, and train the masker on $E$ instead of $Z$:

$$A = [a_t \in [1024]^{N_q} \mid t \in [T]], \qquad e_t = \sum_{i=1}^{N_q} \text{lookup}(a_t^{(i)}), \tag{3}$$

$$E = [e_t]_{t=1}^T \approx Z, \qquad \tilde{E}_s = M_s \odot E, \qquad \tilde{y}_s(t) = Dec(\tilde{E}_s) \tag{4}$$

**Compressed bitstream path (codes-in, codes-out).** Given $E \approx Z$ and the element-wise masking operation, the estimated source embeddings satisfy

$$\tilde{E}_s = M_s \odot E \approx M_s \odot Z = Z_s$$

When a uniform codec pathway (or codes-out interface) is required, we re-quantize the masked embeddings and optionally decode via the codec:

$$\hat{A}_s = Quant(\tilde{E}_s), \qquad \hat{E}_s = \text{lookup}(\hat{A}_s), \qquad \tilde{y}_s(t) = Dec(\hat{E}_s)$$

In deployments that only need to return a bitstream, the server can emit $\hat{A}_s$ directly and defer decoding to the client; otherwise, decoding $Dec(\tilde{E}_s)$ or $Dec(\hat{E}_s)$ yields the waveform on-device or server-side, respectively.

*Embedding alignment (CodecFormer-EL Yip et al. (2024a)).* To tighten $\tilde{E}_s \approx Z_s$, we can optimize an embedding-level consistency loss as in CodecFormer-EL Yip et al. (2024a)instead of our SI-SDR objective:

$$\mathcal{L}_{\text{emb}} = \sum_s \|\tilde{E}_s - Z_s\|_1$$

**Why this works.** By design of the codec, $E$ *approximates* $Z$ at the operating bitrate; the decoder already reconstructs $\hat{x}(t) = Dec(E)$ with high fidelity. Since our separator is a *masker* (selective modulation) rather than a generator, replacing $Z$ with $E$ preserves the semantics needed for separation while enabling direct operation on bitstreams. In practice, the extension amounts to training (or fine-tuning) the same FiLM-conditioned transformer on $E$ with unchanged objectives.

**Summary of scope.** We present results on $Z$ to (i) isolate separator performance without discrete-index training complications, (ii) leverage continuous gradient flow and stable optimization, and (iii) align FiLM conditioning with a smooth latent manifold. The deployment-ready path to discrete code streams is immediate: switch the masker input to $E$ via lookup from $A$ and train/fine-tune accordingly, with optional distribution alignment and bitrate-robustness augmentation.

### B.5 Deployment advantages of CodecSep over spectrogram-domain separators (AudioSep)

**Central motivation.** In realistic deployments, edge devices already run a neural audio codec; they transmit *code streams* rather than raw waveforms. CodecSep treats the codec backbone as part of the separation stack and operates *directly* on codec representations, thereby eliminating redundant *decode $\rightarrow$ separate $\rightarrow$ re-encode* cycles required by spectrogram-domain systems.

**Pipeline comparison (server-side separation).**   *Traditional (spectrogram) pipeline:*

$$\text{Edge: } \underbrace{Enc(.)}_{\text{codec}} \Rightarrow \text{Code} \Rightarrow \underbrace{\text{Server: } Dec(.) + g(\cdot, e_\tau) + Enc(.)}_{\text{decode + separate + re-encode}} \Rightarrow \text{Code} \Rightarrow \text{Edge: } Dec(.)$$

*CodecSep pipeline:*

$$\text{Edge: } \underbrace{Enc(.)}_{\text{codec}} \Rightarrow \text{Code} \Rightarrow \underbrace{\text{Server: } Mask(\cdot, e_\tau)}_{\text{mask on } E \approx Z} \Rightarrow \text{Code} \Rightarrow \text{Edge: } Dec(.)$$

CodecSep performs FiLM-modulated masking in the codec latent domain and returns separated code streams for edge-side decoding; spectrogram systems must decode to waveform (or magnitude/phase), separate in $X$, and re-encode.

**Complexity accounting.**   Let $C_{\text{enc}}$ and $C_{\text{dec}}$ be codec encode/decode costs, $C_{\text{spec}}$ the spectrogram separator cost, and $C_{\text{mask}}$ the CodecSep masker cost.

    **Code-stream input (typical):**   AudioSep: $C_{\text{dec}} + C_{\text{spec}} + C_{\text{enc}}$,        CodecSep: $C_{\text{mask}}$.

    **Audio-stream input (edge-only):**   AudioSep: $C_{\text{spec}}$,        CodecSep: $C_{\text{enc}} + C_{\text{mask}} + C_{\text{dec}}$.

In the common (code-stream) case, CodecSep removes both decode and re-encode on the server. Moreover, *within* the separator, CodecSep operates on $Z/E$ with $|Z| \ll |X|$ (cf. Sec. B), reducing activation memory and bandwidth throughout attention and MLP blocks.

**Interface compatibility with codec bitstreams.**   When only quantized codes $A = [a_t | t \in [T]]$ are available, we reconstruct embeddings by codebook lookup $E = [e_t | t \in [T]]$ with $e_t = \sum_{i=1}^{N_q} \text{lookup}(a_t^{(i)}) \approx Z$ and apply the same masker:

$$A \Rightarrow E \approx Z \xrightarrow{Mask(\cdot, e_\tau)} \tilde{E}_s \Rightarrow \text{Code stream out.}$$

No architectural change is required; separation remains in the codec latent domain and stays fully compatible with streaming/edge ecosystems.

**Why spectrogram-domain systems incur extra overhead.**   Spectrogram separators (e.g., AudioSep) are defined on $X = \text{STFT}(x)$. Given code-stream inputs, they must first run $Dec(.)$ to obtain a waveform, compute $X$, perform separation, and then run $Enc(.)$ to return codes. This decode + separate + re-encode loop adds latency, memory traffic, and energy cost on the server path and scales poorly with concurrent streams.

**Operational advantages of CodecSep.**

- **Eliminates redundant codec cycles in server workflows:** With code streams, we reconstruct embeddings $E \approx Z$ by codebook lookup (Sec. B.4), avoiding server-side decode/re-encode; only the edge decodes the final stems.
- **Smaller working representation during separation:** masking in $Z/E$ (e.g., $d{=}64$) reduces intermediate activations, lowering memory bandwidth and enabling tighter batching.
- **Non-iterative, single-pass conditioning:** FiLM-conditioning within the masker adds negligible overhead and preserves low latency (no iterative sampling).
- **Seamless edge/server hybrid and edge-only modes:** identical separator logic serves both modalities; with code-stream inputs, the server path remains *separator-only*.
- **Maintains codec manifold structure:** by modulating $Z/E$ rather than rebuilding $X$, CodecSep preserves periodicity/timbre/transients already organized by the codec, supporting stable, high-fidelity stems at deployment.

# C  DATASET DETAILS

## C.1  DIVIDE AND REMASTER V 2.0 (DNR-V2)

dnr-v2 Petermann et al. (2022) dataset consists of $60s$-duration artificial mixtures of speech, music, and SFX sampled from LibriSpeech Panayotov et al. (2015), Free Music Archive (FMA) Defferrard et al. (2017), and Freesound Dataset $50K$ (FSD50$K$) Fonseca et al. (2022), respectively. It includes $3,406$ ($56.7hrs$) training, $487$ ($8.13hrs$) validation, and $973$ ($16.22hrs$) test mixtures, each provided with its three individual source audios. The mixtures are generated by normalizing each source to fixed Loudness Units Full-Scale (LUFS) levels: $-17$ dB (speech), $-24$ dB (music), and $-21$ dB (SFX), with $\pm 2$ dB random perturbations. Any source exceeding a peak threshold is normalized to $0.5$ dB. The sources are mixed and normalized to $-27$ dB LUFS with additional random perturbations. The validation and test sets are trimmed for silence and split into $5s$ or $10s$ segments. Segments where sources are present for less than $50\%$ of the duration are removed, resulting in $2,852$ ($\approx 3.96hrs$) validation and $1,840$ ($\approx 5.11hrs$) test mixtures.

While originally developed for 3-stem separation, we adapt dnr-v2 to the USS setting by replacing fixed source labels with natural language descriptions. For speech or music stem, we use broad, category-level prompts (e.g., "speech," "music"), reflecting realistic usage in production workflows. In contrast, SFX sources are more complex—often containing three or more overlapping events. We generate prompts to query the SFX stem using FSD50K's hierarchical annotations, combining fine-grained class labels with their parent categories. This results in long-form, compositional queries that reflect the structure of the mixture (e.g., "dog barking, Animal, engine rumbling, motor vehicle").

## C.2  OPEN-DOMAIN BENCHMARKS

We benchmark on five open-domain datasets spanning captioned audio, environmental sounds, and large multi-event corpora: AudioCaps Kim et al. (2019), an AudioSet-derived collection of $> 46$k $10$ s YouTube clips paired with human-written captions describing the dominant sound events (used by us to synthesize training and test mixtures); ESC-50 Piczak, a curated environmental sound dataset of $2,000$ clips ($5$ s each) organized into $50$ classes with $40$ examples per class across five meta-categories (animals, natural, human non-speech, domestic, exterior/urban); Clotho-v2 Drossos et al. (2020), $6,974$ audio samples ($15$–$30$ s) each annotated with five human captions ($8$–$20$ words) covering open-domain events; AudioSet Gemmeke et al. (2017), the evaluation split of AudioSet comprising human-labeled $10$ s YouTube clips over an ontology of $632$ audio event classes in a multi-label setting; and VGGSound Chen et al. (2020a), an AudioSet-derived audio–visual corpus with $550+$ hours of $10$ s segments covering a wide variety of everyday sound categories. For AudioCaps we form both training and testing mixtures (same scale of test data as dnr-v2) by summing three clips (validation segmented into $5$ s, test preserves clips up to $20$ s), while for ESC-50, Clotho-v2, AudioSet-eval, and VGGSound we construct test-only mixtures using the same three-clip protocol.

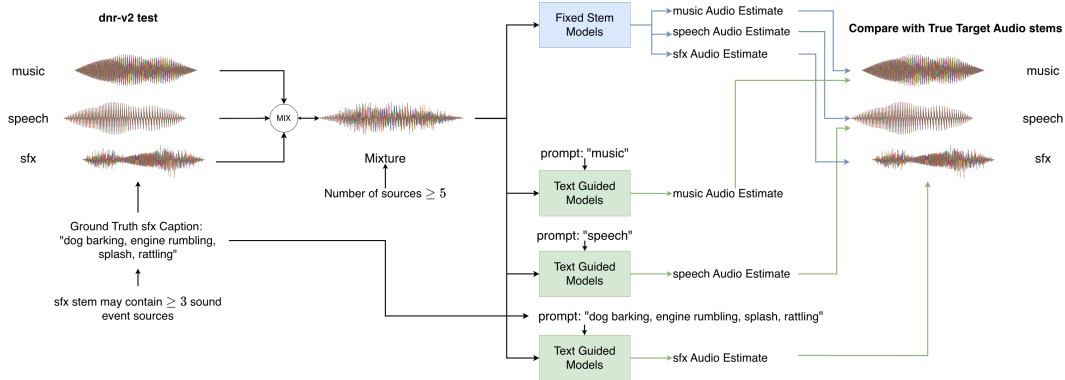

Figure 3: Evaluation workflow for dnr-v2. Each mixture contains multi-source stems: speech (often multi-speaker), music (multi-instrument), and SFX ($\geq 3$ overlapping events). Fixed-stem baselines predict a fixed set of outputs (e.g., 3 stems), whereas CodecSep and other text-guided models generate only the prompted source. Speech and music are evaluated using generic prompts, while SFX uses long-form compositional prompts listing all SFX events in each mixture. Extracted signals are compared with ground-truth category stems using SI-SDR and ViSQOL.

## D    EVALUATION DETAILS

### D.1    DIVIDE AND REMASTER V 2.0 (DNR-V2)

The dnr-v2 benchmark presents a challenging open-domain separation setting: although the dataset provides three category labels—speech, music, and sound effects—each category represents a multi-source stem. A single mixture frequently contains five to ten underlying acoustic sources, including overlapping speakers, multiple musical instruments, and several sound-effect events occurring either concurrently or in sequence. The three reported stems are therefore semantic groupings that support interpretability and reproducibility, rather than an indication that the mixture contains only three sources. Any evaluation methodology must respect this structure. Figure 3 provides an overview of our dnr-v2 evaluation workflow and highlights how these multi-source stems are handled across fixed-stem un-guided and text-guided models.

For fixed-stem unguided architectures, evaluation is performed by mapping each predicted output stem to one of the three ground-truth stems and computing SI-SDR and ViSQOL on a per-category basis. Importantly, we do not employ PIT or MixIT training objectives for these baselines; instead, we train dedicated three-stem models that directly predict speech, music, and SFX stems.

Text-guided models, including CodecSep, follow a fundamentally different inference and evaluation paradigm. Speech and music stems are recovered using generic prompts ("speech," "music"), which reliably capture their multi-source content. In contrast, SFX stems require mixture-specific prompts because sound effects span a wide and open-domain label space. For each mixture, we use a long-form compositional prompt enumerating all SFX events present in the ground truth. This ensures that the model has sufficient semantic context to extract the full SFX stem. The number of sfx events present in a mixture varies considerably. We additionally perform an ambiguous-prompt evaluation, where deliberately underspecified prompts for speech and music are used to assess robustness to vague or incomplete semantic queries. After inference, the extracted waveform for each category is directly compared with the corresponding ground-truth stem using SI-SDR and ViSQOL. This evaluation design ensures fairness between fixed-stem and text-guided systems while faithfully reflecting the multi-source structure of dnr-v2 mixtures

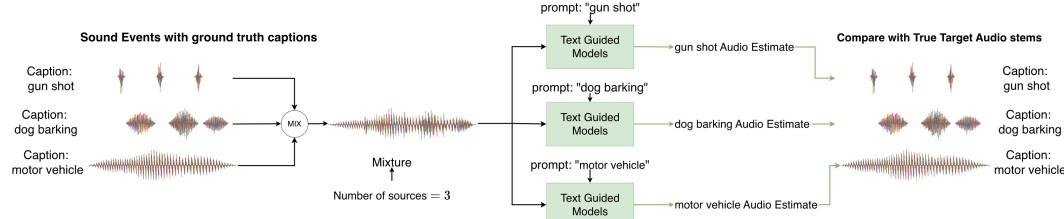

Figure 4: Evaluation workflow for the standardized three-source benchmarks (AudioCaps, ESC-50, Clotho, VGGSound, and AudioSet-eval). Following prior USS protocols, each mixture is constructed by combining three isolated events drawn from distinct classes. For each class, the corresponding textual prompt is supplied to the separator (e.g., "dog barking," "gun shot," "motor vehicle"), and the extracted signal is compared with the ground-truth isolated source using SI-SDR and ViSQOL.

## D.2    OPEN-DOMAIN BENCHMARKS

For AudioCaps, ESC-50, Clotho, VGGSound, and AudioSet-eval, we adopt the standardized three-source mixture protocol. Following established practice, each mixture is constructed by combining three isolated events drawn from different classes. Each source is then extracted by the text-guided models using its corresponding textual caption as prompt, and evaluation metrics are computed against the ground-truth isolated audio. Figure 4 illustrates the evaluation workflow used for these benchmarks, highlighting how class-specific prompts are applied and how the resulting predictions are matched against the ground-truth isolated sources. Although these datasets do not reflect the complex multi-source structure of dnr-v2, the standardized 3-way protocol enables direct benchmarking against prior work (AudioSep) under consistent conditions.

# E    TRAINING DETAILS

The complete model, including the query module $query(.)$, is trained for $400K$ iterations with DAC Kumar et al. (2023) and CLAP Wu et al. (2023) modules frozen. Validation is conducted every $5K$ iterations and test every $10K$ iterations. We use ADAM Kingma & Ba (2017) as our optimizer and train with a batch size of $4$ examples, each 2 seconds in duration, and a learning rate of $1.5e^{-4}$ on a single 24GB NVIDIA A-30 GPU. Training employs a $ReduceLRonPlateau$ Mukherjee et al. (2019) scheduler, which reduces the learning rate by a factor of $0.5$ if the validation loss does not improve for two consecutive validation checks. We train two versions of CodecSep, one using the dnr-v2 dataset and the other using AudioCaps, to evaluate performance across different training distributions. We refer to these models using the suffixes +dnr-v2 and +AudioCaps, respectively, to indicate which dataset each model was trained on.

Since TDANet and CodecFormer were originally designed for speech separation, we re-train newly initialized 3-stem versions on the dnr-v2 training set using the same configuration as CodecSep. We also train a 3-stem Sudo rm-rf model to compare against compute-efficient separators. For AudioSep, we evaluate both the publicly available pretrained model—trained on diverse datasets—and versions re-trained on dnr-v2 and AudioCaps for consistency. We include SDCodec using the official pretrained checkpoints released by the authors. Finally, we incorporate the USS-pretrained variant of BiModalSS and re-train SudoRmRf+FiLM on dnr-v2 with CLAP text conditioning. To ensure a fair comparison, all inputs to TDANet, Sudo rm-rf, AudioSep, BiModalSS, and Sudo rm-rf+FiLM undergo codec processing with a full-band stereo-capable $48\,\text{kHz}$ EnCodec during training and inference. This accounts for codec-induced distortions and artifacts, reflecting realistic deployment scenarios where audio is typically processed through compression pipelines in cloud-based systems.

Table 6: Results: Using ambiguous prompts for Speech and Music (**dnr-v2-test**)

| Model | Metric ($\uparrow$) | Music | Speech |
|---|---|---|---|
| AudioSep + dnr-v2 | SI-SDR | $-6.4^{\pm 3.29}$ | $4.1^{\pm 3.77}$ |
| | ViSQOL | $2.5^{\pm 0.57}$ | $2.6^{\pm 0.47}$ |
| **CodecSep + dnr-v2** | SI-SDR | $\mathbf{-5.6^{\pm 3.61}}$ | $\mathbf{4.2^{\pm 4.18}}$ |
| | ViSQOL | $\mathbf{2.6^{\pm 0.58}}$ | $\mathbf{2.7^{\pm 0.51}}$ |

# F    ROBUSTNESS TO PROMPT PARAPHRASING.

To probe lexical sensitivity, we re-evaluated both CodecSep + dnr-v2 and AudioSep + dnr-v2 on the dnr-v2 test split by replacing the generic training-time prompts for *speech* and *music* with three unseen paraphrases per class—*speech*: {"spoken voice", "human conversation", "people talking"}; *music*: {"instrumental music", "band playing", "melody with instruments"}. This constitutes a zero-shot paraphrase generalization test: models are trained with generic category cues but must respond to synonymic, potentially broader descriptors at inference. We observe a consistent qualitative pattern (cf. Table 6): (i) both systems exhibit the expected degradation when moving from generic to paraphrased prompts, confirming that lexical ambiguity weakens query–audio alignment; (ii) CodecSep degrades more gracefully overall, maintaining stronger separation and perceptual quality for *speech*, and retaining a small but persistent advantage for *music*; and (iii) the gap between models narrows under paraphrasing, yet the relative ranking is preserved, suggesting that FiLM-conditioned masking over structured codec latents confers robustness to synonym-level shifts. Notably, this study isolates lexical paraphrases; we did not incorporate paraphrases with explicit temporal qualifiers (e.g., "applause follows a song"), which we leave to future work.

Table 7: Generalization and transfer results for universal sound separation.

(a) Generalization on **AudioCaps-test**.

| Model | Separation | |
|---|---|---|
| | SI-SDR ($\uparrow$) | ViSQOL ($\uparrow$) |
| AudioSep | $\mathbf{-2.5}^{\pm \mathbf{12.14}}$ | $\mathbf{2.4}^{\pm \mathbf{1.08}}$ |
| AudioSep + dnr-v2 (zero-shot) | $-6.4^{\pm 11.48}$ | $2.3^{\pm 1.08}$ |
| **CodecSep + dnr-v2** (zero-shot) | $\mathbf{-6.1}^{\pm \mathbf{11.62}}$ | $2.2^{\pm 1.16}$ |
| AudioSep + AudioCaps | $-9.2^{\pm 18.71}$ | $2.3^{\pm 1.11}$ |
| **CodecSep + AudioCaps** | $\mathbf{-6.2}^{\pm \mathbf{10.58}}$ | $2.1^{\pm 1.00}$ |

(b) Transfer to **dnr-v2-test** when trained on AudioCaps (zero-shot on dnr-v2).

| Model | Metric ($\uparrow$) | Music | Speech | Sfx |
|---|---|---|---|---|
| AudioSep + AudioCaps | SI-SDR | $-14.9^{\pm 23.08}$ | $-7.1^{\pm 25.80}$ | $-14.6^{\pm 23.26}$ |
| (zero-shot) | ViSQOL | $2.4^{\pm 0.71}$ | $2.4^{\pm 0.70}$ | $2.2^{\pm 0.79}$ |
| **CodecSep + AudioCaps** | SI-SDR | $-8.5^{\pm 2.78}$ | $2.5^{\pm 2.91}$ | $-5.9^{\pm 4.33}$ |
| (zero-shot) | ViSQOL | $2.3^{\pm 0.53}$ | $2.6^{\pm 0.47}$ | $2.1^{\pm 0.72}$ |

# G    CROSS-BENCHMARK PERFORMANCE: AUDIOCAPS AND DNR-V2.

Table 7a reports generalization results on the AudioCaps-test set derived from AudioSet. AudioSep benefits from distributional alignment, having been trained on diverse datasets, including AudioSet, and consequently achieves the strongest separation performance. CodecSep+dnr-v2 generalizes well to AudioCaps and outperforms AudioSep+dnr-v2 in SI-SDR while maintaining competitive ViSQOL scores. When retrained on AudioCaps, CodecSep again outperforms AudioSep in separation quality, demonstrating strong cross-domain robustness. Table 7b further supports this trend on the more challenging dnr-v2 test set, where CodecSep+AudioCaps outperforms AudioSep+AudioCaps in SI-SDR across all sources while maintaining comparable perceptual quality. However, both models experience a performance drop on dnr-v2 due to its increased mixture complexity, often containing speech, music, and three or more overlapping SFX sources—making it significantly more challenging than the simpler mixtures seen during AudioCaps training.

Table 8: Relative gains (%) of **CodecSep** over **AudioSep** under matched training/prompt settings. Each sub-table reports percent improvements for a specific evaluation setup.

(a) DnR-v2 test set

| Metric | Relative Gain (%) | | |
|---|---|---|---|
| | Speech | Music | SFX |
| SI-SDR | +29.8 | +120.7 | +119.1 |
| ViSQOL | +26.1 | +10.5 | +0.5 |

(b) Ambiguous prompts (speech & music paraphrases)

| Metric | Relative Gain (%) | |
|---|---|---|
| | Speech | Music |
| SI-SDR | +1.2 | +13.0 |
| ViSQOL | +3.8 | +1.2 |

(c) Additional open-domain benchmarks

| Metric | AudioCaps | ESC-50 | Clotho-v2 | AudioSet | VGGSound |
|---|---|---|---|---|---|
| SI-SDR | +5.5 | +24.3 | +30.0 | +16.4 | +13.0 |
| ViSQOL | −4.3 | +2.2 | +2.4 | +2.9 | +2.9 |

(d) Training on AudioCaps

| Metric | AudioCaps-test | dnr-v2 | | |
|---|---|---|---|---|
| | | Music | Speech | SFX |
| SI-SDR | +32.5 | +43.1 | +134.7 | +59.5 |
| ViSQOL | −6.5 | −3.8 | +5.4 | -4.2 |

# H DISCUSSION OF RELATIVE−GAIN SUMMARIES.

Tables 8a–8d consolidate relative improvements of CodecSep over AudioSep under matched training data and prompt protocols, complementing the absolute results in the main text. On dnr-v2 (cf. Table 8a), CodecSep delivers large SI-SDR gains—especially for music and SFX—together with a clear perceptual lift. Under paraphrased prompts (cf. Table 8b), gains are smaller but remain positive, indicating robustness to lexical variation. Across additional open-domain benchmarks (cf. Table 8c), SI-SDR gains are consistent while ViSQOL deltas are modest, aligning with cross-domain trends reported earlier. Finally, when trained on AudioCaps (cf. Table 8d), CodecSep maintains an advantage on AudioCaps-test and yields strong improvements on dnr-v2, supporting the claim that codec-latent masking generalizes well across datasets and prompt regimes.

Table 9: Results: Extending CodecSep to 48 kHz full-band (**dnr-v2-test**)

| Model | Sampling Rate | Metric ($\uparrow$) | Music | Speech | Sfx |
|---|---|---|---|---|---|
| AudioSep (zero-shot) | 32 kHz | SI-SDR | $-2.5^{\pm 4.06}$ | $4.9^{\pm 4.21}$ | $-0.3^{\pm 5.39}$ |
| | | ViSQOL | $2.9^{\pm 0.63}$ | $3.1^{\pm 0.56}$ | $\mathbf{2.6^{\pm 0.77}}$ |
| AudioSep + dnr-v2 | 32 kHz | SI-SDR | $-5.6^{\pm 2.89}$ | $7.7^{\pm 3.0}$ | $-4.5^{\pm 3.68}$ |
| | | ViSQOL | $2.6^{\pm 0.57}$ | $2.5^{\pm 0.37}$ | $2.3^{\pm 0.7}$ |
| **CodecSep + dnr-v2** **(DAC Backbone)** | 16 kHz | SI-SDR | $\mathbf{1.2^{\pm 3.29}}$ | $\mathbf{10.0^{\pm 2.92}}$ | $\mathbf{0.9^{\pm 4.22}}$ |
| | | ViSQOL | $\mathbf{2.9^{\pm 0.57}}$ | $\mathbf{3.1^{\pm 0.45}}$ | $2.3^{\pm 0.73}$ |
| **CodecSep + dnr-v2** **(DAC Backbone)** | 24 kHz | SI-SDR | $0.2^{\pm 3.3}$ | $8.8^{\pm 2.9}$ | $0.6^{\pm 4.2}$ |
| | | ViSQOL | $2.7^{\pm 0.56}$ | $3.0^{\pm 0.44}$ | $2.3^{\pm 0.72}$ |
| **CodecSep + dnr-v2** **(DAC Backbone)** | 44.1 kHz | SI-SDR | $-2.3^{\pm 3.27}$ | $5.9^{\pm 2.48}$ | $-0.3^{\pm 3.79}$ |
| | | ViSQOL | $2.5^{\pm 0.46}$ | $2.7^{\pm 0.42}$ | $2.4^{\pm 0.68}$ |
| **CodecSep + dnr-v2** **(EnCodec Backbone)** | 48 kHz | SI-SDR | $-2.8^{\pm 3.5}$ | $5.4^{\pm 2.36}$ | $-0.5^{\pm 3.83}$ |
| | | ViSQOL | $2.4^{\pm 0.5}$ | $2.6^{\pm 0.41}$ | $2.4^{\pm 0.65}$ |

## I  BANDWIDTH SCALING: EXTENDING CODECSEP TO FULL-BAND AUDIO

Table 9 studies bandwidth scaling by swapping the frozen codec backbone from a 16 kHz DAC to a 48 kHz EnCodec (stereo–capable), while keeping the FiLM–conditioned masker and training objective unchanged. Although our paper targets *mono* separation, we evaluate the 48 kHz backbone in the same mono setting for apples–to–apples comparison. As expected, increasing the sampling rate $F_s$ makes separation harder: higher bandwidth introduces more high–frequency structure and lengthens the latent sequence ($T \uparrow$), which raises modeling difficulty and compute, and tends to reduce absolute SI-SDR/ViSQOL compared to the 16 kHz setting. Nevertheless, the codec–latent formulation remains intact—$\tilde{Z}_s = M_s \odot Z, \tilde{y}_s = Dec(\tilde{Z}_s)$—and the system continues to operate in a compact, perceptually aligned representation, preserving the same deployment pathway. Practically, complexity scales with bandwidth due to longer latent timelines and richer spectral content, but in code–stream regimes the masker–only path is unchanged; improving high–bandwidth performance is thus a matter of codec/backbone choice, capacity tuning, and data scale rather than architectural redesign. Extending this study, we additionally evaluate 24 kHz and 44.1 kHz DAC variants. The results reveal a consistent trend: for a fixed masker capacity, performance decreases monotonically as sampling rate increases—reflecting the rising difficulty of full-band, wide-bandwidth separation—yet DAC reliably outperforms EnCodec at comparable bandwidths. We attribute this to DAC's architectural advantages (factorized RVQ codes, $\ell_2$-normalized codebooks, and Snake activations), which yield more structured and perceptually aligned latents, enabling the FiLM-conditioned masker to operate more effectively even as bandwidth increases. We view these results as an initial step toward full–band (and stereo/spatial) operation within the same masking interface.

Table 10: Results: Reconstruction Performance, Universal Sound Separation (**dnr-v2-test**)

| Model | Metric ($\uparrow$) | Reconstruction | | | |
|---|---|---|---|---|---|
| | | **Mixture** | **Music** | **Speech** | **Sfx** |
| **3-Stem: Fixed Stem, Non Text-guided** | | | | | |
| TDANet | SI-SDR | $-3.3^{\pm 7.78}$ | $\mathbf{8.0^{\pm 5.29}}$ | $\mathbf{11.1^{\pm 3.32}}$ | $\mathbf{4.7^{\pm 5.11}}$ |
| | ViSQOL | $3.9^{\pm 0.35}$ | $\mathbf{4.2^{\pm 0.46}}$ | $4.5^{\pm 0.32}$ | $4.1^{\pm 0.39}$ |
| CodecFormer | SI-SDR | $-47.6^{\pm 9.51}$ | $-47.1^{\pm 10.97}$ | $-47.8^{\pm 9.65}$ | $-48.2^{\pm 9.87}$ |
| | ViSQOL | $1.0^{\pm 0.07}$ | $1.0^{\pm 0.12}$ | $1.0^{\pm 0.06}$ | $1.2^{\pm 0.47}$ |
| **CodecSep + dnr-v2** | SI-SDR | $3.4^{\pm 1.85}$ | $4.1^{\pm 3.97}$ | $6.2^{\pm 2.87}$ | $0.8^{\pm 5.16}$ |
| **(unguided, 3-stem)** | ViSQOL | $3.2^{\pm 0.20}$ | $3.0^{\pm 0.33}$ | $3.5^{\pm 0.24}$ | $3.2^{\pm 0.46}$ |
| SDCodec | SI-SDR | $\mathbf{7.0^{\pm 2.49}}$ | $7.7^{\pm 4.60}$ | $8.3^{\pm 3.26}$ | $2.5^{\pm 5.65}$ |
| | ViSQOL | $\mathbf{4.3^{\pm 0.15}}$ | $4.0^{\pm 0.28}$ | $4.4^{\pm 0.15}$ | $4.0^{\pm 0.34}$ |
| **Text-guided** | | | | | |
| AudioSep | SI-SDR | $5.5^{\pm 1.96}$ | $4.7^{\pm 5.36}$ | $11.0^{\pm 2.99}$ | $-2.0^{\pm 5.68}$ |
| (zero-shot) | ViSQOL | $4.1^{\pm 0.38}$ | $3.8^{\pm 0.65}$ | $4.6^{\pm 0.13}$ | $3.2^{\pm 0.77}$ |
| AudioSep + dnr-v2 | SI-SDR | $6.5^{\pm 2.26}$ | $8.0^{\pm 4.55}$ | $8.1^{\pm 3.35}$ | $2.3^{\pm 5.95}$ |
| | ViSQOL | $4.2^{\pm 0.18}$ | $4.1^{\pm 0.21}$ | $3.0^{\pm 0.29}$ | $3.8^{\pm 0.47}$ |
| **CodecSep + dnr-v2** | SI-SDR | $4.1^{\pm 2.06}$ | $3.9^{\pm 3.93}$ | $6.1^{\pm 2.86}$ | $0.7^{\pm 5.29}$ |
| | ViSQOL | $3.7^{\pm 0.22}$ | $3.4^{\pm 0.33}$ | $3.8^{\pm 0.24}$ | $3.5^{\pm 0.44}$ |
| **CodecSep + dnr-v2** | SI-SDR | $\mathbf{12.2^{\pm 2.42}}$ | $\mathbf{12.6^{\pm 3.81}}$ | $\mathbf{13.6^{\pm 2.59}}$ | $\mathbf{8.7^{\pm 4.17}}$ |
| (ablate **Masker**) | ViSQOL | $\mathbf{4.4^{\pm 0.14}}$ | $\mathbf{4.1^{\pm 0.31}}$ | $\mathbf{3.9^{\pm 0.34}}$ | $3.8^{\pm 0.54}$ |
| AudioSep + AudioCaps | SI-SDR | $6.7^{\pm 2.52}$ | $8.1^{\pm 4.63}$ | $8.4^{\pm 3.21}$ | $2.4^{\pm 6.12}$ |
| (zero-shot) | ViSQOL | $4.2^{\pm 0.19}$ | $4.1^{\pm 0.21}$ | $4.2^{\pm 0.21}$ | $\mathbf{3.8^{\pm 0.46}}$ |
| **CodecSep + AudioCaps** | SI-SDR | $0.6^{\pm 1.89}$ | $-0.2^{\pm 5.15}$ | $-11.^{\pm 5.21}$ | $1.2^{\pm 4.84}$ |
| (zero-shot) | ViSQOL | $3.3^{\pm 0.23}$ | $2.9^{\pm 0.64}$ | $1.7^{\pm 0.48}$ | $3.4^{\pm 0.41}$ |

## J   FURTHER STUDIES: RECONSTRUCTION PERFORMANCE.

Table 10 assesses performance under a single-source reconstruction setting on the dnr-v2-test set, where each model is prompted to reproduce the input source. In addition, we report mixture reconstruction scores obtained by summing the separated sources and comparing them to the original mixture.

Among non-text-guided models, TDANet yields the best single-source reconstruction, while SDCodec performs better on mixture reconstruction. Replacing decoder-style generation in CodecFormer with a Transformer masker over codec latents in CodecSep+ dnr-v2, (unguided 3-stem) markedly improves both per-stem and mixture reconstruction fidelity and perceptual quality. Masking modulates information already organized in the codec manifold ($Z$) rather than re-synthesizing it from scratch, avoiding collapse/artifacts and yielding tighter mixture consistency.

Among the text-guided models, CodecSep achieves reconstruction performance comparable to the pre-trained and retrained AudioSep variants across all source types. AudioSep consistently excels in reconstruction due to its STFT-based masking pipeline, which enables more controlled and artifact-free waveform synthesis. However, CodecSep surpasses the pretrained AudioSep in SFX reconstruction—across both SI-SDR and ViSQOL on dnr-v2. Notably, the masker ablated CodecSep variant delivers the best reconstruction performance on dnr-v2. With isolated single-source input and matching prompts, direct conditioning minimally disturbs the NAC latent space, allowing high-fidelity reconstruction in this variant. However, strong mixture reconstruction despite poor separation suggests source leakage across separated outputs.

## K   LIMITATIONS AND CLARIFICATIONS.

We discuss the limitations of our work as follows–

(1) *Data and prompts.* Training data scale and prompt diversity are modest relative to open-domain audio. As shown in Table 2, finer SFX supervision sharpens SFX extraction *and* improves speech/music stems; larger, more heterogeneous corpora spanning multiple prompt granularities—including temporal/relational cues—should yield further gains.

(2) *Temporal prompting.* While CodecSep is robust to synonymic paraphrases, we did not evaluate prompts with explicit temporal structure (e.g., causal ordering), which remains an open direction.

(3) *Perceptual SFX quality.* In some settings, SFX perceptual quality trails the best competing scores despite superior SI-SDR; improving SFX naturalness without sacrificing separation is future work.

## L   DECLARATION OF LLM USAGE.

LLM is used only to aid or polish writing and does not impact the core methodology, scientific rigorousness, or originality of the research.

