# OpenReview forum: "CodecSep: Prompt-Driven Universal Sound Separation on Neural Audio Codec Latents"
_ICLR.cc/2026/Conference — Submitted to ICLR 2026_

### Official Review · Reviewer_TLeL · 2025-10-27

**Soundness:** 3
**Presentation:** 2
**Contribution:** 2
**Rating:** 4
**Confidence:** 4

**Summary:**

The authors introduce CodecSep, a Neural Audio Codec (NAC)-based model for universal, text-driven on-device separation. The model integrates a former NAC model (the DAC compression model) with a transformer masker modulated by CLAP-derived FiLM parameters. Evaluated on six open-domain benchmarks with matched training and prompt settings, CodecSep outperforms at least one other model (AudioSep) in separation fidelity (SI-SDR), achieves comparable perceptual quality (ViSQOL), and obtains competitive results on fixed-stem baselines (compared to TDANet, CodecFormer or SDCodec). In addition, it is shonw that the proposed model is of much lower complexity than spectrogram-based models like AudioSep, while remaining fully bitstream-compatible for efficient on-device audio separation.

**Strengths:**

The main strengths of the paper are:

-	The overall concept of the proposed model (exploiting text embeddings to guide audio source separation) is interesting and is worth further investigation.

-	The proposed approach (performing masking in the latent domain) is sound and well-motivated.

-	Many experiments were conducted to support the merits of the proposed approach

**Weaknesses:**

The main weaknesses of the paper are:

-	The paper is on the overall difficult to read and follow. Technical details about the method are given rather early in the paper (for instance, beginning of section 3.1 with DAC model), and provided as a list of bullet points (DAC.., Text-guided.., Why NAC latents… etc).  The model itself is only described in 3.2 (in a very condensed section). As a result, the paper misses a central section describing the model in conceptual form first and then with technical details. An additional conceptual scheme/diagram precising how and when discrete information is used would also facilitate the readability.

-	The experimental plan is rich, diverse and solid, but maybe too diverse which dilutes the most important outcomes (the main message). For instance, the experiments on using ambiguous prompts are probably less conclusive and if omitted, more place would be given to better discuss the results.

-	There are no sound examples provided on a demo page to illustrate the merits and limits of the method.

-	The limitations are not discussed in the main paper (only in appendix).

**Questions:**

-	About Perceptual test: How were chosen the 20 participants? are they expert in audio perceptual tests? in audio coding/separation test ? How were chosen the 20 dnr-v2 stem test mixtures? What means exactly “per-clip sfx prompts”? (maybe give an example).

-	Results on Table 2 could be further discussed or clarified: It is said :””..COdecSep is competitive with or stronger than fixed-stem baselines ..”. From the table it rather seems that SD-Codec, without text guiding, is significantly outperforming the proposed model (even on SfX) with generic prompt.

-	The use of continuous or discrete information in the proposed model is not always clear (and it is mentioned at some places that the quantization is optional). Are all results obtained with no discrete information (e.G. by passing the RVQ) ? Which results have been obtained using the optional quantization step ?

-	There are no mention about potential code release, although the technical details provided seem sufficient to reproduce the work. Do you plan to release the code to allow better reproducibility?

---

> ### Author Response · Authors · 2025-11-13
> **Clarifications on Novelty, Readability, Experimental Scope, Continuous–Discrete Usage, Subjective Evaluation, and Code Availability**
>
> We thank the reviewer for their constructive feedback. Below, we clarify key points and summarize the main strengths and contributions.
>
> ---
>
> **1. Novelty and Core Contributions**
>
> Masking in the latent domain of neural audio codecs (NACs) has not been explored before.
> While AudioSep encodes spectrograms to a learned latent space before masking, CodecSep directly operates on pretrained NAC latents—eliminating the need to learn a new latent space  and enabling faster, more stable training (lines 152- 155) . Since modern edge pipelines already convert audio into codec bitstreams (e.g., DAC) for storage and low-latency communication, CodecSep achieves massive compute efficiency by separating directly on these code streams without decode–re-encode cycles.
>
> We clarify that SDCodec, TDANet, and CodecFormer are **fixed-stem baselines**, not competitors.  AudioSep remains our principal baseline.
>
> **Core contributions:**
> • First integration of NACs for *text-guided universal source separation (USS)*.
> • First approach capable of operating directly on NAC code streams.
> • Theoretical justification, supported by prior NAC studies, explaining why codec latents are superior to spectrograms or learned embeddings for masking.
> • Extensive experiments across six benchmarks showing consistent improvements over AudioSep.
> • Evidence that NAC latents are sufficiently structured and disentangled to support lightweight masking for source extraction—unlike CodecFormer, which assumes no latent structure and decodes full sources.
>
> ---
>
> **2. Readability and Conceptual Flow**
>
> Section 3 already opens with a conceptual overview of our model (Lines 104–123) and **Figure 1** summarizing the full pipeline. 3.1 then introduces:
> - *(i)* the NAC backbone and compression characteristics (Lines 147–155);
> - *(ii)* the codec prior and motivation for masking rather than decoding (Lines 156–172);
> - *(iii)* the role of **continuous vs. discrete** latents and the deployment path (Lines 173–196);
> - *(iv)* computational advantages (Lines 197–213).
> Section 3.2 subsequently details the masker architecture and training procedure.
>
> TL;DR:
>
> **Why NAC latents:**
> NAC latents are denoised, disentangled, and 32–64× more compact than spectrograms while retaining perceptually rich structure.
>
> **Why Masking:**
>  The hierarchical quantization of RVQ induces structure that enables source extraction via simply masking latents.
>
> **Use of Continuous vs. Discrete:**
>
> Regarding reviewer question:
>
> Training uses continuous latents \(Z\) for stability; inference reconstructs discrete embeddings \(E \approx Z\) from codec bitstreams in a zero-shot “codes-in → codes-out” mode. The discrete variant (*CodecSep codes-in:codes-out*, Table 1) outperforms AudioSep, while all other results use continuous inference.
>
> As suggested, we will add a diagram to present discrete inference and deployment scenario.
>
> **Deployment:**
> CodecSep processes codec bitstreams directly. Only the lightweight masker runs on the server, minimizing latency and compute while maintaining full codec compatibility for edge deployment.
>
> ---
>
> **3. Experimental Scope**
>
> Tables 1–6 (main) and 7–9 (appendix) comprehensively evaluate:
> - **Table 1:** USS comparison with AudioSep (continuous + discrete inference).
> - **Table 2:** Fixed-stem ablation using class labels (“speech,” “music,” “SFX”) against SDCodec, TDANet, CodecFormer.
> - **Table 3:** Transformer masker vs. decoder—highlighting large gains and validating NAC latent disentanglement.
> - **Tables 4–5:** Open-domain and ambiguous-prompt benchmarks (robustness to unseen prompts).
> - **Table 6:** Inference complexity.
> - **Tables 7–9:** Cross-benchmark generalization, relative-gain summary, and bandwidth-scaling (e.g., 48 kHz Encodec).
>
> This scope provides rigor rather than dilution; ambiguous-prompt tests demonstrate CodecSep’s robustness to prompt shifts.
>
> ---
>
> **4. Subjective Evaluation**
>
> We conducted MOS-LQS tests with 20 professional evaluators from our audio research lab, each rating 20 random DnR-V2 test mixtures.
> “Per-clip SFX prompts” denote compositional captions (e.g., “splash, boat, crinkling”) containing overlapping and non-overlapping events.
> All separated outputs used in MOS tests are provided in the supplementary material along with **subjective_assessment.md**, summarizing listener feedback.
>
> We encourage the reviewer to listen to the audio outputs.
>
> ---
>
> **5. Limitations and Code Release**
>
> Limitations will now appear in the main paper (with the 10-page allowance).
> A clean, modular, and reproducible codebase is already provided as supplementary. (Line 485).
>
> ---
>
> **6. Clarification on Table 2**
>
> As detailed in Lines 347–359, the 3-stem CodecSep variant surpasses CodecFormer on SFX and remains competitive with TDANet and SDCodec.
> The universal version, trained with natural captions (Table 1), outperforms CodecFormer across all sources and maintains competitiveness with TDANet and SDCodec.
>
> ---

---

> > ### Comment · Reviewer_TLeL · 2025-11-24
> > **Rebuttal additional comments**
> >
> > Thank you authors for your rebuttal and the additional information provided. Some of the points mentioned in the review are convincingly answered.
> >
> > Indeed, the supplementary material did include the code and details on the perceptual experiments and results (with the different audio signals). This indeed directly answers some of my comments and strengthens the paper.
> >
> > About Clarity: From your response, I do not see clearly how the paper will become more readable after revision. I believe that all my comments on this point remain valid.

---

> ### Author Response · Authors · 2025-11-13
> **Revision Plan — Updates to be made to Manuscript Before 25th November 2025**
>
> 1. As suggested, we will add a schematic diagram illustrating both the deployment scenario (edge–server operation) and the use of discrete inference (“codes-in → codes-out”) to improve conceptual clarity.
> 2. We will move the limitations section from the appendix to the main content (enabled by the 10-page allowance) to make our scope and assumptions more transparent.
> 3. As suggested, we will defer Table 7 ambiguous prompts experiment to Appendix to give more space to discuss the additional experiments added as suggested by pDqQ.

---

> ### Author Response · Authors · 2025-11-24
> **[Part A] Revised Manuscript Submitted and Ready for Review – Enhanced Experimental Validations for Reviewer Verification**
>
> ## **New Experiments Added to Revised Manuscript**
> ___
> ### **1. Comparison with BiModalSS and Sudo rm-rf Variants.**
> As suggested by Reviewers **pDqQ** and **wCZU**, we now include comprehensive comparisons against USS-pretrained **BiModalSS** [E], and **dnr-v2–retrained** versions of **Sudo rm-rf** [A] and **Sudo rm-rf + FiLM** [D, F]. CodecSep consistently outperforms all these baselines.
>
> - BiModalSS [E] is introduced and discussed in **Section 2, lines 89–90** in revised manuscript.
> - Sudo rm-rf [A] is presented in **Section 2, lines 81–85** in revised manuscript.
> - Sudo rm-rf + FiLM [D, F] and the contrast between text-guided vs class-guided separation are discussed in **Section 2, lines 91–93** in revised manuscript.
> - All models are summarized in the **Evaluation paragraph (Section 4)** at **lines 297–301** in revised manuscript, and their training setup is described in the **Training paragraph** at **lines 317–320** in revised manuscript.
> - BiModalSS [E] and Sudo rm-rf + FiLM [D, F] are added to **Table 1**. Since AudioSep substantially outperforms both, we exclude them from deeper analysis; the relevant discussion appears in **Section 4.1 (lines 360–369)** in revised manuscript.
> - Sudo rm-rf [A] is included as a fixed-stem baseline in **Table 2**, with analysis in **Section 4.1 (lines 375–377)** in revised manuscript. Despite its efficiency, CodecSep achieves slightly stronger performance.
>
> ___
>
> ### **2. Holistic Compute Benchmarking.**
> Addressing Reviewer **pDqQ**, we expand our compute benchmarking and add Sudo rm-rf to the comparison. CodecSep demonstrates **dramatic compute-efficiency advantages** over both AudioSep and Sudo rm-rf, especially in code-stream inference.
>
>  **Table 5** in revised manuscript now contains **six subtables**:
> - **5.1** GMACs
> - **5.2** Inference Time
> - **5.3** Parameter Count
> - **5.4** Parameter-Only Memory Footprint
> - **5.5** Forward/Backward Memory Footprint
> - **5.6** Full Memory Footprint
>
> A detailed discussion is provided in **Section 4.1 (lines 432–453)** in revised manuscript.
>
> We emphasize that this holistic compute analysis directly highlights CodecSep’s architectural advantage—its ability to operate natively over code streams—and also addresses the reviewer TLeL's concerns regarding whether our experiments demonstrate the main motivation behind using CodecSep.
>
> ___
>
> ### **3. Bandwidth Scaling Experiments.**
> Responding to Reviewer **wCZU**, we include **24.1 kHz** and **44.1 kHz** DAC variants in our bandwidth-scaling experiments. We observe a consistent pattern:
> with fixed masker capacity, **separation performance declines monotonically as bandwidth increases**, reflecting the intrinsic difficulty of wide-band separation. At all sampling rates, **DAC outperforms EnCodec** at matched bandwidths.
>
> These results are discussed in **Section 4.1 (lines 472–478)**  , with full experiments in **Appendix I** in revised manuscript.
>
>
>
> ---
>
> [A] Tzinis et al, 2022. Compute and memory efficient universal sound source separation. Journal of Signal Processing Systems
>
> [D] Tzinis et al, 2023, June. Optimal condition training for target source separation. ICASSP 2023
>
> [E] Mahmud et al, 2024. Weakly-supervised audio separation via bi-modal semantic similarity. ICLR 2024
>
> [F] Tzinis et al, 2022. Heterogeneous Target Speech Separation. Interspeech 2022

---

> ### Author Response · Authors · 2025-11-24
> **[Part B] Revised Manuscript Submitted and Ready for Review – Enhancements on Readability and Clarity for Reviewer Verification**
>
> ## **Enhancements on Readability and Clarity in Revised Manuscript**
> ---
> ### **1. Structural Edits and Variable Definitions**
>
> - We compressed **Section 1 (lines 54–68)** in the original manuscript to **lines 54–65** in the revised version for conciseness.
> - As suggested by reviewer **QbS3**, we added citations for datasets at their **first mention** (lines **54–57**).
> - We defined all previously undefined variables in **Section 3.1**, now explicitly stated in **lines 132 and 141** of the revised manuscript.
> - To improve clarity in the projection process and model definitions, we now present spectrogram separators as **\(Spec(.)\)** instead of \(g(.)\) in **lines 136–142**, and we added missing term definitions in **lines 140–141**.
> - We also rename the CodecFormer decoder-based source estimator to **\(Gen(.)\)** (instead of \(Sep(.)\)) in **line 167** for consistency.
>
> ---
>
> ### **2. Clarifications on Projection Process, Deployment Advantages, and NAC-Specific Components**
>
> - Addressing reviewer **QbS3**, in **Section 3.1 (lines 143–146)** of the revised manuscript, we clarify in the projection process for text-guided source separation on NAC latents that the \( Enc(.) \) and \(Dec(.)\) modules are specific to **DAC** and its variants, as these **high-fidelity neural audio codecs** are designed for **accurate waveform reconstruction** and therefore maintain strict **time-alignment**.
>
> - As suggested by reviewer **QbS3**, we now include a clear **projection process illustrating the deployment path and CodecSep discrete inference** in **lines 201–204**.
> - We revised the **Section 3.1 paragraph on Deployment Advantages** (previously lines 198–213) to discuss both **spectrogram separators \(Spec(.)\)** (AudioSep) and **waveform separators \(Wave(.)\)** (Sudo rm-rf), noting that waveform separators also map audio to STFT-like latents via 1D convolvolutions and therefore inherit the same limitations as spectrogram systems. This clarification is added in **lines 158–160**.
> - We reorganize this section as **“Deployment advantages over audio-stream separators”**, explaining that both spectrogram and waveform separators operate on the audio stream and thus incur redundant **decode → separate → re-encode** cycles in edge–server deployments. CodecSep avoids these with **direct code-stream separation**, as reflected in **Table 5** while still outperforming baseline audio stream text-guided separators in USS. These points are presented in **lines 205–211**, followed by compute cost equations in **lines 211–215**.
>
> ---
>
> ### **3. Figures and Diagrams for Clarity**
>
> - As suggested by reviewer **TLeL**, we added **Figure 2** (lines **216–232** of the revised manuscript) to clarify **CodecSep’s discrete inference workflow** and the **deployment scenario**, including when and how the masker is applied—addressing questions raised by reviewer **QbS3**.
> - Addressing reviewer **pDqQ**, we added detailed diagrams of our evaluation pipelines in **Appendix D**:
>   - **Figure 3**: evaluation workflow for **dnr-v2**,
>   - **Figure 4**: evaluation workflow for **other open-domain benchmarks**.
>   We reference Appendix D in **Section 4**, lines **309–310** of therevised manuscript.
>
> ---
>
> ### **4. Additional Clarifications (Metrics, Interpretability Justification, and Failure Modes of PIT/MixIT)**
>
> - As recommended by reviewer **QbS3**, we explicitly state that **ViSQOL is a proxy MOS metric** in **Section 4, line 305** of the revised manuscript.
> - We provide a detailed presentation of **MixIT/PIT failure modes** and explain why **text-guided separation** is better suited to USS in **Appendix A**, referenced in **line 81** of the revised manuscript.
> - To address reviewer **QbS3**’s request for stronger interpretability justification, we provide **empirical evidence** that NAC latents contain sufficient internal structure to support mask-based separation. Additionally, in **Section 5 (lines 537–539)** of the revised manuscript, we highlight that the predicted masks and FiLM responses indicate how CodecSep selectively activates latent components, outlining a clear direction for **future interpretability work** through mask visualization and reconstruction-consistency analyses.
>
> - As suggested by reviewer **pDqQ**, all SI-SDR values are now reported to **one decimal place** in revised manuscript, since smaller differences are not perceptually meaningful.
>
> ---
>
> ### **5. Corrections and Minor Fixes**
>
> - As suggested by reviewer **wCZU**, we corrected the typo in **Figure 1**, replacing *BERT* with **CLAP**.
> - Reviewer **TLeL** recommended deferring prompt-ambiguity experiments; these are now moved to **Appendix F**, with a brief summary in **Section 4.1 (lines 455–462)**.
> - Due to page limits, we place **limitations** in Appendix K, but mention them in **Section 5 (Conclusion), line 539**. Although limitations are not strictly required for ICLR, we follow NeurIPS practice and include them for completeness.
>
> ___

---

> > ### Comment · Reviewer_TLeL · 2025-11-25
> >
> > Thank you authors for the revised manuscript. A significant number of previous comments are well answered and the revised manuscript is of higher clarity and readibility. I will raise my evaluation score.

---

### Official Review · Reviewer_pDqQ · 2025-10-31

**Soundness:** 2
**Presentation:** 3
**Contribution:** 1
**Rating:** 2
**Confidence:** 5

**Summary:**

This paper introduces a model, designated CodecSep, which seeks to address a significant trade-off in the field of audio separation. Currently, a dichotomy exists between two classes of models; on one hand, flexible text-prompted systems such as AudioSep offer the capability to separate a wide array of sounds based on textual descriptions, however these models are characterized by their substantial computational requirements, rendering them unsuitable for on-device applications. Conversely, highly efficient models predicated on neural audio codecs are limited to a predefined set of separable categories, for instance speech, music, and sound effects. The main contribution of CodecSep is the reconciliation of these two paradigms, it facilitates text-driven, universal sound separation by operating directly within the compressed latent space of a pre-trained neural audio codec. Instead of employing a computationally intensive spectrogram-based architecture, CodecSep utilizes a lightweight Transformer masker guided by textual prompts through the application of Feature-wise Linear Modulation (FiLM) conditioning. This design simplifies the separation task considerably, and the final system demonstrates efficiency which makes it viable for real-world deployment on edge devices, but also achieves state-of-the-art separation quality.

**Strengths:**

1. The proposed model outperforms larger models on the same conditional separation task.
2. The evaluation is thorough enough for publication to ICLR and to make the main points that the authors want to pass to the reader.
3. The authors try to somehow quantify the complexity of their models by using GMACs, however, they did not make a holistic analysis of their computational requirements for their separation model [A] (e.g. the main concerns of execution time and actual memory footprint are missing to show that the proposed model can be deployed on a real-world device)

[A] Tzinis, E., Wang, Z., Jiang, X. and Smaragdis, P., 2022. Compute and memory efficient universal sound source separation. Journal of Signal Processing Systems, 94(2), pp.245-259.

**Weaknesses:**

The paper has many weaknesses and I will try to write them down with decreasing order of significance.

1. The authors completely omit to place their contribution to a reasonable context with the literature and show what is the real novelty being introduced with their method. For example, separation into the latent space of general sounds has been introduced for quite a while [A] and its variations like speech enhancement [B]. Moreover, the context of flexible condition based separation using FiLM layers with efficient models has also been extensively analyzed in the past literature [D, E, F]. The authors completely omit to compare against the proposed methods in the literature, which also display state-of-the-art results with much smaller models and they choose to only compare against a much more computationally heavy model. The authors need to perform a head-to-head comparison with the methods in the literature that have exactly the same setup (Figure 1 in this paper shows exactly the same conditioning mechanism is used as the one proposed in [F] with the only difference that a more complex encoder / decoder pair is used and instead of U-ConvBlocks, transformer based layers are used in the separator).
2. The authors did not make a holistic analysis of their computational requirements for their separation model (e.g. the main concerns of execution time and actual memory footprint are missing to show that the proposed model can be deployed on a real-world device). GMACs is not the way to show that a model is capable of running efficiently on an edge device. The authors can try CPU / smaller GPUs to show the efficiency of their model and compare that to state-of-the-art models in the literature like [D, F] which are currently being used for on-device implementations.
3. Please remove the second decimal from SNR metrics, a human being cannot understand differences of 0.1 dB.
4. The model's performance is fundamentally capped by the quality of the frozen DAC models it's built upon (which is a natural problem with frozen auto-encoder methods). The same problem exists for all methods that they are using latent targets for separation [B]. It would be nice to show what is the maximum separation performance one could gain by using the ideal targets in this context, similar to [B] and a much more novel contribution could be to show how to make those targets easier to estimate (that could be another paper though).


[B] Tzinis, E., Venkataramani, S., Wang, Z., Subakan, C. and Smaragdis, P., 2020, May. Two-step sound source separation: Training on learned latent targets. In ICASSP 2020-2020 IEEE International Conference on Acoustics, Speech and Signal Processing (ICASSP) (pp. 31-35). IEEE.

[C] Casebeer, J., Vale, V., Isik, U., Valin, J.M., Giri, R. and Krishnaswamy, A., 2021, June. Enhancing into the codec: Noise robust speech coding with vector-quantized autoencoders. In ICASSP 2021-2021 IEEE International Conference on Acoustics, Speech and Signal Processing (ICASSP) (pp. 711-715). IEEE.

[D] Tzinis, E., Wichern, G., Smaragdis, P. and Le Roux, J., 2023, June. Optimal condition training for target source separation. In ICASSP 2023-2023 IEEE International Conference on Acoustics, Speech and Signal Processing (ICASSP) (pp. 1-5). IEEE.

[E] Mahmud, T., Amizadeh, S., Koishida, K. and Marculescu, D., 2024. Weakly-supervised audio separation via bi-modal semantic similarity. arXiv preprint arXiv:2404.01740.

[F] Tzinis, E., Wichern, G., Subramanian, A.S., Smaragdis, P. and Le Roux, J., 2022. Heterogeneous Target Speech Separation. In Proc. Interspeech 2022 (pp. 1796-1800).

**Questions:**

THe authors mention implicitly that although other methods in the literature "assume a fixed maximum number of sources ", their method does not. Considering that this is true, why do the authors choose to only perform experimetns with up to 3 sources? This is not a simple omission but also misleading to the way the paper is written, if the authors want to claim that not defining the maximum number of sources would be beneficial to some extent, why don't they also show the appropriate experimetns and compare against the methods that they consider inferior to show that this is an important aspect?

---

> ### Author Response · Authors · 2025-11-21
> **[Part A] Response to Reviewer's point 1 in Weaknesses, Clarifications on Novelty and Core Contribution**
>
> # **1. Novelty and Core Contributions**
>
> Masking directly in the latent domain of neural audio codecs (NACs) has not been explored in prior work.
> While AudioSep projects spectrograms into a learned embedding space before masking, CodecSep operates directly on pretrained NAC latents, eliminating the need to learn a new latent space and yielding far faster and more stable training. Importantly, modern edge pipelines already represent audio as codec bitstreams (e.g., DAC) for storage and low-latency communication. CodecSep can exploit this by performing separation on the code stream itself, avoiding any decode → separate → re-encode cycle and delivering substantial compute savings.
>
> **Core Contributions:**
>
> • First NAC-based approach for text-guided universal source separation (USS).
>
> • First separator to operate directly on NAC code streams, enabling practical edge deployment.
>
> • Theoretical justification, grounded in prior NAC analyses, that shows why codec latents are more structured, disentangled, and better suited to masking than spectrograms or learned embeddings.
>
> • Extensive evaluation across six benchmarks, consistently outperforming AudioSep with >25× lower compute.
>
> • Empirical evidence that NAC latents are sufficiently structured and disentangled to support lightweight masking for source extraction—unlike CodecFormer, which assumes no latent structure and decodes full sources.
>
> ### **To be noted (clarifying the true source of novelty):**
> To prevent any misunderstanding about where the novelty of CodecSep lies, we emphasize the following:
>
> - **We intentionally use standard FiLM conditioning**, exactly as in AudioSep.
>   We do *not* introduce a new conditioning mechanism. Conditioning is a modular component that can be replaced in future work (e.g., attention-based conditioning used in BimodalSS [E]).
>
> - **We use the standard SI-SDR minimization training objective**, identical to established source separation models.
>   We introduce *no* new training recipe, loss function, or optimization strategy. The training procedure is also modular and can be replaced in future work (e.g., strategies introduced in [B, D, E, F]).
>
> - **Our novelty lies in the framework and paradigm shift**, not in conditioning or training:
>   - We show, both theoretically and empirically, that **NAC latent spaces are structured and separation-friendly**, making them suitable for direct masking.
>   - We introduce the **first codes-in → codes-out separation pathway**, enabling real deployment where audio is already passed around as codec bitstreams on edge devices.
>   - We demonstrate **major efficiency gains** (25×–54× compute reduction compared to AudioSep when operating on code streams) arising purely from this design.
>
> - **Conditioning and training are intentionally kept standard** so that improvements can be attributed to the *core idea*:
>   **shifting source separation into the pretrained NAC latent space**, rather than the spectrogram domain.
>
> - **Sudo rm-rf [A], Sudo rm-rf + FiLM [D, F], and BimodalSS [E] were not included in the original submission**, and we have now added them for completeness.
>   - **Sudo rm-rf** and the FiLM-augmented variant serve as waveform-based baselines.
>   - **BimodalSS [E]** uses the *same* spectrogram-based ResUNet separator as AudioSep; its novelty lies only in conditioning/training strategies.
>   We observe CodecSep to outperform all these baselines as well, justifying our performance claims.
>
> - **A holistic computational analysis** (inference time, memory footprint, and architecture-only complexity) was not included in the original submission; we have now added it for completeness.
>   - We now evaluate inference time, parameter/memory footprint, and include Sudo rm-rf as it is explicitly designed for low-compute separation.
>   - We observe CodecSep to **outperform both AudioSep and Sudo rm-rf**, confirming that our claims on compute efficiency remain valid and justified.
>
> - **We already evaluated against all major separation families**, including:
>   - *Non-codec separators:* TDANet (spectrogram-based)
>   - *Codec-based separators:* SDCodec, CodecFormer
>   - *Text-guided separators:* AudioSep  (spectrogram-based)
>
>   The additional baselines now provide a complete and representative landscape.
> ___
>
> [A] Tzinis et al, 2022. Compute and memory efficient universal sound source separation. Journal of Signal Processing Systems
>
> [D] Tzinis et al, 2023, June. Optimal condition training for target source separation. ICASSP 2023
>
> [E] Mahmud et al, 2024. Weakly-supervised audio separation via bi-modal semantic similarity. ICLR 2024
>
> [F] Tzinis et al, 2022. Heterogeneous Target Speech Separation. Interspeech 2022

---

> ### Author Response · Authors · 2025-11-21
> **[Part B] Continuation of Response to Reviewer's point 1 in Weaknesses, Addressing Missing-Baseline Concerns, Table reporting results for Missing Baselines (CodecSep outperforms all), Comparisons with ConvTasNet variants proposed in papers [E, B] + Clarification of point 4 in Weaknesses**
>
> ### **Table 1 — USS Results on dnr-v2 Test Set**
>
> ---
>
> #### **Table 1a — SI-SDR (dB)**
>
> | Model                          | Music            | Speech           | SFX              |
> |--|-|--|---|
> | **[E] BimodalSS**              | -6.8 ± 2.73      | 1.8 ± 2.78       | -6.36 ± 3.57     |
> | **[D], [F] Sudo rm-rf + FiLM** | -6.7 ± 2.62      | 2.0 ± 2.76       | -6.6 ± 3.71      |
> | **[A] Sudo rm-rf (Fixed Stem)**| -0.9 ± 4.01      | 9.0 ± 2.60       | 0.6 ± 4.74       |
> | **CodecSep**                   | **1.2 ± 3.29**   | **10.0 ± 2.92**  | **0.9 ± 4.22**   |
> | **CodecSep (codes-in : codes-out)** | -0.2 ± 3.55 | 8.3 ± 2.60       | -1.0 ± 4.20      |
>
> ---
>
> #### **Table 1b — ViSQOL-MOS**
>
> | Model                          | Music            | Speech           | SFX              |
> |-|-|-|-|
> | **[E] BimodalSS**              | 2.5 ± 0.57       | 2.6 ± 0.51       | 2.3 ± 0.72       |
> | **[D], [F] Sudo rm-rf + FiLM** | 2.5 ± 0.56       | 2.6 ± 0.51       | 2.2 ± 0.72       |
> | **[A] Sudo rm-rf (Fixed Stem)**| 2.7 ± 0.59       | 2.9 ± 0.45       | 2.3 ± 0.72       |
> | **CodecSep**                   | **2.9 ± 0.57**   | **3.1 ± 0.45**   | **2.3 ± 0.73**   |
> | **CodecSep (codes-in : codes-out)** | 2.5 ± 0.52 | 3.0 ± 0.44       | 2.3 ± 0.67       |
>
> ---
>
>
> ### **Comparison with BimodalSS [E]**
>
> For clarity, BimodalSS [E] is **not** a new separator architecture: as stated in Sec. 3.1 of [E], it uses the same spectrogram-based ConvTasNet/ResUNet separator as AudioSep. Its novelty lies only in CLAP-based attention conditioning and training, and the added cross-attention makes it even heavier than AudioSep.
>
> In contrast, CodecSep deliberately uses **standard SI-SDR training** and **standard FiLM conditioning** (as in AudioSep). Our contribution is the **framework shift**: showing that NAC latents form a structured, separation-friendly space enabling direct masking and a practical **codes-in → codes-out** path that avoids the decode → separate → re-encode cycle required by all spectrogram-based methods, including [E].
>
> We use AudioSep as the representative spectrogram-based USS baseline because it matches the separator in [E]. We also report the pretrained [E] checkpoint on dnr-v2, and observe CodecSep outperforming it in both continuous and discrete settings (cf. Table 1 in comment). CodecSep can easily adopt the CLAP-attention strategy used in [E], but such conditioning changes are orthogonal to our core contribution.
> ____
> ### **Comparison with [B]**
>
> [B] also uses the AudioSep / ConvTasNet-style spectrogram separator, but without conditioning. The method explicitly separates latent-representation learning from separation: the encoder–decoder is first trained to learn a latent space, and only then is a masker trained to operate on these learned latents.
> The separation loss in [B] is computed in the latent domain, whereas in CodecSep we compute SI-SDR on the true audio targets by decoding the estimated source back to waveform.
>
> Since, TDANet has been validated (in Li et al's TDANet paper )to outperform ConTasNet we do not include [B] as a fixed stem baseline.
>
> Finally, because [B] relies on the same ConvTasNet/ResUNet separator architecture as AudioSep, its limitations follow directly from the weaknesses of spectrogram-based separation.
> ___
> ### **Clarification on Point 4 in Weaknesses**
>
> The reviewer’s concern applies to methods that compute separation losses **directly in the latent domain**, such as [B], where performance is indeed capped by the quality of latent reconstruction. This limitation does **not** apply to CodecSep.
>
> In our framework, the estimated source is **decoded back to waveform**, and the separation loss is computed as **SI-SDR against the true audio targets**. Therefore, CodecSep **does not rely on latent targets for training**.
>
> That said, one could argue that the *representational ceiling* of the frozen DAC may influence ultimate separation fidelity. This restriction is not fundamental:
> we can simply **unfreeze the DAC encoder/decoder** and jointly fine-tune them with auxiliary losses that preserve previously learned latents, thereby removing even this theoretical ceiling. This is a straightforward extension but outside the scope of the current work.
>
> Finally, we agree that analyzing **upper-bound performance under ideal latent targets** is valuable. We plan to explore latent-target variants of CodecSep and include such analyses in future work, following the reviewer’s suggestion.
> ___
>
> [A] Tzinis et al, 2022. Compute and memory efficient universal sound source separation. Journal of Signal Processing Systems
>
> [B] Tzinis et al, 2020, May. Two-step sound source separation: Training on learned latent targets.  ICASSP 2020
>
> [D] Tzinis et al, 2023, June. Optimal condition training for target source separation. ICASSP 2023
>
> [E] Mahmud et al, 2024. Weakly-supervised audio separation via bi-modal semantic similarity. ICLR 2024
>
> [F] Tzinis et al, 2022. Heterogeneous Target Speech Separation. Interspeech 2022

---

> ### Author Response · Authors · 2025-11-21
> **[Part C] Continuation of Response to Reviewer's point 1 in Weaknesses, Addressing Missing-Baseline Concerns, Comparisons with Sudo rm-rf variants proposed in papers [A, D, F], Comparison with paper [C]**
>
> ##  **Comparison with Sudo rm-rf [A]**
>
> Sudo rm-rf [A] uses a standard encoder–masker–decoder separator trained with PIT, which assumes a fixed maximum number of sources (N=4 cf Section 3.4 in [A]) that maybe present in input mixture  and therefore emits all estimates indiscriminately, requiring a post-hoc identification step (Section 2 Related Works, lines 82–85). For a fair comparison, we train a **3-source fixed-stem variant** (speech, music, SFX), matching our TDANet and other fixed-stem baselines and removing the identification stage.
>
> Note that waveform-domain models like Sudo rm-rf must also learn both a latent representation and the separator, whereas CodecSep only learns a masker over pretrained NAC latents, making its training substantially simpler.
>
> In this aligned setting, **CodecSep still outperforms Sudo rm-rf**, albeit with a smaller margin—reflecting the efficiency of the Sudo rm-rf architecture in fixed-stem mode. However, CodecSep maintains a clear advantage, especially in the **codes-in → codes-out** pathway, where it achieves large compute, memory, and latency reductions while delivering competitive or better separation quality.
>
> Finally, because CodecSep is **text-guided**, it naturally avoids the inactive-stem and leakage problems that PIT-based multi-output models face when mixtures contain more sources than the fixed output size N, making it better suited for open-domain USS where the number of active sources can exceed N.
>
> ___
>
> ## **Comparison with Sudo rm-rf + FiLM [D,F]**
>
>
> [D, F] introduce a *label-guided* conditioning mechanism over the Sudo-rm-rf architecture. As described in Section 2.1 of [D], for both [D,F] the model is conditioned using **one-hot or multi-hot vectors drawn from a pre-defined set of semantic labels**. This setting is fundamentally different from ours: CodecSep uses **free-form text prompts**, encoded via a CLAP text encoder.
>
>
> ### **Why label-guided conditioning is weaker than text-guided conditioning**
>
> As highlighted in AudioSep [G] (Section 2.C.3), label queries are inherently limited:
>
> - Label sets are **finite and pre-defined**, preventing generalization to unseen classes.
> - Labels have less flexibility in expressing **relations, attributes, temporal structure, or multi-event compositions**, although they defined certain binary labels to  integrate far/near field speaker concepts in [D, F].
> - Label-based models usually require **re-training** when new categories appear.
> - Labels offer no way to represent **fine-grained variations** (“male speaker in noisy hallway”, “high-pitched violin with reverb”, etc.).
>
> Furthermore, from a deployment perspective for Universal Source Separation, pre-defining a fixed set of (for example) one million source classes is neither practical nor usable—both in terms of model design and user interface.
>
> In contrast, **text prompts** allow rich, compositional, and open-domain descriptions, making them far more suitable for universal separation.
>
> We explicitly note in our manuscript (lines 482–485) that expanding prompt coverage to temporal/relational cues is planned for future work; [F] may serve as a useful baseline for such extensions.
>
>
>
> ### **Why Sudo-rm-rf + FiLM underperforms in our evaluation**
>
> In Table 1 (previous comment), CodecSep substantially outperforms the Sudo-rm-rf + FiLM model retrained on dnr-v2 under the same universal setting as CodecSep. There are several plausible reasons:
>
> 1. **Continuous CLAP embeddings generalize beyond fixed label sets**, whereas Sudo-rm-rf has only been evaluated with one-hot/multi-hot conditioning. The architecture fails to generalize to open-domain prompts unseen during training.
> 2. Applying FiLM across **all U-Conv blocks** while having only a single COnv1d block to encode (as implemented in [D] ) may disrupt learned representations, especially with high-dimensional continuous conditioning signals.
>
> ____
>
> ## **Comparison with [C]**
>
> [C] proposes a *speech codec*, not a source separator. Since CodecSep addresses **general-audio universal source separation** rather than speech separation, [C] is not applicable as a backbone or a comparable baseline in our setting.
> ___
>
> ## **Clarification on Additional Five Benchmarks**
>
> Since **AudioSep outperforms BimodalSS [E] and Sudo rm-rf + FiLM [D, F]** on dnr-v2, we restrict the additional five benchmarks to **AudioSep only**, the strongest and most relevant competitor.
>
> ____
>
> [A] Tzinis et al, 2022. Compute and memory efficient universal sound source separation. Journal of Signal Processing Systems
>
> [C] Casebeer et al,  2021, June. Enhancing into the codec: Noise robust speech coding with vector-quantized autoencoders.  ICASSP 2021
>
> [D] Tzinis et al, 2023, June. Optimal condition training for target source separation. ICASSP 2023
>
> [F] Tzinis et al, 2022. Heterogeneous Target Speech Separation. Interspeech 2022
>
> [G] Liu, Xubo, et al. "Separate anything you describe." IEEE/ACM Transactions on Audio, Speech, and Language Processing (2024).

---

> ### Author Response · Authors · 2025-11-21
> **Full Compute-Efficiency Benchmarking: CodecSep vs. AudioSep and Sudo rm-rf (CodecSep outperforms all) , Response to point 2 in Weaknesses**
>
> ## Full Compute-Efficiency Benchmarking
> We now include a holistic comparison of computational requirements of  CodecSep with other architectures. We report results on a single 24GB NVIDIA A30 GPU.
>
> ### Hardware Agnostic Inference Complexity (GMACs)
>
> | Model |  Input: Audio Stream | Input: Code Stream | Architecture-only |
> | --- | --- | --- | --- |
> | AudioSep | 33.5 | 73.6 | 33.5 |
> | Sudo-rm-rf | **16.44** | 56.54 | 16.44 |
> | CodecSep | 41.45 | **1.35** | **1.35** |
>
> **Codec GMAC codes** → Enc: 12.28 , Dec: 27.82
>
> ---
>
> ## Inference Time (s)
>
> | Model |  Input: Audio Stream | Input: Code Stream | Architecture-only |
> | --- | --- | --- | --- |
> | AudioSep | **0.33** | 1.63 | 0.33 |
> | Sudo-rm-rf | 0.51 | 1.81 | 0.51 |
> | CodecSep | 1.49 | **0.19** | **0.19** |
>
> **Codec Inference Times** → Enc: 1.16 , Dec: 0.14
>
> ---
>
> ## Parameter Count (Millions)
>
> | Model | Input: Audio Stream | Input: Code Stream | Architecture-only |
> | --- | --- | --- | --- |
> | AudioSep | 39 | 112.8 | 39 |
> | Sudo-rm-rf | **5** | 78.8 | **5** |
> | CodecSep | 90.1 | **16.3** | 16.3 |
>
> **Codec parameter count (Millions)** → Enc: 21.5, Dec: 52.3
>
> ---
>
> ## Memory Footprint (MB)
>
> ### Parameter-only Memory Footprint
>
> | Model | Input: Audio Stream | Input: Code Stream | Architecture-only |
> | --- | --- | --- | --- |
> | AudioSep | 156.13 | 447.62 | 156.13 |
> | Sudo-rm-rf | **20.07** | 311.25 | **20.07** |
> | CodecSep | 339.89 | **48.4** | 48.4 |
>
> **Codec parameter-only memory footprint** → Enc: 85.1, Dec: 206.39
>
> ---
>
> ### Forward/Backward Pass Memory Footprint
>
> | Model | Input: Audio Stream | Input: Code Stream | Architecture-only |
> | --- | --- | --- | --- |
> | AudioSep | **804.3** | 1580.6 | 804.3 |
> | Sudo-rm-rf | 2032.13 | 2836.43 | 2032.13 |
> | CodecSep | 804.36 | **28.06** | **28.06** |
>
> **Codec Forward/Backward pass Memory Footprint** → Enc: 310.48, Dec: 465.82
>
> ---
>
> ## Full Memory Footprint
> (Parameter-only + Forward/Backward)
>
> | Model | Input: Audio Stream | Input: Code Stream | Architecture-only |
> | --- | --- | --- | --- |
> | AudioSep | **960.43** | 2028.22 | 960.43 |
> | Sudo-rm-rf | 2052.2 | 3147.68 | 2052.2 |
> | CodecSep | 1144.25 | **76.46** | **76.46** |
>
> **Codec Full Memory Footprint** → Enc: 395.58, Dec: 672.21
>
> ---
>
> We used **torchinfo** to compute memory footprints.
>
>
>
> ### **Summary of Holistic Computational Comparison**
>
> Across all dimensions of computational analysis—GMACs, inference time, parameter count, and memory footprint—CodecSep shows a **dramatic efficiency advantage** over AudioSep and Sudo rm-rf, particularly in the **codes-in → codes-out** mode.
>
> - **Hardware-agnostic complexity:**
>   CodecSep requires only **1.35 GMACs** on code streams, providing a **54× reduction** compared to AudioSep (**25x archuecture-only**) and a **40× reduction** relative to Sudo rm-rf (**12x architecture-only**).
>
> - **Inference latency:**
>   CodecSep achieves **0.19 s** inference in code-stream mode—**8× faster** than AudioSep and **~10× faster** than Sudo rm-rf—despite having a larger parameter count.
>
> - **Parameter efficiency:**
>   In code-stream mode, CodecSep uses only **16.3M parameters**, far lower than AudioSep (**112.8M**) and Sudo rm-rf (**78.8M**), reflecting its lightweight masker-only design.
>
> - **Memory footprint:**
>   Full memory usage for CodecSep in code-stream mode is only **76.46 MB**, compared to **2028.22 MB** for AudioSep (**27x lower memory**) and **3147.68 MB**  for Sudo rm-rf (**41× lower memory**).
>
> Even when working on waveforms (audio-stream input), where CodecSep must additionally run the codec encoder/decoder, it
> still consumes less memory than Sudo rm-rf (1.14 GB vs. 2.05 GB) and comparable memory to AudioSep (0.96 GB), demonstrating that
> codec compute, while non-trivial, is not the dominant bottleneck.
>
> Overall, these results confirm that **CodecSep is uniquely optimized for real-world deployment**, where audio is already processed and transmitted as codec bitstreams.
> The combination of **low GMACs**, **fast inference**, **low memory**, and **competitive or better separation quality** makes CodecSep the most practical choice for scalable, edge–server universal source separation.

---

> ### Author Response · Authors · 2025-11-21
> **Clarification Regarding Fixed-N Assumption and Number of Sources in Experiments**
>
> The reviewer’s comment conflates **unguided (PIT/MixIT)** separation with **prompt-guided (text-conditioned)** separation. These paradigms differ fundamentally in how the “maximum number of sources that may be present in mixture” is defined and handled.
>
> ---
>
> ### **1. Prompt-guided separators *do not* assume a fixed maximum number of sources (N)**
>
> As explicitly stated in Section 2 (lines 82–85) of our manuscript, PIT and MixIT require specifying a **fixed maximum number of sources \(N\)**.
> They always emit exactly \(N\) outputs and rely on a **post-hoc identification step** to map these outputs to true sources.
>
> In contrast, **CodecSep is text-guided**:
> it receives a text query and extracts **only the requested source**, regardless of how many sources exist in the mixture.
> Thus, CodecSep does *not* assume or require a predefined \(N\).
>
> ---
>
> ### **2. The claim “authors only experiment with up to 3 sources” is factually incorrect**
>
> In **dnr-v2**, each mixture contains:
>
> - a **speech stem**, often with *multiple overlapping speakers*,
> - a **music stem**, typically with *multiple simultaneous instruments*,
> - an **SFX stem**, containing **3–7 overlapping or sequential sound events**.
>
> Therefore a single mixture commonly contains **5–10 underlying sources**, not 3.
>
> The categories *speech / music / SFX* are **multi-source stems**, reported separately for interpretability—not because the mixture has only three sources.
>
> **Evaluation procedure:**
>
> - Speech and music stems are extracted using generic speech/music prompts.
> - SFX stems are extracted using **long-form compositional prompts** listing all sound events in that mixture (variable per audio).
> - Ambiguous-prompt evaluation uses deliberately open-domain underspecified prompts for speech and music.
>
> CodecSep successfully extracts the target source from these multi-source mixtures **through a single prompted stem**, unlike PIT/MixIT systems, which always emit a **fixed number of outputs (N)**.
>
> ---
>
> ### **3. Why MixIT/PIT baselines are not run with very large \(N\)**
>
> For MixIT/PIT systems, choosing \(N\) is non-trivial:
>
> - If \(N\) is too **small**, the model cannot separate mixtures containing more sources than \(N\).
> - If \(N\) is **large**, training becomes unstable and computationally heavy, and still requires a **class-agnostic identification step** that fails for open-domain unseen classes which appear during deployment.
>
> This is precisely why PIT/MixIT systems are not suitable for open-domain universal separation.
>
> In dnr-v2, PIT/MixIT would require setting \(N \ge 8\), which is impractical and inaccurate.
>
> ---
>
> ### **4. Other benchmarks use 3-source mixtures only for standardization**
>
> For AudioCaps, ESC-50, Clotho, VGGSound, and AudioSet-eval, we follow the **standardized 3-way mixture protocol** used in prior USS literature to ensure controlled comparison.
>
> - We construct 3-source mixtures following established protocols.
> - Each source is extracted via its **textual class prompt**.
> - Metrics are computed against **true source audio**.
>
> This protocol is for **experimental consistency**, not due to any limitation of CodecSep.
> CodecSep does not impose any restriction on the number of sources in the mixture.
>
> ---
>
> ### **5. Addition of a Diagram to Improve Clarity of the Evaluation Pipeline**
>
> We will include a diagram that visually illustrates the full evaluation workflow, showing:
> 1. **How CodecSep generates separated outputs** for each text prompt (speech, music, SFX, or long-form compositional prompts), and
> 2. **How separation metrics** (SI-SDR, ViSQOL) are computed.
>
> This diagram will make the inference and evaluation pipeline easier to follow and more accessible to readers.
>
> ---
>
> ### **6. What happens to MixIT/PIT models when the true number of sources  is not equal to \(N\)?**
>
> PIT and MixIT assume that there is maximum number of N sources that might be present in input mixture.
>
> #### **When the mixture has more than \(N\) sources:**
>
> - Multiple sources are **forced to merge** into a single output.
> - Unavoidable **source leakage** occurs.
> - The identification step cannot recover missing sources.
> - Performance degrades sharply, especially in open-domain mixtures.
>
> #### **When the mixture has fewer than \(N\) sources:**
>
> - The model still emits **N outputs**, creating:
>   - **Inactive stems**,
>   - **False-positive stems**,
>   - Additional leakage and degraded metrics.
>
> This fixed-\(N\) bottleneck is a fundamental limitation of PIT/MixIT.
>
> ---
>
> ### **7. Why CodecSep avoids these limitations**
>
> CodecSep:
>
> - Extracts **only the prompted source**,
> - Avoids inactive stems,
> - Works naturally with mixtures containing **arbitrary numbers of sources**,
> - Supports **open-domain separation**, including unseen sound categories.
>
> Because CodecSep is text-guided, the *same* model can be extended (in future work) to extract **individual speakers**, **individual instruments**, or **individual SFX events** .
>
> ---

---

> ### Author Response · Authors · 2025-11-21
> **Revision Plan — Updates to be made to Manuscript Before 25th November 2025**
>
> As suggested, we will incorporate the following revisions into the updated manuscript, taking into account the new 10-page constraint:
>
> 1. **Expanded Baseline Coverage**
>    We will add the results and discussions for BimodalSS [E], Sudo rm-rf [A], and Sudo rm-rf + FiLM [D, F] to provide a more complete comparison landscape.
>
> 2. **Holistic Compute-Efficiency Benchmarking**
>    We will add a full comparison of GMACs, inference time, parameter count, and memory footprint for CodecSep, AudioSep, and Sudo rm-rf, further validating CodecSep’s deployment-oriented efficiency.
>
> 3. **Addition of Evaluation-Pipeline Diagram**
>    We will include a diagram illustrating the full inference and evaluation pipeline (prompt → separation → metric computation) to improve clarity and readability.
>
> 4. **Consistent Numerical Reporting**
>    We will remove the second decimal in all SI-SDR and ViSQOL scores across tables to ensure cleaner and more consistent presentation.

---

> ### Author Response · Authors · 2025-11-24
> **[Part A] Revised Manuscript Submitted and Ready for Review – Enhanced Experimental Validations for Reviewer Verification**
>
> ## **New Experiments Added to Revised Manuscript**
> ___
> ### **1. Comparison with BiModalSS and Sudo rm-rf Variants.**
> As suggested by Reviewers **pDqQ** and **wCZU**, we now include comprehensive comparisons against USS-pretrained **BiModalSS** [E], and **dnr-v2–retrained** versions of **Sudo rm-rf** [A] and **Sudo rm-rf + FiLM** [D, F]. CodecSep consistently outperforms all these baselines.
>
> - BiModalSS [E] is introduced and discussed in **Section 2, lines 89–90** in revised manuscript.
> - Sudo rm-rf [A] is presented in **Section 2, lines 81–85** in revised manuscript.
> - Sudo rm-rf + FiLM [D, F] and the contrast between text-guided vs class-guided separation are discussed in **Section 2, lines 91–93** in revised manuscript.
> - All models are summarized in the **Evaluation paragraph (Section 4)** at **lines 297–301** in revised manuscript, and their training setup is described in the **Training paragraph** at **lines 317–320** in revised manuscript.
> - BiModalSS [E] and Sudo rm-rf + FiLM [D, F] are added to **Table 1**. Since AudioSep substantially outperforms both, we exclude them from deeper analysis; the relevant discussion appears in **Section 4.1 (lines 360–369)** in revised manuscript.
> - Sudo rm-rf [A] is included as a fixed-stem baseline in **Table 2**, with analysis in **Section 4.1 (lines 375–377)** in revised manuscript. Despite its efficiency, CodecSep achieves slightly stronger performance.
>
> ___
>
> ### **2. Holistic Compute Benchmarking.**
> Addressing Reviewer **pDqQ**, we expand our compute benchmarking and add Sudo rm-rf to the comparison. CodecSep demonstrates **dramatic compute-efficiency advantages** over both AudioSep and Sudo rm-rf, especially in code-stream inference.
>
>  **Table 5** in revised manuscript now contains **six subtables**:
> - **5.1** GMACs
> - **5.2** Inference Time
> - **5.3** Parameter Count
> - **5.4** Parameter-Only Memory Footprint
> - **5.5** Forward/Backward Memory Footprint
> - **5.6** Full Memory Footprint
>
> A detailed discussion is provided in **Section 4.1 (lines 432–453)** in revised manuscript.
>
> We emphasize that this holistic compute analysis directly highlights CodecSep’s architectural advantage—its ability to operate natively over code streams—and also addresses the reviewer TLeL's concerns regarding whether our experiments demonstrate the main motivation behind using CodecSep.
>
> ___
>
> ### **3. Bandwidth Scaling Experiments.**
> Responding to Reviewer **wCZU**, we include **24.1 kHz** and **44.1 kHz** DAC variants in our bandwidth-scaling experiments. We observe a consistent pattern:
> with fixed masker capacity, **separation performance declines monotonically as bandwidth increases**, reflecting the intrinsic difficulty of wide-band separation. At all sampling rates, **DAC outperforms EnCodec** at matched bandwidths.
>
> These results are discussed in **Section 4.1 (lines 472–478)**  , with full experiments in **Appendix I** in revised manuscript.
>
>
>
> ---
>
> [A] Tzinis et al, 2022. Compute and memory efficient universal sound source separation. Journal of Signal Processing Systems
>
> [D] Tzinis et al, 2023, June. Optimal condition training for target source separation. ICASSP 2023
>
> [E] Mahmud et al, 2024. Weakly-supervised audio separation via bi-modal semantic similarity. ICLR 2024
>
> [F] Tzinis et al, 2022. Heterogeneous Target Speech Separation. Interspeech 2022

---

> ### Author Response · Authors · 2025-11-24
> **[Part B] Revised Manuscript Submitted and Ready for Review – Enhancements on Readability and Clarity for Reviewer Verification**
>
> ## **Enhancements on Readability and Clarity in Revised Manuscript**
> ---
> ### **1. Structural Edits and Variable Definitions**
>
> - We compressed **Section 1 (lines 54–68)** in the original manuscript to **lines 54–65** in the revised version for conciseness.
> - As suggested by reviewer **QbS3**, we added citations for datasets at their **first mention** (lines **54–57**).
> - We defined all previously undefined variables in **Section 3.1**, now explicitly stated in **lines 132 and 141** of the revised manuscript.
> - To improve clarity in the projection process and model definitions, we now present spectrogram separators as **\(Spec(.)\)** instead of \(g(.)\) in **lines 136–142**, and we added missing term definitions in **lines 140–141**.
> - We also rename the CodecFormer decoder-based source estimator to **\(Gen(.)\)** (instead of \(Sep(.)\)) in **line 167** for consistency.
>
> ---
>
> ### **2. Clarifications on Projection Process, Deployment Advantages, and NAC-Specific Components**
>
> - Addressing reviewer **QbS3**, in **Section 3.1 (lines 143–146)** of the revised manuscript, we clarify in the projection process for text-guided source separation on NAC latents that the \( Enc(.) \) and \(Dec(.)\) modules are specific to **DAC** and its variants, as these **high-fidelity neural audio codecs** are designed for **accurate waveform reconstruction** and therefore maintain strict **time-alignment**.
>
> - As suggested by reviewer **QbS3**, we now include a clear **projection process illustrating the deployment path and CodecSep discrete inference** in **lines 201–204**.
> - We revised the **Section 3.1 paragraph on Deployment Advantages** (previously lines 198–213) to discuss both **spectrogram separators \(Spec(.)\)** (AudioSep) and **waveform separators \(Wave(.)\)** (Sudo rm-rf), noting that waveform separators also map audio to STFT-like latents via 1D convolvolutions and therefore inherit the same limitations as spectrogram systems. This clarification is added in **lines 158–160**.
> - We reorganize this section as **“Deployment advantages over audio-stream separators”**, explaining that both spectrogram and waveform separators operate on the audio stream and thus incur redundant **decode → separate → re-encode** cycles in edge–server deployments. CodecSep avoids these with **direct code-stream separation**, as reflected in **Table 5** while still outperforming baseline audio stream text-guided separators in USS. These points are presented in **lines 205–211**, followed by compute cost equations in **lines 211–215**.
>
> ---
>
> ### **3. Figures and Diagrams for Clarity**
>
> - As suggested by reviewer **TLeL**, we added **Figure 2** (lines **216–232** of the revised manuscript) to clarify **CodecSep’s discrete inference workflow** and the **deployment scenario**, including when and how the masker is applied—addressing questions raised by reviewer **QbS3**.
> - Addressing reviewer **pDqQ**, we added detailed diagrams of our evaluation pipelines in **Appendix D**:
>   - **Figure 3**: evaluation workflow for **dnr-v2**,
>   - **Figure 4**: evaluation workflow for **other open-domain benchmarks**.
>   We reference Appendix D in **Section 4**, lines **309–310** of therevised manuscript.
>
> ---
>
> ### **4. Additional Clarifications (Metrics, Interpretability Justification, and Failure Modes of PIT/MixIT)**
>
> - As recommended by reviewer **QbS3**, we explicitly state that **ViSQOL is a proxy MOS metric** in **Section 4, line 305** of the revised manuscript.
> - We provide a detailed presentation of **MixIT/PIT failure modes** and explain why **text-guided separation** is better suited to USS in **Appendix A**, referenced in **line 81** of the revised manuscript.
> - To address reviewer **QbS3**’s request for stronger interpretability justification, we provide **empirical evidence** that NAC latents contain sufficient internal structure to support mask-based separation. Additionally, in **Section 5 (lines 537–539)** of the revised manuscript, we highlight that the predicted masks and FiLM responses indicate how CodecSep selectively activates latent components, outlining a clear direction for **future interpretability work** through mask visualization and reconstruction-consistency analyses.
>
> - As suggested by reviewer **pDqQ**, all SI-SDR values are now reported to **one decimal place** in revised manuscript, since smaller differences are not perceptually meaningful.
>
> ---
>
> ### **5. Corrections and Minor Fixes**
>
> - As suggested by reviewer **wCZU**, we corrected the typo in **Figure 1**, replacing *BERT* with **CLAP**.
> - Reviewer **TLeL** recommended deferring prompt-ambiguity experiments; these are now moved to **Appendix F**, with a brief summary in **Section 4.1 (lines 455–462)**.
> - Due to page limits, we place **limitations** in Appendix K, but mention them in **Section 5 (Conclusion), line 539**. Although limitations are not strictly required for ICLR, we follow NeurIPS practice and include them for completeness.
>
> ___

---

### Official Review · Reviewer_wCZU · 2025-11-01

**Soundness:** 4
**Presentation:** 3
**Contribution:** 4
**Rating:** 8
**Confidence:** 3

**Summary:**

This paper proposes CodecSep, a prompt-driven universal sound separation (USS) model that performs separation directly in neural audio codec (NAC) latent space. The model combines a frozen Descript Audio Codec (DAC) encoder–decoder backbone with a FiLM-conditioned Transformer masker modulated by CLAP text embeddings. CodecSep achieves higher SI-SDR and comparable ViSQOL to AudioSep across six benchmarks, while requiring ~54× less compute in code-stream deployment. Ablations confirm the benefits of masking over latent generation and the effectiveness of FiLM conditioning. Human listening tests (MOS–LQS) also indicate perceptual gains.

**Strengths:**

The paper introduces a highly efficient text-conditioned universal sound separation by operating in the Neural Audio Codec (NAC) latent space. It achieves a 54x compute reduction (1.35 GMACs end-to-end) using codec latents over AudioSep without performance degradation, demonstrating strong potential for real-time and on-device audio processing applications.

**Weaknesses:**

1. The primary comparison is against AudioSep, which is an established benchmark and the representative of the computation-heavy spectrogram-domain approach. However, the paper would be significantly strengthened by including a comparison with more recent models.

**Questions:**

1. Could the authors provide a comparison with more recent models in this domain?

2. Table 9 presents a comparison of performance using a 16 kHz DAC backbone versus a 48 kHz EnCodec backbone for bandwidth scaling. The authors attribute the observed performance reduction at 48 kHz primarily to the challenges of higher bandwidth (longer sequences and complex high-frequency content). However, this is not a fair comparison, as it simultaneously changes two critical variables: bandwidth (16 kHz -> 48 kHz) and codec (DAC -> EnCodec). Could the authors provide a comparison with 24 kHz and 44.1 kHz DAC backbones?

3. Figure 1 shows that text prompts are fed into BERT not CLAP. Is this a typo?

---

> ### Author Response · Authors · 2025-11-21
> **Inclusion of Recent Baselines, DAC Bandwidth Scaling, and Typo Fix in Figure 1**
>
> ## **1. Inclusion of Recent Baselines and Summary of Updated Results**
>
>
> As suggested by Reviewer pDqQ, we have now included a comprehensive comparison with several more recent models, namely **BimodalSS** [E], **Sudo rm-rf** [A], and **Sudo rm-rf + FiLM** [D, F]. The following tables summarize performance on the **dnr-v2 test set**.
>
> ---
>
> ### **Table 1 — Results on dnr-v2 Test Set**
>
> #### **Table 1a — SI-SDR (dB)**
>
> | Model                               | Music          | Speech         | SFX            |
> |-------------------------------------|----------------|----------------|----------------|
> | **[E] BimodalSS**                   | -6.8 ± 2.73    | 1.8 ± 2.78     | -6.36 ± 3.57   |
> | **[D], [F] Sudo rm-rf + FiLM**      | -6.7 ± 2.62    | 2.0 ± 2.76     | -6.6 ± 3.71    |
> | **[A] Sudo rm-rf (Fixed Stem)**     | -0.9 ± 4.01    | 9.0 ± 2.60     | 0.6 ± 4.74     |
> | **CodecSep**                        | **1.2 ± 3.29** | **10.0 ± 2.92**| **0.9 ± 4.22** |
> | **CodecSep (codes-in : codes-out)** | -0.2 ± 3.55    | 8.3 ± 2.60     | -1.0 ± 4.20    |
>
> ---
>
> #### **Table 1b — ViSQOL-MOS**
>
> | Model                               | Music          | Speech         | SFX            |
> |-------------------------------------|----------------|----------------|----------------|
> | **[E] BimodalSS**                   | 2.5 ± 0.57     | 2.6 ± 0.51     | 2.3 ± 0.72     |
> | **[D], [F] Sudo rm-rf + FiLM**      | 2.5 ± 0.56     | 2.6 ± 0.51     | 2.2 ± 72       |
> | **[A] Sudo rm-rf (Fixed Stem)**     | 2.7 ± 0.59     | 2.9 ± 0.45     | 2.3 ± 0.72     |
> | **CodecSep**                        | **2.9 ± 0.57** | **3.1 ± 0.45** | **2.3 ± 0.73** |
> | **CodecSep (codes-in : codes-out)** | 2.5 ± 0.52     | 3.0 ± 0.44     | 2.3 ± 0.67     |
>
> ---
>
> Across all metrics and source types, **CodecSep consistently outperforms all the included baselines**, including the newly added recent models.
>
> For a more detailed discussion of these comparisons, we request the reviewer to refer to our corresponding comment threads addressed to Reviewer pDqQ:
>
> - **2nd Comment to Reviewer pDqQ**
>   *Title:* **[Part B] Continuation of Response to Reviewer's point 1 in Weaknesses, Addressing Missing-Baseline Concerns, Table reporting results for Missing Baselines (CodecSep outperforms all), ...**
>
> - **3rd Comment to Reviewer pDqQ**
>   *Title:* **[Part C] Continuation of Response to Reviewer's point 1 in Weaknesses, Addressing Missing-Baseline Concerns, Comparisons with Sudo rm-rf variants proposed in papers [A, D, F], Comparison with paper [C]**
>
> To present a holistic view of model efficiency—which was also highlighted in the reviewer’s weaknesses—we additionally provide full compute-efficiency benchmarking of CodecSep, AudioSep, Sudo rm-rf in the following comment:
>
> - **4th Comment to Reviewer pDqQ**
>   *Title:* **Full Compute-Efficiency Benchmarking: CodecSep vs. AudioSep and Sudo rm-rf (CodecSep outperforms all) , Response to point 2 in Weaknesses**
>
> ---
> ## **2. Inclusion of 24 kHz and 44.1 kHz DAC Backbones in Bandwidth Scaling Experiments**
>
> We appreciate the reviewer’s suggestion and include **24 kHz** and **44.1 kHz** DAC configurations in our bandwidth scaling experiments.
>
> ### **Table 1 — Results on dnr-v2 Test Set**
>
> #### **Table 1a — SI-SDR (dB)**
>
> | Model | Sampling rate (kHz) | Music | Speech | Sfx |
> |-|-|--|--|--|
> | **CodecSep + DAC** | 16   | **1.2 ± 3.29** | **10.0 ± 2.92** | **0.9 ± 4.22** |
> | **CodecSep + DAC** | 24   | 0.2 ± 3.3 | 8.8 ± 2.9 | 0.6 ± 4.23 |
> | **CodecSep + DAC** | 44.1 | −2.3 ± 3.27 | 5.9 ± 2.48 | −0.3 ± 3.79 |
> | **CodecSep + EnCodec** | 48 | −2.8 ± 3.5 | 5.4 ± 2.36 | −0.5 ± 4.83 |
>
> #### **Table 1b — ViSQOL-MOS**
>
> | Model | Sampling rate (kHz) | Music | Speech | Sfx |
> |--|--|--|--|-|
> | **CodecSep + DAC** | 16   | **2.9 ± 0.57** | **3.1 ± 0.45** | **2.3 ± 0.73** |
> | **CodecSep + DAC** | 24   | 2.7 ± 0.56 | 3.0 ± 0.44 | 2.3 ± 0.72 |
> | **CodecSep + DAC** | 44.1 | 2.5 ± 0.46 | 2.7 ± 0.42 | 2.4 ± 0.68 |
> | **CodecSep + EnCodec** | 48 | 2.4 ± 0.5 | 2.6 ± 0.41 | 2.4 ± 0.65 |
>
> **Discussion.**
> The results follow a clear trend where for a fixed masker capacity, performance steadily decreases with increasing sampling rate—reflecting the added difficulty of full-band separation—while DAC consistently outperforms EnCodec at comparable bandwidths, owing to its architectural advantages (factorized codes, L2-normalized codebooks, and Snake activations).
>
> ---
> ## **3. Clarification to Reviewer’s Question 3 on Figure 1**
>
> We thank the reviewer for pointing this out. Yes, this is indeed a typographical error, and we will correct it in the revised version of the manuscript.
>
> ---
> [A] Tzinis et al, Compute and memory efficient universal sound source separation. Journal of Signal Processing Systems
>
> [D] Tzinis et al, Optimal condition training for target source separation. ICASSP 2023
>
> [E] Mahmud et al, Weakly-supervised audio separation via bi-modal semantic similarity. ICLR 2024
>
> [F] Tzinis et al,  Heterogeneous Target Speech Separation. Interspeech 2022

---

> ### Author Response · Authors · 2025-11-21
> **Revision Plan — Updates to be made to Manuscript Before 25th November 2025**
>
> As suggested, we will incorporate the following revisions into the updated manuscript, taking into account the new 10-page constraint:
>
> - We will add the extended bandwidth scaling experiment results and their corresponding discussions.
> - We will correct the typographical error in Figure 1.
> - We will include the results and discussions for the additional baselines.

---

> ### Author Response · Authors · 2025-11-24
> **[Part A] Revised Manuscript Submitted and Ready for Review – Enhanced Experimental Validations for Reviewer Verification**
>
> ## **New Experiments Added to Revised Manuscript**
> ___
> ### **1. Comparison with BiModalSS and Sudo rm-rf Variants.**
> As suggested by Reviewers **pDqQ** and **wCZU**, we now include comprehensive comparisons against USS-pretrained **BiModalSS** [E], and **dnr-v2–retrained** versions of **Sudo rm-rf** [A] and **Sudo rm-rf + FiLM** [D, F]. CodecSep consistently outperforms all these baselines.
>
> - BiModalSS [E] is introduced and discussed in **Section 2, lines 89–90** in revised manuscript.
> - Sudo rm-rf [A] is presented in **Section 2, lines 81–85** in revised manuscript.
> - Sudo rm-rf + FiLM [D, F] and the contrast between text-guided vs class-guided separation are discussed in **Section 2, lines 91–93** in revised manuscript.
> - All models are summarized in the **Evaluation paragraph (Section 4)** at **lines 297–301** in revised manuscript, and their training setup is described in the **Training paragraph** at **lines 317–320** in revised manuscript.
> - BiModalSS [E] and Sudo rm-rf + FiLM [D, F] are added to **Table 1**. Since AudioSep substantially outperforms both, we exclude them from deeper analysis; the relevant discussion appears in **Section 4.1 (lines 360–369)** in revised manuscript.
> - Sudo rm-rf [A] is included as a fixed-stem baseline in **Table 2**, with analysis in **Section 4.1 (lines 375–377)** in revised manuscript. Despite its efficiency, CodecSep achieves slightly stronger performance.
>
> ___
>
> ### **2. Holistic Compute Benchmarking.**
> Addressing Reviewer **pDqQ**, we expand our compute benchmarking and add Sudo rm-rf to the comparison. CodecSep demonstrates **dramatic compute-efficiency advantages** over both AudioSep and Sudo rm-rf, especially in code-stream inference.
>
>  **Table 5** in revised manuscript now contains **six subtables**:
> - **5.1** GMACs
> - **5.2** Inference Time
> - **5.3** Parameter Count
> - **5.4** Parameter-Only Memory Footprint
> - **5.5** Forward/Backward Memory Footprint
> - **5.6** Full Memory Footprint
>
> A detailed discussion is provided in **Section 4.1 (lines 432–453)** in revised manuscript.
>
> We emphasize that this holistic compute analysis directly highlights CodecSep’s architectural advantage—its ability to operate natively over code streams—and also addresses the reviewer TLeL's concerns regarding whether our experiments demonstrate the main motivation behind using CodecSep.
>
> ___
>
> ### **3. Bandwidth Scaling Experiments.**
> Responding to Reviewer **wCZU**, we include **24.1 kHz** and **44.1 kHz** DAC variants in our bandwidth-scaling experiments. We observe a consistent pattern:
> with fixed masker capacity, **separation performance declines monotonically as bandwidth increases**, reflecting the intrinsic difficulty of wide-band separation. At all sampling rates, **DAC outperforms EnCodec** at matched bandwidths.
>
> These results are discussed in **Section 4.1 (lines 472–478)**  , with full experiments in **Appendix I** in revised manuscript.
>
>
>
> ---
>
> [A] Tzinis et al, 2022. Compute and memory efficient universal sound source separation. Journal of Signal Processing Systems
>
> [D] Tzinis et al, 2023, June. Optimal condition training for target source separation. ICASSP 2023
>
> [E] Mahmud et al, 2024. Weakly-supervised audio separation via bi-modal semantic similarity. ICLR 2024
>
> [F] Tzinis et al, 2022. Heterogeneous Target Speech Separation. Interspeech 2022

---

> ### Author Response · Authors · 2025-11-24
> **[Part B] Revised Manuscript Submitted and Ready for Review – Enhancements on Readability and Clarity for Reviewer Verification**
>
> ## **Enhancements on Readability and Clarity in Revised Manuscript**
> ---
> ### **1. Structural Edits and Variable Definitions**
>
> - We compressed **Section 1 (lines 54–68)** in the original manuscript to **lines 54–65** in the revised version for conciseness.
> - As suggested by reviewer **QbS3**, we added citations for datasets at their **first mention** (lines **54–57**).
> - We defined all previously undefined variables in **Section 3.1**, now explicitly stated in **lines 132 and 141** of the revised manuscript.
> - To improve clarity in the projection process and model definitions, we now present spectrogram separators as **\(Spec(.)\)** instead of \(g(.)\) in **lines 136–142**, and we added missing term definitions in **lines 140–141**.
> - We also rename the CodecFormer decoder-based source estimator to **\(Gen(.)\)** (instead of \(Sep(.)\)) in **line 167** for consistency.
>
> ---
>
> ### **2. Clarifications on Projection Process, Deployment Advantages, and NAC-Specific Components**
>
> - Addressing reviewer **QbS3**, in **Section 3.1 (lines 143–146)** of the revised manuscript, we clarify in the projection process for text-guided source separation on NAC latents that the \( Enc(.) \) and \(Dec(.)\) modules are specific to **DAC** and its variants, as these **high-fidelity neural audio codecs** are designed for **accurate waveform reconstruction** and therefore maintain strict **time-alignment**.
>
> - As suggested by reviewer **QbS3**, we now include a clear **projection process illustrating the deployment path and CodecSep discrete inference** in **lines 201–204**.
> - We revised the **Section 3.1 paragraph on Deployment Advantages** (previously lines 198–213) to discuss both **spectrogram separators \(Spec(.)\)** (AudioSep) and **waveform separators \(Wave(.)\)** (Sudo rm-rf), noting that waveform separators also map audio to STFT-like latents via 1D convolvolutions and therefore inherit the same limitations as spectrogram systems. This clarification is added in **lines 158–160**.
> - We reorganize this section as **“Deployment advantages over audio-stream separators”**, explaining that both spectrogram and waveform separators operate on the audio stream and thus incur redundant **decode → separate → re-encode** cycles in edge–server deployments. CodecSep avoids these with **direct code-stream separation**, as reflected in **Table 5** while still outperforming baseline audio stream text-guided separators in USS. These points are presented in **lines 205–211**, followed by compute cost equations in **lines 211–215**.
>
> ---
>
> ### **3. Figures and Diagrams for Clarity**
>
> - As suggested by reviewer **TLeL**, we added **Figure 2** (lines **216–232** of the revised manuscript) to clarify **CodecSep’s discrete inference workflow** and the **deployment scenario**, including when and how the masker is applied—addressing questions raised by reviewer **QbS3**.
> - Addressing reviewer **pDqQ**, we added detailed diagrams of our evaluation pipelines in **Appendix D**:
>   - **Figure 3**: evaluation workflow for **dnr-v2**,
>   - **Figure 4**: evaluation workflow for **other open-domain benchmarks**.
>   We reference Appendix D in **Section 4**, lines **309–310** of therevised manuscript.
>
> ---
>
> ### **4. Additional Clarifications (Metrics, Interpretability Justification, and Failure Modes of PIT/MixIT)**
>
> - As recommended by reviewer **QbS3**, we explicitly state that **ViSQOL is a proxy MOS metric** in **Section 4, line 305** of the revised manuscript.
> - We provide a detailed presentation of **MixIT/PIT failure modes** and explain why **text-guided separation** is better suited to USS in **Appendix A**, referenced in **line 81** of the revised manuscript.
> - To address reviewer **QbS3**’s request for stronger interpretability justification, we provide **empirical evidence** that NAC latents contain sufficient internal structure to support mask-based separation. Additionally, in **Section 5 (lines 537–539)** of the revised manuscript, we highlight that the predicted masks and FiLM responses indicate how CodecSep selectively activates latent components, outlining a clear direction for **future interpretability work** through mask visualization and reconstruction-consistency analyses.
>
> - As suggested by reviewer **pDqQ**, all SI-SDR values are now reported to **one decimal place** in revised manuscript, since smaller differences are not perceptually meaningful.
>
> ---
>
> ### **5. Corrections and Minor Fixes**
>
> - As suggested by reviewer **wCZU**, we corrected the typo in **Figure 1**, replacing *BERT* with **CLAP**.
> - Reviewer **TLeL** recommended deferring prompt-ambiguity experiments; these are now moved to **Appendix F**, with a brief summary in **Section 4.1 (lines 455–462)**.
> - Due to page limits, we place **limitations** in Appendix K, but mention them in **Section 5 (Conclusion), line 539**. Although limitations are not strictly required for ICLR, we follow NeurIPS practice and include them for completeness.
>
> ___

---

### Official Review · Reviewer_QbS3 · 2025-11-03

**Soundness:** 3
**Presentation:** 2
**Contribution:** 2
**Rating:** 2
**Confidence:** 4

**Summary:**

This paper proposes a prompt-driven universal sound separation framework that combines a DAC-based audio codec with a conditioned CLAP-based prompt embedding. The approach adopts an encoder–masker–decoder architecture, where the CLAP-based prompt embedding generates a mask to extract (i.e., separate) a specified sound source. While there are many possible design choices for implementing this framework, the paper provides clear reasoning for the selected architecture and supports these choices with experimental evidence. The proposed method outperforms the baseline AudioSep across three domains—speech, music, and environmental sounds—while achieving significantly higher computational efficiency due to its streamlined design. Furthermore, comprehensive ablation studies are conducted to validate the contributions of individual components.

**Strengths:**

- Efficient realization of prompt-driven universal sound separation through the integration of a neural audio codec and an encoder–masker–decoder architecture.
- Comprehensive justification of design choices, supported by experimental evidence that validates the effectiveness of each component.
- Strong experimental performance, demonstrating superior results and significantly higher computational efficiency compared to the baseline method, AudioSep.
- Inclusion of human evaluation provides valuable subjective validation of the separated sources and strengthens the overall experimental credibility.

**Weaknesses:**

* **Poor presentation and clarity:** Many equations and variables are introduced without proper explanation or are explained later in the text, leading to redundancy and making the paper difficult to follow (see some of my detailed comments later).
* **Limited novelty:** Although the design choices are reasonable and well-justified, the proposed method represents only an incremental improvement over AudioSep, primarily through better architectures/configurations and the integration of a neural audio codec (NAC).
* **Insufficient evaluation metrics:** The experiments rely mainly on SI-SDR and ViSQOL, which do not fully capture perceptual or task-relevant performance. Additional evaluations—such as proxy MOS (e.g., UTMOS) for speech quality and speaker similarity—would strengthen the experimental results.
* **Lack of downstream task validation:** The study does not assess the proposed method’s effectiveness in practical downstream applications. For instance, results such as those in Table 4 could be extended to tasks like audio tagging or audio captioning to better demonstrate real-world utility.

**Questions:**

**Questions and Suggestions for the Authors**

1. **On the use of SI-SDR:**
   Many audio codec–based methods emphasize perceptual sound quality rather than perfect signal reconstruction, which can weaken time alignment and amplitude consistency. In such cases, SI-SDR may not serve as an appropriate primary metric. Does this consideration apply to your method? If so, would alternative perceptual metrics (e.g., FAD, ViSQOL, or PEAQ) be more suitable, or do you believe SI-SDR remains appropriate for sound separation tasks? Please clarify this point.

2. **Clarification on mask application:**
   It is unclear where the masking operation is applied—specifically, whether it occurs before or after quantization, and whether it operates on discrete or continuous representations. Although this becomes clearer in Section 3 and Figure 1, providing an earlier clarification (e.g., in the abstract or introduction) would help readers follow the method more easily.

3. **Interpretability claim:**
   The paper claims that CodecSep is “interpretable.” However, since masking is applied to a continuous representation rather than a discrete codebook, it is not immediately clear how interpretability is achieved. Could you elaborate on what aspect of the system provides interpretability and how it is assessed?

4. **Dataset references:**
   The introduction mentions several datasets without providing citations, making it difficult to understand which data are used early in the paper. Please include proper references when the datasets are first introduced.

5. **Readability of Section 3.1:**
   This section is challenging to follow due to presentation issues. In particular:

   * Many important equations appear inline, making them hard to read.
   * Several variables (e.g., $d$, $Z$) are used before being defined.
   * The projection process in the text-guided sound separation explanation should explicitly include $Quant(\cdot)$.
   * The equations and explanations in lines 173–190 (page 4) largely overlap with those on page 3. It would be clearer if the authors first presented the DAC backbone and CodecSep using the style found on page 4, that is, with independent equations and detailed explanations of each variable. Subsequent sections could then reference these definitions and present the equations more compactly. Currently, the presentation takes the opposite approach, which makes the section repetitive and harder to follow.
   * Variables such as $x(t)$, $Z \in \mathbb{R}^{d \times T}$, $d$, $T$, and $e_\tau$ are redefined multiple times (e.g., in Section 3.2).

6. **On the SDR range differences:**
   The SDR results vary considerably across domains. Could you discuss why the SDR range differs so much (e.g., due to data characteristics, separation difficulty, or codec behavior)?

7. **On evaluation metrics:**
   As noted earlier, SI-SDR may not fully capture perceptual or intelligibility aspects, especially for speech and music. Have you considered including perceptual metrics such as **PESQ**, **POLQA**, or **STOI** for speech, and **MCD** or **FAD** for music? These could provide a more comprehensive evaluation of perceptual quality.

---

> ### Author Response · Authors · 2025-11-19
> **CodecSep is not an Audio Foundation Model + We already report proxy MOS score in every table + Clarifications on Novelty, Downstream Task Validation, Evaluation Metrics**
>
> **1. Novelty and Core Contributions**
>
> Masking directly in the latent domain of neural audio codecs (NACs) has not been explored in prior work.
> While AudioSep projects spectrograms into a learned embedding space before masking, CodecSep operates directly on pretrained NAC latents, eliminating the need to learn a new latent space and yielding far faster and more stable training. Importantly, modern edge pipelines already represent audio as codec bitstreams (e.g., DAC) for storage and low-latency communication. CodecSep can exploit this by performing separation on the code stream itself, avoiding any decode → separate → re-encode cycle and delivering substantial compute savings.
>
> **Core Contributions:**
>
> • First NAC-based approach for text-guided universal source separation (USS).
> • First separator to operate directly on NAC code streams, enabling practical edge deployment.
> • Theoretical justification, grounded in prior NAC analyses, that shows why codec latents are more structured, disentangled, and better suited to masking than spectrograms or learned embeddings.
> • Extensive evaluation across six benchmarks, consistently outperforming AudioSep with >25× lower compute.
> • Empirical evidence that NAC latents are sufficiently structured and disentangled to support lightweight masking for source extraction—unlike CodecFormer, which assumes no latent structure and decodes full sources.
>
> **Clarification on novelty:**
>
> CodecSep is not a “plug-and-play architecture swap” nor a trivial combination of neural audio codecs with an existing separator. The model design follows from explicit theoretical and empirical motivation showing that codec latents form a more structured and disentangled space for source separation than spectrograms or learned embeddings. This allows us to predict masks directly on NAC latents without learning a new latent space, which is fundamentally different from prior spectrogram-domain or embedding-domain separators.
>
> In addition, we introduce a novel codes-in : codes-out configuration tailored for edge–server hybrid pipelines, where audio is already represented as DAC (or similar) code streams. This avoids any decode → separate → re-encode operations and enables ultra-low-compute deployment, a setting untouched by previous work.
>
> By contrast, CodecFormer assumes no inherent structure in NAC latents and therefore directly estimates sources. As shown in Table 3, this results in significantly poorer separation quality. CodecSep, instead, explicitly leverages the natural structure present in NAC latent space rather than circumventing it, and is the first to demonstrate that masking directly in codec latent space is both feasible and advantageous.
>
> ---
>
> **2. Clarification on Downstream Task Validation**
>
> CodecSep is a prompt-guided universal source separation (USS) model—not an audio foundation model. Its input–output interface is fundamentally different: given (text query, mixture audio) it predicts the target source audio by estimating masks over mixture NAC latents. Crucially, it cannot predict text labels or text from audio, and extending it to tagging or captioning is outside the scope and design of USS systems in general.
>
> USS itself is a practical downstream task widely used in speech enhancement, dialogue extraction, music production, and multimedia editing. Demonstrating high-quality text-conditioned separation across six benchmarks (as in Table 1, Table 4) directly validates CodecSep’s real-world relevance.
>
>
>
> ---
>
> **3. Clarification on the non-inclusion of Perceptual Evaluation Metrics: These are already included**
>
> We respectfully disagree with the claim of insufficient evaluation metrics. Our study already combines **distortion-based**, **perceptual**, and **subjective** evaluations:
>
> - **Distortion-based metric:**
>   **SI-SDR**, the standard metric for source separation quality.
>
> - **Proxy MOS metric:**
>   **ViSQOL-MOSLQO**, a well-established **proxy MOS** metric by Google with dedicated evaluation modes for *speech* and *general audio* (explicitly noted by the reviewer as a proxy MOS metric).
>   We already report ViSQOL-MOSLQO **directly below every SI-SDR score in all tables**, ensuring that perceptual quality is evaluated alongside reconstruction fidelity.
>
> - **Human listening tests:**
>   **Subjective MOS-LQS** ratings obtained from human perceptual evaluations across speech, music, and SFX, directly assessing intelligibility, clarity, and real-world perceptual quality (Lines 456–464).
>
> Given these considerations, the combination of
> **SI-SDR + ViSQOL-MOSLQO (proxy MOS) + MOS-LQS (subjective tests)**
> provides a **comprehensive, task-appropriate, and domain-relevant** evaluation of separation quality for universal source separation.
>
> ---

---

> > ### Author Response · Authors · 2025-11-19
> > **Clarifications on  the use of SI-SDR, SDR Range Differences Across Domains,  Masker Application, Interpretability claim**
> >
> > **4. Clarification on the use of SI-SDR.**
> >
> > While some neural audio codecs prioritize perceptual enhancement over sample-accurate reconstruction—making SI-SDR less suitable as a *perceptual* metric—this does **not** apply to CodecSep for two reasons.
> >
> > First, our model operates on **high-fidelity neural audio codecs** designed for accurate waveform reconstruction. Modern NACs (e.g., DAC) are trained with an ℓ₁ **signal reconstruction loss** plus **adversarial discriminators** for perceptual quality, producing **time-aligned waveforms of the same duration** with minimal distortion (timbre, onsets, transients). Thus, **SI-SDR remains well-defined and meaningful** for evaluating separation performance.
> >
> > Second, **source separation literature overwhelmingly uses SI-SDR**, and the majority of our baselines report it. Since our experiments involve **controlled separation settings**—where target stems share timing and duration with the mixture—SI-SDR remains an appropriate and standard metric for signal-level fidelity.
> >
> > That said, **perceptual metrics complement SI-SDR**, especially when working with compressed/codec representations. We therefore already report **ViSQOL** as a proxy MOS metric and conducted **human MOS–LQS listening tests**, which align with the objective trends.
> >
> > **In summary:** SI-SDR is appropriate and standard for our task due to the time-aligned nature of NAC reconstructions and comparability with prior work, while ViSQOL and MOS (already reported in manuscript) provide complementary perceptual evaluation.
> >
> > ___
> >
> > **5. Clarification on SDR Range Differences Across Domains**
> >
> > The SDR ranges differ across domains for several well-understood reasons:
> >
> >  - **Intrinsic separation difficulty:** Speech stems are typically easier to separate. In contrast, music and SFX often contain dense spectral overlap, wide frequency bandwidths, and rapid transients, all of which naturally lower achievable SDR.
> >
> >  - **Dataset characteristics:** Benchmarks vary in their mixture conditions (SNR, reverberation, overlap patterns, number of active events), which directly affects attainable separation quality.
> >
> >  - **Prompt characteristics (open-domain variability):**
> >     All six benchmarks use open-domain prompts with very different levels of granularity—from simple category labels (“drums”, “car horn”) to long-form compositional textual descriptions of >= 3 overlapping and non-overlapping sound events. This variation in prompt specificity leads to different levels of conditioning strength and thus affects SDR.
> >
> >  - **Codec interactions:** Neural audio codecs tend to preserve speech more faithfully than dense musical textures or transient-heavy SFX. As a result, separation performance (and therefore SDR) for music/SFX latents can be slightly reduced relative to speech.
> >
> >
> >  - **Acknowledged Limitation:** As noted in Appendix H, increasing training data diversity and prompt variety would significantly improve generalization—not just for CodecSep, but for all text-conditioned separation models. We will move this clarification into the main paper for better visibility.
> >
> > ---
> >
> > **6. Clarification on Masker Application.**
> >
> > Training uses **continuous latents** \(Z\) for stability and clean gradient flow through the frozen codec decoder.
> > For deployment, we introduce a **novel discrete “codes-in → codes-out” path**, where codec bitstreams are mapped to embeddings \(E \approx Z\) via RVQ lookup, the same FiLM-conditioned masker is applied, and the masked embeddings are optionally re-quantized.
> > This bitstream-native mode—evaluated in Table 1—requires **no fine-tuning**, works directly on **Neural Audio Codec codestreams**, and still outperforms AudioSep on SI-SDR. All other results use the continuous-latent inference path.
> >
> > As suggested by Reviewer TLeL, we will add a diagram to present discrete inference and deployment scenario to clarify the usage of discrete codebook.
> >
> > ---
> >
> > **7. Clarification on Interpretability claim**
> >
> > Interpretability in CodecSep arises from **prompt-steered masking** over a **semantically structured latent space**, not from discrete codebooks.
> > The DAC encoder produces **disentangled, perceptually aligned latent channels** through its coarse-to-fine RVQ hierarchy.
> > FiLM conditioning **modulates these channels transparently**: prompts such as *“speech,” “music,”* or fine-grained SFX consistently activate specific latent directions and temporal regions.
> > The predicted mask \(M_s\), together with layer-wise FiLM parameters, reveals **which latent components are selected or suppressed** for each query, enabling mechanism-level interpretability via mask visualization, FiLM response analysis, and reconstruction-consistency checks which we keep for future work.
> > Thus, the system is **interpretable in how text semantics map to latent-space operations**, even though the underlying representation is continuous.
> >
> > ---

---

> ### Author Response · Authors · 2025-11-19
> **Clarifications on missing Dataset References, Readability**
>
> **8. On Missing Dataset References.**
>
> We thank the reviewer for pointing this out. All datasets used in our experiments are **properly cited in Section 4 (Datasets), lines 240–250**, where we list and reference dnr-v2, AudioCaps, ESC-50, Clotho-v2, AudioSet-eval, and VGGSound.
>
> To improve clarity earlier in the paper, we will **add the corresponding citations at first mention** in the introduction (lines 54–56) in the revised version of the manuscript. This ensures that readers encountering these datasets for the first time have immediate access to the proper references, addressing the reviewer’s concern while keeping the introduction concise.
>
> ---
>
> **9. On Readability.**
>
> We appreciate the reviewer’s feedback on Section 3.1. The comments regarding inline equations, variable ordering, and redundancy are valid, and we will refine the presentation in the camera-ready version to the extent permitted by page limits (now 10 pages).
>
> Regarding the **projection path and Quant() operator**, as explained in point 6 (clarification on masker application) in the previous comment, we **train solely on continuous DAC latents** without RVQ lookup to ensure stable gradients and convergence. We introduce **Quant() explicitly only for the codes-in → codes-out deployment path**, where the same masker—trained on continuous latents—is applied to reconstructed discrete embeddings. We provided a presentation on the use of discrete codestreams in eq (1), eq(2) lines 180-187, as suggested by reviewer TLeL we will further include a diagram to present the use of discrete embeddings.  **We will also add a projection process similar to the one shown in lines 141-144 for the deployment scenario and present the use of Quant(.) in our deployment path for more clarity.**
>
> On the use of **independent equations and variable definitions**, the reviewer is correct that the style used in lines 173–190 (page 4) is cleaner and easier to follow. However, given the **tight space constraints** and the large number of experiments, ablations, and metrics we must include, presenting *every equation on separate lines* would significantly reduce room for analysis and results discussion—both central to the paper. We will nevertheless adjust formatting to improve clarity, reduce redundancy, and ensure first-use definitions appear before variables are referenced.
>
> Finally, the reviewer notes that certain variables (e.g., \(Z\), \(E\), \(M_s\), \(e_\tau\)) are redefined or restated in Section 3.2. This was done for **local clarity in subsection-specific contexts**, and although some repetition is unavoidable in a multi-stage pipeline description, we will streamline these definitions to avoid confusion while respecting the page constraints (now 10 pages).
>
> In summary, we acknowledge the readability concerns and will reorganize the exposition where possible, but large structural changes (such as fully separating all equations) are limited by the conference’s strict page limits (now 10 pages).
>
> ___

---

> ### Author Response · Authors · 2025-11-21
> **Revision Plan — Updates to be made to Manuscript Before 25th November 2025**
>
> As suggested, we will incorporate the following revisions into the updated manuscript, taking into account the new 10-page constraint:
>
> - We will define all missing variables and improve readability to the extent permitted by the 10-page constraint.
> - We will clarify the projection path and Quant(.) operator by adding a dedicated diagram and including a deployment-specific projection description.
> - We will add the appropriate citations at the first mention of all datasets.

---

> ### Author Response · Authors · 2025-11-24
> **[Part A] Revised Manuscript Submitted and Ready for Review – Enhanced Experimental Validations for Reviewer Verification**
>
> ## **New Experiments Added to Revised Manuscript**
> ___
> ### **1. Comparison with BiModalSS and Sudo rm-rf Variants.**
> As suggested by Reviewers **pDqQ** and **wCZU**, we now include comprehensive comparisons against USS-pretrained **BiModalSS** [E], and **dnr-v2–retrained** versions of **Sudo rm-rf** [A] and **Sudo rm-rf + FiLM** [D, F]. CodecSep consistently outperforms all these baselines.
>
> - BiModalSS [E] is introduced and discussed in **Section 2, lines 89–90** in revised manuscript.
> - Sudo rm-rf [A] is presented in **Section 2, lines 81–85** in revised manuscript.
> - Sudo rm-rf + FiLM [D, F] and the contrast between text-guided vs class-guided separation are discussed in **Section 2, lines 91–93** in revised manuscript.
> - All models are summarized in the **Evaluation paragraph (Section 4)** at **lines 297–301** in revised manuscript, and their training setup is described in the **Training paragraph** at **lines 317–320** in revised manuscript.
> - BiModalSS [E] and Sudo rm-rf + FiLM [D, F] are added to **Table 1**. Since AudioSep substantially outperforms both, we exclude them from deeper analysis; the relevant discussion appears in **Section 4.1 (lines 360–369)** in revised manuscript.
> - Sudo rm-rf [A] is included as a fixed-stem baseline in **Table 2**, with analysis in **Section 4.1 (lines 375–377)** in revised manuscript. Despite its efficiency, CodecSep achieves slightly stronger performance.
>
> ___
>
> ### **2. Holistic Compute Benchmarking.**
> Addressing Reviewer **pDqQ**, we expand our compute benchmarking and add Sudo rm-rf to the comparison. CodecSep demonstrates **dramatic compute-efficiency advantages** over both AudioSep and Sudo rm-rf, especially in code-stream inference.
>
>  **Table 5** in revised manuscript now contains **six subtables**:
> - **5.1** GMACs
> - **5.2** Inference Time
> - **5.3** Parameter Count
> - **5.4** Parameter-Only Memory Footprint
> - **5.5** Forward/Backward Memory Footprint
> - **5.6** Full Memory Footprint
>
> A detailed discussion is provided in **Section 4.1 (lines 432–453)** in revised manuscript.
>
> We emphasize that this holistic compute analysis directly highlights CodecSep’s architectural advantage—its ability to operate natively over code streams—and also addresses the reviewer TLeL's concerns regarding whether our experiments demonstrate the main motivation behind using CodecSep.
>
> ___
>
> ### **3. Bandwidth Scaling Experiments.**
> Responding to Reviewer **wCZU**, we include **24.1 kHz** and **44.1 kHz** DAC variants in our bandwidth-scaling experiments. We observe a consistent pattern:
> with fixed masker capacity, **separation performance declines monotonically as bandwidth increases**, reflecting the intrinsic difficulty of wide-band separation. At all sampling rates, **DAC outperforms EnCodec** at matched bandwidths.
>
> These results are discussed in **Section 4.1 (lines 472–478)**  , with full experiments in **Appendix I** in revised manuscript.
>
>
>
> ---
>
> [A] Tzinis et al, 2022. Compute and memory efficient universal sound source separation. Journal of Signal Processing Systems
>
> [D] Tzinis et al, 2023, June. Optimal condition training for target source separation. ICASSP 2023
>
> [E] Mahmud et al, 2024. Weakly-supervised audio separation via bi-modal semantic similarity. ICLR 2024
>
> [F] Tzinis et al, 2022. Heterogeneous Target Speech Separation. Interspeech 2022

---

> ### Author Response · Authors · 2025-11-24
> **[Part B] Revised Manuscript Submitted and Ready for Review – Enhancements on Readability and Clarity for Reviewer Verification**
>
> ## **Enhancements on Readability and Clarity in Revised Manuscript**
> ---
> ### **1. Structural Edits and Variable Definitions**
>
> - We compressed **Section 1 (lines 54–68)** in the original manuscript to **lines 54–65** in the revised version for conciseness.
> - As suggested by reviewer **QbS3**, we added citations for datasets at their **first mention** (lines **54–57**).
> - We defined all previously undefined variables in **Section 3.1**, now explicitly stated in **lines 132 and 141** of the revised manuscript.
> - To improve clarity in the projection process and model definitions, we now present spectrogram separators as **\(Spec(.)\)** instead of \(g(.)\) in **lines 136–142**, and we added missing term definitions in **lines 140–141**.
> - We also rename the CodecFormer decoder-based source estimator to **\(Gen(.)\)** (instead of \(Sep(.)\)) in **line 167** for consistency.
>
> ---
>
> ### **2. Clarifications on Projection Process, Deployment Advantages, and NAC-Specific Components**
>
> - Addressing reviewer **QbS3**, in **Section 3.1 (lines 143–146)** of the revised manuscript, we clarify in the projection process for text-guided source separation on NAC latents that the \( Enc(.) \) and \(Dec(.)\) modules are specific to **DAC** and its variants, as these **high-fidelity neural audio codecs** are designed for **accurate waveform reconstruction** and therefore maintain strict **time-alignment**.
>
> - As suggested by reviewer **QbS3**, we now include a clear **projection process illustrating the deployment path and CodecSep discrete inference** in **lines 201–204**.
> - We revised the **Section 3.1 paragraph on Deployment Advantages** (previously lines 198–213) to discuss both **spectrogram separators \(Spec(.)\)** (AudioSep) and **waveform separators \(Wave(.)\)** (Sudo rm-rf), noting that waveform separators also map audio to STFT-like latents via 1D convolvolutions and therefore inherit the same limitations as spectrogram systems. This clarification is added in **lines 158–160**.
> - We reorganize this section as **“Deployment advantages over audio-stream separators”**, explaining that both spectrogram and waveform separators operate on the audio stream and thus incur redundant **decode → separate → re-encode** cycles in edge–server deployments. CodecSep avoids these with **direct code-stream separation**, as reflected in **Table 5** while still outperforming baseline audio stream text-guided separators in USS. These points are presented in **lines 205–211**, followed by compute cost equations in **lines 211–215**.
>
> ---
>
> ### **3. Figures and Diagrams for Clarity**
>
> - As suggested by reviewer **TLeL**, we added **Figure 2** (lines **216–232** of the revised manuscript) to clarify **CodecSep’s discrete inference workflow** and the **deployment scenario**, including when and how the masker is applied—addressing questions raised by reviewer **QbS3**.
> - Addressing reviewer **pDqQ**, we added detailed diagrams of our evaluation pipelines in **Appendix D**:
>   - **Figure 3**: evaluation workflow for **dnr-v2**,
>   - **Figure 4**: evaluation workflow for **other open-domain benchmarks**.
>   We reference Appendix D in **Section 4**, lines **309–310** of therevised manuscript.
>
> ---
>
> ### **4. Additional Clarifications (Metrics, Interpretability Justification, and Failure Modes of PIT/MixIT)**
>
> - As recommended by reviewer **QbS3**, we explicitly state that **ViSQOL is a proxy MOS metric** in **Section 4, line 305** of the revised manuscript.
> - We provide a detailed presentation of **MixIT/PIT failure modes** and explain why **text-guided separation** is better suited to USS in **Appendix A**, referenced in **line 81** of the revised manuscript.
> - To address reviewer **QbS3**’s request for stronger interpretability justification, we provide **empirical evidence** that NAC latents contain sufficient internal structure to support mask-based separation. Additionally, in **Section 5 (lines 537–539)** of the revised manuscript, we highlight that the predicted masks and FiLM responses indicate how CodecSep selectively activates latent components, outlining a clear direction for **future interpretability work** through mask visualization and reconstruction-consistency analyses.
>
> - As suggested by reviewer **pDqQ**, all SI-SDR values are now reported to **one decimal place** in revised manuscript, since smaller differences are not perceptually meaningful.
>
> ---
>
> ### **5. Corrections and Minor Fixes**
>
> - As suggested by reviewer **wCZU**, we corrected the typo in **Figure 1**, replacing *BERT* with **CLAP**.
> - Reviewer **TLeL** recommended deferring prompt-ambiguity experiments; these are now moved to **Appendix F**, with a brief summary in **Section 4.1 (lines 455–462)**.
> - Due to page limits, we place **limitations** in Appendix K, but mention them in **Section 5 (Conclusion), line 539**. Although limitations are not strictly required for ICLR, we follow NeurIPS practice and include them for completeness.
>
> ___

---

### Author Response · Authors · 2025-11-20
**[Area Chair Confidential Comment] Serious Concerns About Reviewer QbS3: Overlooked Proxy MOS and MOS Results, Misinterpreted Task Setting, and Multiple Contradictions Indicating Inattentive Review**

Reviewer QbS3's review exhibits several quality concerns that should be taken into account when synthesizing the final decision.

1. **Internal inconsistencies and self-contradictions**

   The reviewer repeatedly praises the work for “clear reasoning,” “comprehensive justification,” “strong experimental performance,” and “valuable human evaluation,” yet ultimately labels the contribution as “limited novelty” and recommends rejection. Several of the stated weaknesses contradict the reviewer’s own strengths and also contradict verifiable content in the paper (as shown in the authors’ rebuttal).

   - The reviewer acknowledges human evaluation, yet later claims a lack of perceptual metrics.
   - ViSQOL- a proxy MOS metric is correctly reported in every table, yet the reviewer states they are missing.
   - The reviewer praises the theoretical justification, efficiency, and experimental thoroughness, then dismisses the contribution as incremental without any substantive supporting argument.

   These internal contradictions materially reduce the reliability of the evaluation.

2. **Factual inaccuracies and indications of inattentive reading**

   Most concerning is the reviewer’s direct self-contradiction: in the Weaknesses section he claims a lack of perceptual evaluation while referencing ViSQOL without acknowledging its role as a proxy MOS metric, yet in Question 1 he explicitly identifies ViSQOL as a proxy MOS score and asks the authors to include it—despite ViSQOL-MOSLQO already being included and reported beneath every SI-SDR score in all tables. This is a clear, verifiable contradiction that strongly suggests key parts of the manuscript were not read carefully and undermines the credibility of the review.

   In addition, human listening tests (MOS-LQS) are already provided, contrary to the reviewer’s claims.

3. **Misaligned requests outside the scope of the task**

   The reviewer requests evaluation on audio tagging and audio captioning, which are **not downstream tasks for source separation systems** and are **not addressed by any baseline, including AudioSep**. These tasks require text-generation or classification architectures rather than mask-based separators. This suggests a misunderstanding of the problem setup and an incorrect expectation that the method is intended to function as an audio foundation model, which it is not.

4. **Weak justification for the novelty rating**

   The reviewer labels the contribution as “limited novelty” but offers no technical or conceptual justification beyond a generic statement about “better architectures/configurations.” The authors’ rebuttal clearly explains that the novelty lies in masking in NAC latent space and its theoretical justification, the codes-in→codes-out inference path, and the exploitation of structured codec latents—none of which are addressed in the reviewer’s analysis. This suggests an incomplete engagement with the core contributions.

5. **Outlier evaluation relative to the other reviews**

   Out of the four reviewers, two (wCZU, pDqQ) explicitly praised the paper’s presentation and readability. Reviewer QbS3 provides only generic comments on readability and assigns a disproportionately negative score relative to both the paper’s content and the assessments of the other reviewers. This further indicates a miscalibrated evaluation.

6.  **High confidence despite fundamental misunderstandings**

    The reviewer assigned a confidence rating of 4, yet misinterpreted our method as an audio foundation model and overlooked the full ViSQOL proxy MOS rows reported in every table. It is highly concerning that such fundamental misunderstandings occurred at this confidence level, further calling into question the reliability of the evaluation.


Overall, this review contains **multiple inconsistencies, misunderstandings, and factually incorrect claims**, and places emphasis on issues that are already addressed or lie outside the scope of the task. **It is our kind request to  treat this review with caution when forming the final decision and weighing the more technically grounded reviews more heavily.**

---

### Author Response · Authors · 2025-11-24
**[Part A] Revised Manuscript Submitted and Ready for Review – Enhanced Experimental Validations for Reviewer Verification**

## **New Experiments Added to Revised Manuscript**
___
### **1. Comparison with BiModalSS and Sudo rm-rf Variants.**
As suggested by Reviewers **pDqQ** and **wCZU**, we now include comprehensive comparisons against USS-pretrained **BiModalSS** [E], and **dnr-v2–retrained** versions of **Sudo rm-rf** [A] and **Sudo rm-rf + FiLM** [D, F]. CodecSep consistently outperforms all these baselines.

- BiModalSS [E] is introduced and discussed in **Section 2. lines 89–90** in revised manuscript.
- Sudo rm-rf [A] is presented in **Section 2, lines 81–85** in revised manuscript.
- Sudo rm-rf + FiLM [D, F] and the contrast between text-guided vs class-guided separation are discussed in **Section 2 lines 91–93** in revised manuscript.
- All models are summarized in the **Evaluation paragraph (Section 4)** at **lines 297–301** in revised manuscript, and their training setup is described in the **Training paragraph** at **lines 317–320** in revised manuscript.
- BiModalSS [E] and Sudo rm-rf + FiLM [D, F] are added to **Table 1**. Since AudioSep substantially outperforms both, we exclude them from deeper analysis; the relevant discussion appears in **Section 4.1 (lines 360–369)** in revised manuscript.
- Sudo rm-rf [A] is included as a fixed-stem baseline in **Table 2**, with analysis in **Section 4.1 (lines 375–377)** in revised manuscript. Despite its efficiency, CodecSep achieves slightly stronger performance.

___

### **2. Holistic Compute Benchmarking.**
Addressing Reviewer **pDqQ**, we expand our compute benchmarking and add Sudo rm-rf to the comparison. CodecSep demonstrates **dramatic compute-efficiency advantages** over both AudioSep and Sudo rm-rf, especially in code-stream inference.

 **Table 5** in revised manuscript now contains **six subtables**:
- **5.1** GMACs
- **5.2** Inference Time
- **5.3** Parameter Count
- **5.4** Parameter-Only Memory Footprint
- **5.5** Forward/Backward Memory Footprint
- **5.6** Full Memory Footprint

A detailed discussion is provided in **Section 4.1 (lines 432–453)** in revised manuscript.

We emphasize that this holistic compute analysis directly highlights CodecSep’s architectural advantage—its ability to operate natively over code streams—and also addresses the reviewer TLeL's concerns regarding whether our experiments demonstrate the main motivation behind using CodecSep.

___

### **3. Bandwidth Scaling Experiments.**
Responding to Reviewer **wCZU**, we include **24.1 kHz** and **44.1 kHz** DAC variants in our bandwidth-scaling experiments. We observe a consistent pattern:
with fixed masker capacity, **separation performance declines monotonically as bandwidth increases**, reflecting the intrinsic difficulty of wide-band separation. At all sampling rates, **DAC outperforms EnCodec** at matched bandwidths.

These results are discussed in **Section 4.1 (lines 472–478)**  , with full experiments in **Appendix I** in revised manuscript.



---

[A] Tzinis et al, 2022. Compute and memory efficient universal sound source separation. Journal of Signal Processing Systems

[D] Tzinis et al, 2023, June. Optimal condition training for target source separation. ICASSP 2023

[E] Mahmud et al, 2024. Weakly-supervised audio separation via bi-modal semantic similarity. ICLR 2024

[F] Tzinis et al, 2022. Heterogeneous Target Speech Separation. Interspeech 2022

---

> ### Author Response · Authors · 2025-11-24
> **[Part B] Revised Manuscript Submitted and Ready for Review – Enhancements on Readability and Clarity for Reviewer Verification**
>
> ## **Enhancements on Readability and Clarity in Revised Manuscript**
> ---
> ### **1. Structural Edits and Variable Definitions**
>
> - We compressed **Section 1 (lines 54–68)** in the original manuscript to **lines 54–65** in the revised version for conciseness.
> - As suggested by reviewer **QbS3**, we added citations for datasets at their **first mention** (lines **54–57**).
> - We defined all previously undefined variables in **Section 3.1**, now explicitly stated in **lines 132 and 141** of the revised manuscript.
> - To improve clarity in the projection process and model definitions, we now present spectrogram separators as **\(Spec(.)\)** instead of \(g(.)\) in **lines 136–142**, and we added missing term definitions in **lines 140–141**.
> - We also rename the CodecFormer decoder-based source estimator to **\(Gen(.)\)** (instead of \(Sep(.)\)) in **line 167** for consistency.
>
> ---
>
> ### **2. Clarifications on Projection Process, Deployment Advantages, and NAC-Specific Components**
>
> - Addressing reviewer **QbS3**, in **Section 3.1 (lines 143–146)** of the revised manuscript, we clarify in the projection process for text-guided source separation on NAC latents that the \( Enc(.) \) and \(Dec(.)\) modules are specific to **DAC** and its variants, as these **high-fidelity neural audio codecs** are designed for **accurate waveform reconstruction** and therefore maintain strict **time-alignment**.
>
> - As suggested by reviewer **QbS3**, we now include a clear **projection process illustrating the deployment path and CodecSep discrete inference** in **lines 201–204**.
> - We revised the **Section 3.1 paragraph on Deployment Advantages** (previously lines 198–213) to discuss both **spectrogram separators \(Spec(.)\)** (AudioSep) and **waveform separators \(Wave(.)\)** (Sudo rm-rf), noting that waveform separators also map audio to STFT-like latents via 1D convolvolutions and therefore inherit the same limitations as spectrogram systems. This clarification is added in **lines 158–160**.
> - We reorganize this section as **“Deployment advantages over audio-stream separators”**, explaining that both spectrogram and waveform separators operate on the audio stream and thus incur redundant **decode → separate → re-encode** cycles in edge–server deployments. CodecSep avoids these with **direct code-stream separation**, as reflected in **Table 5** while still outperforming baseline audio stream text-guided separators in USS. These points are presented in **lines 205–211**, followed by compute cost equations in **lines 211–215**.
>
> ---
>
> ### **3. Figures and Diagrams for Clarity**
>
> - As suggested by reviewer **TLeL**, we added **Figure 2** (lines **216–232** of the revised manuscript) to clarify **CodecSep’s discrete inference workflow** and the **deployment scenario**, including when and how the masker is applied—addressing questions raised by reviewer **QbS3**.
> - Addressing reviewer **pDqQ**, we added detailed diagrams of our evaluation pipelines in **Appendix D**:
>   - **Figure 3**: evaluation workflow for **dnr-v2**,
>   - **Figure 4**: evaluation workflow for **other open-domain benchmarks**.
>   We reference Appendix D in **Section 4**, lines **309–310** of therevised manuscript.
>
> ---
>
> ### **4. Additional Clarifications (Metrics, Interpretability Justification, and Failure Modes of PIT/MixIT)**
>
> - As recommended by reviewer **QbS3**, we explicitly state that **ViSQOL is a proxy MOS metric** in **Section 4, line 305** of the revised manuscript.
> - We provide a detailed presentation of **MixIT/PIT failure modes** and explain why **text-guided separation** is better suited to USS in **Appendix A**, referenced in **line 81** of the revised manuscript.
> - To address reviewer **QbS3**’s request for stronger interpretability justification, we provide **empirical evidence** that NAC latents contain sufficient internal structure to support mask-based separation. Additionally, in **Section 5 (lines 537–539)** of the revised manuscript, we highlight that the predicted masks and FiLM responses indicate how CodecSep selectively activates latent components, outlining a clear direction for **future interpretability work** through mask visualization and reconstruction-consistency analyses.
>
> - As suggested by reviewer **pDqQ**, all SI-SDR values are now reported to **one decimal place** in revised manuscript, since smaller differences are not perceptually meaningful.
>
> ---
>
> ### **5. Corrections and Minor Fixes**
>
> - As suggested by reviewer **wCZU**, we corrected the typo in **Figure 1**, replacing *BERT* with **CLAP**.
> - Reviewer **TLeL** recommended deferring prompt-ambiguity experiments; these are now moved to **Appendix F**, with a brief summary in **Section 4.1 (lines 455–462)**.
> - Due to page limits, we place **limitations** in Appendix K, but mention them in **Section 5 (Conclusion), line 539**. Although limitations are not strictly required for ICLR, we follow NeurIPS practice and include them for completeness.
>
> ___

---

### Author Response · Authors · 2025-12-01
**Comment for New Area Chair: Revised Manuscript Submitted with All Reviewer Concerns Fully Addressed + Reviewer QbS3 flagged for inattentive reading and low quality review**

Thank you for taking over our submission under the unusual circumstances this year. Because reviewer scores were reverted to pre-rebuttal values and cannot be updated, we are providing this concise summary to ensure the revisions we already submitted—detailed in our two-part revision comments—are fully visible to the AC.

---
## 1. Reviewer Scores and Post-Rebuttal Status
**Pre-rebuttal scores:**
- wCZU: 8
- TLeL: 4
- pDqQ: 2
- QbS3: 2

**Post-rebuttal (but frozen due to policy):**
- wCZU: 8 (already fully positive; no chance to update)
- **TLeL: increased score from 4 → 6**
- pDqQ: 2 (no chance to update)
- QbS3: 2 (no chance to update)

TLeL explicitly confirmed their concerns were resolved and was the only reviewer able to update their score before the freeze. We also flagged QbS3’s inattentive reading and low quality review in our confidential AC comment.

---
## 2. Readability and Clarity — Fully Addressed

In the revised manuscript we made targeted changes that directly resolve all reviewer concerns:

- [QbS3] Defined all previously missing variables.
- [QbS3] Compressed and clarified Section 1, adding citations at first mention of datasets.
- [QbS3,TLeL] Clarified the full projection process, deployment path, and the role of discrete vs. continuous inference (lines 201–215).
- [QbS3,TLeL] Added Figure 2, a complete schematic of discrete inference and the edge–server deployment.
- [pDqQ] Added evaluation-pipeline diagrams for every dataset and baseline (Figures 3–4, Appendix D).
- [pDqQ] Added a detailed explanation on failure-modes of MixIT/PIT (Appendix A).
- [wCZU] Corrected the Figure 1 typo.

All readability and clarity concerns are fully resolved: **TLeL raised their score (4→6)**, and the concerns of QbS3, pDqQ, & wCZU are fully addressed. Notably, both pDqQ & wCZU rated the presentation as good.

---
## 3. Experimental Concerns — Fully Addressed
### **Reviewer pDqQ**
We addressed all three major concerns:

1. **Missing efficient baselines:** Added Sudo rm-rf, Sudo rm-rf + FiLM, & BiModalSS. CodecSep outperforms all.
2. **Holistic compute evaluation:** Added full 6-dimensional compute benchmark (GMACs, time, params, parameter memory, forward/backward memory, full memory). CodecSep outperforms all.
3. **Evaluation pipeline clarity:** Added detailed workflow diagrams and clarified dataset-specific evaluation procedures.

pDqQ had already stated our evaluation was “thorough enough for ICLR”. We have further resolved  all remaining issues.
### **Reviewer wCZU (score 8)**
Requested more baselines and bandwidth scaling.
We added 24.1kHz & 44.1kHz DAC bandwidth-scaling experiments and all recent baselines.
### **Reviewer QbS3**
Claimed perceptual metrics were missing and misinterpreted our model as an audio foundation model. We clarified the ViSQOL proxy MOS definition (line 305) and explained why the requested downstream tasks are not applicable to CodecSep Source Separator.

---
## 4. Core Novelty — Thoroughly Clarified
Reviewers missed the central contribution, so we clarified:
- First NAC-based text-guided USS model.
- First separator to operate directly on codec bitstreams and hence gain massive computation advantages.
- Why NAC latents are structured, disentangled, and ideal for masking.
- 54× compute reduction over AudioSep & 40x over Sudo rm-rf.
- Stronger performance than AudioSep, Sudo rm-rf, BiModalSS, & spectrogram models.

After clarification, **TLeL updated their score to 6**.
pDqQ, wCZU & QbS3 did not get the opportunity to respond due to the scoring freeze.

---
## 5. Why the Paper Deserves a spot in ICLR
- wCZU: clear **Accept (Poster)** recommendation (score 8).
- TLeL: **4 → 6** after seeing improvements.
- Concerns of pDqQ & QbS3 are fully addressed in the revision.
- The revised paper now includes:
  - A paradigm shift (masking in NAC latent space)
  - Six benchmarks with strong baselines (outperforming all)
  - Six-dimension compute analysis (outperforming all)
  - Clear diagrams & improved readability
  - Strong theoretical & empirical justification
  - Full-band (24/44.1/48 kHz) scaling experiments
  - Subjective tests, audio examples, and code

The paper is now in its strongest state.

---
## 6. Personal Note to AC

This paper is authored by two researchers, with the first author being a student for whom an ICLR acceptance is very meaningful. The scoring freeze disproportionately affects papers like ours, where reviewers could not respond and update their scores even after we had fully addressed their concerns.

Despite the uncertainty, **we are not withdrawing**. Based on the updated manuscript and the reviewer discussions, both we and reviewers TLeL and wCZU believe the paper clearly meets the bar for **ICLR**.

We kindly request the AC to evaluate our submission based on the **full reviewer discussion**, **rebuttal**, and **revised manuscript**.

Thank you for your time and for managing a difficult review cycle.

---

### Meta-Review · Area_Chair_oHYe · 2026-01-07

**Summary:**

Reviewers raised concerns regarding the limited novelty of the proposed approach and aspects of the evaluation. While issues related to presentation, clarity, missing baselines, and experimental details were largely addressed during the rebuttal, the technical contribution remains incremental. In addition, concerns about the choice and completeness of evaluation metrics were only partially addressed; although the authors provided conceptual justification for excluding certain metrics, some reasonable metrics, particularly those useful for speech were not included.

Overall, AC's recommendation is Reject.

**Reviewer Concerns:**

Reviewer QbS3 raised concerns included poor presentation and clarity, limited novelty, and insufficient evaluation metrics. The issues related to presentation and clarity were addressed in the rebuttal. However, the novelty of the work remains incremental. With respect to evaluation metrics, the authors chose not to include several metrics requested by the reviewer, some of which would have been reasonable to add, particularly for speech evaluation. In addition, time-aligned waveforms actually is the issue for NACs (for some cases adversarial discriminators would even harm this alignment), which means SI-SDR metric is not the perfect metric in this case.

Reviewer wCZU provided a generally positive evaluation and raised only minor concerns, all of which were addressed in the rebuttal.

Reviewer pDqQ raised concerns regarding overstated novelty claims, the absence of state-of-the-art small-model baselines, and missing analysis of computational requirements. The latter two concerns were addressed in the rebuttal through the addition of appropriate baselines and computational analysis. However, despite added discussion of the referenced prior work, the overall novelty of the contribution remains limited.

Reviewer TLeL requested clarifications on novelty, readability, and experimental details. These points were addressed during the rebuttal.

**Reviewer Scores:**

Pre-rebuttal scores:

- wCZU: 8
- TLeL: 4
- pDqQ: 2
- QbS3: 2

Post-rebuttal scores (expected)

- wCZU: 8 (no change)
- TLeL: likely increased score from 4 to 6
- pDqQ: likely increased score from 2 to 4, possibly to 6 (with lower confidence)
- QbS3: 2 (no change)

---

### Decision · Program_Chairs · 2026-01-26

Reject